# OPTIMIZED TRADEOFFS FOR PRIVATE MAJORITY ENSEMBLING

## ABSTRACT

We study the problem of computing an $(m\epsilon, \delta)$-differentially private majority of $K$ $(\epsilon, \Delta)$-differentially private algorithms for $m < K$ and $\delta \geq \Delta \geq 0$. Standard methods such as subsampling or randomized response are widely used, but do they provide optimal privacy-utility tradeoffs? Surprisingly, we show that an $(m\epsilon, \delta)$-private majority algorithm with maximal utility can be computed tractably for any $m < K$. Specifically, we introduce Data-dependent Randomized Response Majority (DaRRM), a general privacy framework characterized by a data-dependent noise function $\gamma$ that allows for efficient utility optimization over the class of all private algorithms subject to privacy constraints. By deriving a structural understanding of DaRRM, our novel learning approach is made tractable by reducing infinitely many privacy constraints into a polynomial set. Theoretically, we show that DaRRM enjoys a privacy gain of a factor of 2 over common baselines under i.i.d. teachers and $\delta = 0$. Lastly, we demonstrate the empirical effectiveness of our first-of-its-kind privacy-constrained utility optimization for ensembling labels and gradients from private teachers through applications of private semi-supervised knowledge transfer and private distributed Sign-SGD, highlighting the outstanding performance of our DaRRM framework with an optimized $\gamma$ against several baselines.

## 1 INTRODUCTION

Differential privacy (DP) is a widely applied framework for formally reasoning about privacy leakage when releasing statistics on a sensitive database Erlingsson et al. (2014); Cormode et al. (2018). Differential privacy protects data privacy by obfuscating algorithmic output, ensuring that query responses look similar on adjacent datasets while preserving utility as much as possible Dwork et al. (2006).

Privacy in practice often requires aggregating or composing multiple private procedures. For example, it is common to aggregate multiple private algorithmic or model outputs in methods such as boosting or calibration (Sagi & Rokach, 2018). In federated learning, model training

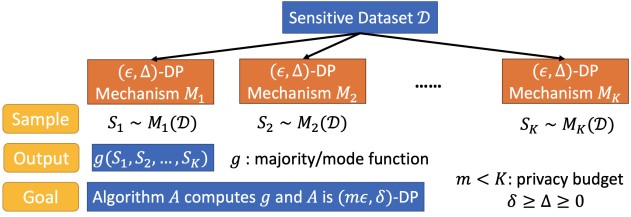

Figure 1: An illustration of the problem setting. The inputs are the dataset $\mathcal{D}$ and $K$ $(\epsilon, \Delta)$-differentially private (DP) mechanisms $M_1, \ldots, M_K$. One draws samples $S_i \sim M_i(\mathcal{D})$ and computes an aggregated output $g(S_1, \ldots, S_K)$ based on all samples seen. Our goal is to design a randomized algorithm $\mathcal{A}$ that approximately computes $g$ and is $(m\epsilon, \delta)$-differentially private for $0 < m < K$ and $\delta \geq \Delta \geq 0$. In this work, we focus on the majority function $g$.

is distributed across multiple edge devices that send locally private training information, such as labels or gradients Konečný et al. (2016) to an aggregating server. When translating from a local privacy guarantee to a centralized one, one needs to reason about the composition of the local privacy leakage Naseri et al. (2020). Thus, we focus on the following ubiquitous setting: Given $K$ $(\epsilon, \Delta)$-differentially private mechanisms, $M_1, \ldots, M_K$, one seeks to compute some aggregation function $g$ applied to the outputs of the mechanisms, releasing $g(M_1(\mathcal{D}), \ldots, M_K(\mathcal{D}))$, while ensuring that the output is $(m\epsilon, \delta)$-differentially private for some private budget $m < K$ and $\delta \geq \Delta \geq 0$ (Figure 1).

For $g$ being the private majority function, it has a wide range of applications in aggregating predictions from a set of models, where each of the models is trained on a private dataset. For example, this occurs in semi-supervised knowledge transfer with private aggregated teacher ensembles (PATE) Papernot et al. (2017; 2018), in ensemble learning algorithms Jia & Qiu (2020); Xiang et al. (2018), and in ensemble feature selection Liu et al. (2018). In the federated setting, the majority aggregation of distributed gradients for private learning is used in algorithms such as Stochastic Sign-SGD Xiang & Su (2023a).

However, some of these works heavily rely on the sensitivity assumption of $g$ for improved privacy bounds and generally provide limited utility guarantees. However, the bounded sensitivity of $g$ can be too pessimistic in practice, as observed in the problem of private hyperparameter optimization (Liu & Talwar, 2019). On the other hand, a naive way to bound privacy loss without restrictive assumptions is to apply simple composition (Theorem B.2) or advanced composition (Theorem B.3) to reason about the final privacy loss after aggregation. A black-box application of the simple composition theorem to compute $g(M_1(\mathcal{D}), \ldots, M_K(\mathcal{D}))$ would incur a $K\epsilon$ privacy cost in the pure differential privacy setting, that is, $\delta = 0$, or if one is willing to tolerate some failure probability $\delta$, advanced composition would yield a $O(\sqrt{K}\epsilon)$ privacy cost Dwork et al. (2014). Thus, a natural baseline algorithm $\mathcal{A}$ that is $(m\epsilon, m\Delta)$-differentially private applies privacy amplification by subsampling and randomly chooses $m$ of the $K$ mechanisms to aggregate and returns the majority of the subsampled mechanisms. This technique is reminiscent of the subsampling procedure used for the maximization function $g$ (Liu & Talwar, 2019) or some general techniques for privacy amplification in the federated setting via shuffling (Erlingsson et al., 2019).

However, standard composition analysis and privacy amplication techniques can be suboptimal for computing a private majority, in terms of both utility and privacy. Observe that if there is a clear majority among the outputs of $M_1(\mathcal{D}), \ldots, M_K(\mathcal{D})$, one can add less noise, since the majority outcome is unlikely to change based on single isolated changes in $\mathcal{D}$. Furthermore, composition theorems make two pessimistic assumptions: 1) the worst-case function $g$ and the dataset $\mathcal{D}$ are considered, and 2) all intermediate mechanism outputs $M_1(\mathcal{D}), \ldots, M_K(\mathcal{D})$ are released, rather than just the final aggregate $g(M_1(\mathcal{D}), \ldots, M_K(\mathcal{D}))$. Therefore, we formally ask the following:

**Problem 1.1** (Private Majority Ensembling (Illustrated in Figure 1)). *Consider $K \geq 1$ $(\epsilon, \Delta)$-differentially private mechanisms $M_1, \ldots, M_K$ for $K$ odd. Given a dataset $\mathcal{D}$, each mechanism outputs a binary answer — that is, $M_i : \mathcal{D} \rightarrow \{0, 1\}, \forall i \in [K]$. Given a privacy allowance $0 < m < K$ and a failure probability $\delta \geq \Delta \geq 0$, how can one find the **optimal utility** of an $(m\epsilon, \delta)$-differentially private mechanism $\mathcal{A}$ to compute the majority function $g(S_1, S_2, \ldots, S_K)$, where $S_i \sim M_i(\mathcal{D})$?*

## 1.1 OUR CONTRIBUTIONS

We give a (perhaps surprising) affirmative answer to this question: we can provably achieve a constant factor improvement in utility over simple subsampling by applying data-dependent noise injection when $M_i$'s are i.i.d. and $\delta = 0$. For general cases, by using our novel data-dependent randomized response framework (DaRRM), which captures all private majority algorithms, we introduce an efficient noise optimization procedure that computes the best possible privacy-utility tradeoffs. To our knowledge, this is the first of its work of its kind that gives a tractable utility optimization over the possibly infinite set of privacy constraints. We detail how our work in the new setting compares to related prior works in Appendix A.

To define utility for evaluating the output of a private majority algorithm $\mathcal{A}$, we will use the error metric $\mathcal{E}$ as the Total Variation (TV) distance between the output distribution of $\mathcal{A}$ and the non-private distribution of $g$, where the randomness is induced by the mechanisms. The utility is then defined as $1 - \mathcal{E}$ and, for optimization purposes, we will consider an average utility over a distribution of the parameters of the underlying mechanisms. The TV distance naturally captures one's intuition that a good private mechanism should have an output close to the true majority.

**Data-dependent Randomized Response Majority (DaRRM).** To motivate our framework, note that a naïve constant $\gamma = O(\frac{m}{K})$ or $\gamma = O(\frac{m}{\sqrt{K}})$ (aka. Randomized Response (RR)) ensures DaRRM to be $m\epsilon$- or $(m\epsilon, \delta)$-differentially private, respectively, but the amount of noise added by this approach, which is determined by the worst case privacy loss, can be too large. The critical observation is that when there is high consensus in the mechanisms' output distributions, we can do exponentially better.

Thus, we propose a general randomized response framework DaRRM (see Algorithm 1) and show that it actually captures all algorithms computing the majority whose outputs are at least as good as a random guess (see Lemma 3.2), including random subsampling (see Lemma 3.1). DaRRM draws a sample $S_i$ from each one of $M_i(\mathcal{D}), \forall i \in [K]$, and faithfully outputs the true majority based on all the samples, i.e. $\mathbb{I}\{\frac{1}{K}\sum_{i=1}^{K} S_i \geq \frac{1}{2}\}$ with probability $\gamma$, while outputting randomly otherwise.

**Designing $\gamma$ with Provable Privacy Amplification.** Our choice of $\gamma$ therefore allows us to explicitly control noise while trading off privacy and utility. As it is data-dependent, we can design $\gamma(\mathcal{L})$ to depend on the observed sum of all mechanisms $\mathcal{L} = \sum_{i=1}^{K} S_i$ and is symmetric around $\frac{K}{2}$, and still ensure privacy in all regimes as long as $\gamma$ does not vary drastically. Using this observation, we show privacy amplification by a factor of 2 in computing the majority of i.i.d mechanisms when $\delta = 0$ through a tighter analysis instead of simply applying simple composition. Specifically, as long as $m \geq \frac{K}{2}$, we can always output the true majority of $K$ mechanisms without any noise addition, and when $m < \frac{K}{2}$, we can provably improve utility, compared to the natural subsampling approach, by a factor of 2 by deriving analytical expressions for tighter privacy-utility tradeoffs (see Theorem 4.1).

**Finding the Best $\gamma$ through Dimension-Reduced Optimization.** Carefully designing $\gamma$ leads to theoretically improved utility, but it is still likely non-optimal. Instead, we exploit the generality of DaRRM by applying a novel optimization-based approach that applies constrained optimization to find the optimal data-dependent $\gamma$ that maximizes some measure of utility; however there are infinitely many privacy constraints. Surprisingly, we show that we can reformulate the privacy constraints, which are infinite dimensional, to a finite polynomial-sized constraint set, allowing us to efficiently constrain the optimization problem to find the best $\gamma$, even for approximate differential privacy (see Lemma 5.1). Empirically, we show that with a small $m$ and $\epsilon$, the optimized $\gamma$ (see Optimized DaRRM$_\gamma$ in Figure 2) achieves the best utility among all $\gamma$ functions, even compared to the subsampling and the data-independent baseline. To our knowledge, this is the first optimal utility guarantee over all private algorithms by constrained optimization with dimension reduction.

In downstream tasks, such as semi-supervised knowledge transfer, we compare our DaRRM with an optimized $\gamma$ to compute the private label majority against PATE Papernot et al. (2018), which indeed has a lower utility as it does not exploit that each mechanism/teacher is $(\epsilon, \Delta)$-differentially private. Furthermore, for collaborative model training using private distributed Sign-SGD on the `MNIST` and `CIFAR10` dataset, we show an improved performance of our optimized DaRRM by $> 8\%$ and $> 3.5\%$ test accuracy on the two datasets, against several baselines with the same privacy guarantee.

## 2 BACKGROUND

Blackbox privacy composition analysis often leads to pessimistic utility guarantees Dwork et al. (2014); Kairouz et al. (2015). Thus, for specific applications, previous work has turned to white-box privacy composition analysis for improved utility. This includes, for example, moment accountant for private SGD Abadi et al. (2016) and the application of contractive maps in stochastic convex optimization Feldman et al. (2018). For the specific case of model ensembles, Papernot et al. (2018) shows a data-dependent privacy bound that vanishes as the probability of disagreement goes to $0$. Their method provides no utility analysis but they empirically observed less privacy loss when there is greater ensemble agreement. When $g$ is the maximization function in aggregating private mechanisms, previous work shows that an approximately maximum value can be outputted with high probability while incurring $O(\epsilon)$ privacy loss, independently of $K$ Liu & Talwar (2019); Papernot & Steinke (2022). Most of these works only claim improved utility and there is no optimality guarantee.

There have been limited works, including Mireshghallah et al. (2020) and Geng & Viswanath (2015),that attempt to derive or learn the best noise distribution. Although our intuition in designing DaRRM also relies on the stability of the mode function $g$, previous usage of stability to improve privacy-utility tradeoffs, e.g., propose-test-release Vadhan (2017); Dwork et al. (2014), requires the testing of such stability, based on which one adds a larger (constant) noise $\gamma$. This can still lead to adding redundant noise in our case. For more related works on this, see a full survey in Appendix A.

### 2.1 PRELIMINARIES

We include formal definitions of differential privacy, simple composition and advanced composition theories in Appendix B. We formalize the error and utility metric as follows:

**Definition 2.1** (Error Metric and Utility Metric). *For the problem setting in Definition 1.1, let the observed (random) outcomes set be $\mathcal{S} = \{S_1, .., S_k\}$, where $S_i \sim M_i(\mathcal{D})$. For a fixed $\mathcal{D}$, we define the error of an algorithm $\mathcal{A}$ in computing the majority function $g$ as the Total Variation (TV) distance between $g(\mathcal{S})$ and $\mathcal{A}(\mathcal{S})$. Specifically, $\mathcal{E}(\mathcal{A}) = \mathbb{E}_{\mathcal{S}}[\mathcal{D}_{TV}(g(\mathcal{S}) \parallel \mathcal{A}(\mathcal{S}))] = \mathbb{E}_{\mathcal{S}}[|\Pr[\mathcal{A}(\mathcal{S}) = 1] - \Pr[g(\mathcal{S}) = 1]|]$. And the utility is defined as $1 - \mathcal{E}(\mathcal{A})$.*

**Notation.** Throughout the paper, we use the same notations defined in Problem 1.1 and Definition 2.1. Furthermore, let $\mathcal{D}$ and $\mathcal{D}'$ to denote a pair of adjacent datasets with one entry being different. Also, let $p_i = \Pr[M_i(\mathcal{D}) = 1]$ and $p'_i = \Pr[M_i(\mathcal{D}') = 1]$, $\forall i \in [K]$. We omit the subscript $i$ when all $p_i$'s or $p'_i$'s are equal. $\mathbb{I}\{\cdot\}$ denotes the indicator function and $[K] = \{1, 2, \ldots, K\}$. For the purpose of analysis, let $\mathcal{L}(\mathcal{D}) = \sum_{i=1}^{K} S_i = \sum_{i=1}^{K} M_i(\mathcal{D})$, i.e. the sum of all observed outcomes on dataset $\mathcal{D}$. Unless specified, we use the function $\gamma : \{0, 1, \ldots, K\} \to [0, 1]$ as input to our algorithms to calibrate the probabilistic noise injection.

## 3 PRIVATE MAJORITY ALGORITHMS

---

**Algorithm 1** DaRRM($\cdot$): Randomized Response Majority

---

1: Input: $K$ $(\epsilon, \Delta)$-DP mechanisms $\{M_i\}_{i=1}^{K}$, $\gamma$ noise function on support $\{0, \ldots, K\}$, dataset $\mathcal{D}$, target privacy cost $m \cdot \epsilon$ for $m < K$, target failure probability $\delta \geq \Delta \geq 0$.
2: $\mathcal{S} = \{S_1, .., S_k\}$, where $S_i \sim M_i(\mathcal{D})$
3: Set probability $p_\gamma \leftarrow \gamma(\sum_{i=1}^{K} S_i)$
4: Flip the $p_\gamma$- biased coin
5: **if** Head (with probability $\gamma$) **then**
6:    Output $\mathbb{I}\{\frac{1}{K}\sum_i S_i \geq \frac{1}{2}\}$
7: **else**
8:    Output $0/1$ with equal probability
9: **end if**

---

To address the problem of private majority ensembling (see Problem 1.1), since the output is discrete, the very first approach to consider is the classical Randomized Response (RR) mechanism Dwork et al. (2014), where one flips a biased coin with a *constant* probability function $\gamma(l) = p_\gamma, \forall l \in \{0, 1, \ldots, K\}$. If the coin is head, we output the true majority base on $K$ samples; if not, then simply output a noisy random answer. However, to make the output $m\epsilon$-differential private, the success probability $p_\gamma$ can be at most $O(\frac{m}{K})$ (or $O(\frac{m}{\sqrt{K}})$) when $\delta = 0$ (or $\delta > 0$) (see Appendix C.1), which is too small for any reasonable utility.

The key observation for improved utility is that the probability of success should not be a *constant*, but should depend on the *unpublished* set of observed outcomes from the mechanisms $\mathcal{S}$. If we see many 1's or 0's in $\mathcal{S}$, then there should be a clear majority even on adjacent datasets. On the other hand, if we see about half 1's and half 0's, this means the majority is highly volatile to data changes, which implies we need more noise to ensure privacy. In summary, if we can calibrate the success probability based on $\mathcal{S}$ to smoothly increase when there is a clear majority, we can improve the utility without affecting privacy.

**Subsampling.** While it seems daunting to design $\gamma$ that is both private and varying, we show that this is indeed possible. One natural baseline is subsampling and outputting the majority of $m$ out of $K$ mechanisms for some $m$. Suppose $\delta \geq m\Delta$, the privacy loss of the aggregated output can be reasoned through simple composition or advanced composition [1]. Interestingly, we show subsampling $m$ out of $K$ mechanisms corresponds to a non-constant polynomial $\gamma$, which we term "$\gamma_{Subsampling}$", in Lemma 3.1 (see a full proof in Appendix C.2). Intuitively, subsampling may be seen as implicitly adding noise by outputting based only a random choice of subset of the mechanism outputs; therefore this implicit noise is inherently *data-dependent* on $\mathcal{L}$.

**Lemma 3.1.** *Consider Problem 1.1, and the sum of observed outcomes of the mechanisms, $l = \sum_{i=1}^{K} S_i \in \{0, 1, \ldots, K\}$. For $m \in \mathbb{Z}^+$, $m \leq K$, if one sets a success probability $\gamma_{Subsampling}$, dependent on the value of $l$, by*

$$\gamma_{Subsampling}(l) = \begin{cases} \gamma_{Subsampling}(K - l) = 1 - 2\sum_{j=\frac{m+1}{2}}^{m} \frac{\binom{l}{j}\binom{K-l}{m-j}}{\binom{K}{m}} & \text{for odd } m \\ \gamma_{Subsampling}(K - l) = 1 - 2\sum_{j=\frac{m}{2}+1}^{m} \frac{\binom{l}{j}\binom{K-l}{m-j}}{\binom{K}{m}} - \frac{\binom{l}{\frac{m}{2}}\binom{K-l}{\frac{m}{2}}}{\binom{K}{m}} & \text{for even } m \end{cases}$$

(1)

---

[1]Deciding on which composition theorem to apply depends on $m$ and $\delta$. When $\delta = 0$, only simple composition applies. When moderate $\delta > 0$, for small $m$, the simple composition indicates less privacy loss; and for larger $m$ or larger $\delta$, the advanced composition is clearly better.

*then outputting the majority of $m$ out of $K$ subsampled mechanisms without replacement and DaRRM$_{\gamma_{Subsampling}}$ have the same output distribution.*

**Data-dependent Randomized Response (DaRRM).** Does subsampling give optimal utility? Inspired by the connection between RR and subsampling, we propose Data-dependent Randomized Response Majority (DaRRM) in Algorithm 1, to study the optimal privacy-utility tradeoffs in private majority ensembling. In particular, DaRRM has a parameterized success probability $p_\gamma$ that depends on the set of observed outcomes $\mathcal{S} = \{S_1, \ldots, S_K\}$. In fact, we can show that DaRRM is general: any *reasonable* algorithm $\mathcal{A}$, name one whose output is at least as good as a random guess, can be captured by the DaRRM framework in Lemma 3.2 (see a full proof in Appendix C.3). We call DaRRM instantiated with a specific noise function $\gamma$ DaRRM$_\gamma$. Note the $\gamma$ function stated in Lemma 3.2 is more general than what we need.

**Lemma 3.2** (Generality of DaRRM). *Let $\mathcal{A}$ be any randomized algorithm to compute the majority function $g$ on $\mathcal{S}$ such that for all $\mathcal{S}$, $\Pr[\mathcal{A}(\mathcal{S}) = g(\mathcal{S})] \geq 1/2$ (i.e. $\mathcal{A}$ is at least as good as a random guess). Then, there exists a general $\gamma: \{0,1\}^{K+1} \to [0,1]$ such that if one sets $p_\gamma$ by $\gamma(\mathcal{S})$ in DaRRM, the output distribution of DaRRM$_\gamma$ is the same as the output distribution of $\mathcal{A}$.*

**Designing the $\gamma$ function.** With the DaRRM framework, we ask how to design a good $\gamma$ function that maximizes the utility? First, to characterize the privacy cost constraint, we introduce two characteristics of $\gamma$ that do not affect the utility, while simplifying the analysis and the empirical optimization. Note that $\gamma_{Subsampling}$ satisfies both characteristics. (a) *A function of the sum of observed samples*: Since the observed samples set $\mathcal{S}$ is a permutation-invariant set, a sufficient statistic that captures the full state of $\mathcal{S}$ is $\mathcal{L} = \sum_i S_i$, the mean vote. This allows us to reduce $\gamma(\mathcal{S}) = \gamma(\mathcal{L})$, and hence, in the rest of the paper, we only consider $\gamma : \{0, 1, \ldots, K\} \to [0, 1]$. (b) *Symmetric around $\frac{K}{2}$*: If $\gamma$ is asymmetric, we can symmetrize by reflecting one region about $\frac{K}{2}$ and achieve better or equal expected utility, where the utility is summed over symmetric distributions of $p_i$. Let $\mathcal{L}(\mathcal{D})$ and $\mathcal{L}(\mathcal{D}')$ denote the sum of observed outcomes on adjacent datasets $\mathcal{D}$ and $\mathcal{D}'$. Also, recall $p_i = \Pr[M_i(\mathcal{D}) = 1]$ and $p_i' = \Pr[M_i(\mathcal{D}') = 1]$ are the output probabilities of the mechanisms on $\mathcal{D}, \mathcal{D}'$. Now, we derive conditions for a $\gamma$ function such that DaRRM$_\gamma$ is $(m\epsilon, \delta)$-differentially private in Lemma 3.3 (see a full proof in Appendix C.4).

**Lemma 3.3** ($\gamma$ privacy condition and privacy cost objective). *Consider using DaRRM to solve Problem 1.1. Let $\alpha_l = \Pr[\mathcal{L}(\mathcal{D}) = l]$ and $\alpha_l' = \Pr[\mathcal{L}(\mathcal{D}') = l]$, for $l$ in support $\{0, \ldots, K\}$ and adjacent datasets $\mathcal{D}, \mathcal{D}'$. For $\gamma : \{0, 1, \ldots, K\} \to [0, 1]$ such that $\gamma(l) = \gamma(K - l), \forall l$, DaRRM$_\gamma$ is $(m\epsilon, \delta)$-differentially private if and only if for all $\alpha_l, \alpha_l'$,*

$$f(p_1, \ldots, p_K, p_1', \ldots, p_K'; \gamma) := \sum_{l=0}^{\frac{K-1}{2}} (e^{m\epsilon}\alpha_l' - \alpha_l) \cdot \gamma(l) + \sum_{l=\frac{K+1}{2}}^{K} (\alpha_l - e^{m\epsilon}\alpha_l') \cdot \gamma(l) \leq e^{m\epsilon} - 1 + 2\delta$$

(2)

*We call $f$ the privacy cost objective.*

## 4 PROVABLE PRIVACY AMPLIFICATION

We theoretically demonstrate that privacy is provably amplified under improved design of $\gamma$ in our DaRRM framework. Specifically, we show when the mechanisms are i.i.d. and $\delta = 0$, we gain privacy amplification by a factor of 2 by carefully designing $\gamma$. Recall $p_i = \Pr[M_i(\mathcal{D}) = 1]$ and $p_i' = \Pr[M_i(\mathcal{D}') = 1]$ are the output probabilities of the mechanisms on adjacent datasets $\mathcal{D}, \mathcal{D}'$.

**Theorem 4.1** (Provable Privacy Amplification by 2). *Consider using DaRRM to solve Problem 1.1 when the mechanisms are i.i.d., i.e., $p_i = p$, $p_i' = p'$, $\forall i \in [K]$ and $\Delta = 0$. Given a privacy allowance $m \in [K]$, if $m \geq \frac{K+1}{2}$, one sets $\gamma(l) = 1, \forall l \in \{0, 1, \ldots, K\}$; and if $m \leq \frac{K-1}{2}$, one sets $\gamma(l) = \begin{cases} 1 - 2h(l) & \forall l \leq \frac{K-1}{2} \\ 2h(l) - 1 & \forall l \geq \frac{K+1}{2} \end{cases}$, where $h(l) = \sum_{i=m}^{2m-1} \frac{\binom{l}{i}\binom{K-l}{2m-1-i}}{\binom{K}{2m-1}}$, then DaRRM$_\gamma$ is $m\epsilon$-differentially private.*

**Interpretation.** First, when $m \leq \frac{K-1}{2}$ is small, the $\gamma(l)$ in Theorem 4.1 corresponds to outputting the majority based on $2m - 1$ outcomes. However, simple composition would have indicated that one can only output the majority based on $m$ outcomes, therefore implying a 2x utility gain. When

$m \geq \frac{K+1}{2}$, the above theorem indicates that we can set a constant $\gamma = 1$, which implies we are optimally outputting the true majority with no noise while still surprisingly ensuring $m\epsilon$ privacy.

This 2x gain is intuitively possible because the majority is only dependent on half of the mechanisms' outputs, therefore the privacy leakage is also halved. To see this, we start by analyzing the privacy cost objective in Eq. 37, where with a careful analysis of its gradient, we show that the maximum indeed occurs $(p^*, p'^*) = (0,0)$ under this assumption. Now, when $(p^*, p'^*) \to 0$, note that the probability ratio of outputting 1 with $2m-1$ outcomes is approximately $e^{m\epsilon}$, where dependence on $m$ follows because the probability of outputting 1 is dominated by the probability that exactly $m$ mechanisms output 1. To rigorize this, we derive sufficient conditions for $\gamma$ functions that satisfy $f(0,0;\gamma) \leq e^{m\epsilon} - 1$ as indicated by Lemma 3.3, to ensure DaRRM to be $m\epsilon$-differentially private and a more detailed overview and the full proof can be found in Appendix D.

## 5   OPTIMIZING THE FUNCTION $\gamma$ IN DARRM

Theoretically designing $\gamma$ and extending privacy amplification results to the $\delta > 0$ case is difficult and it is likely that our crafted $\gamma$ is not even optimal. On the other hand, one can to optimize for such $\gamma^*$ but this involves solving a "Semi-infinite Programming" problem, due to the infinitely many privacy constraints. We show that this is in fact tractable, proposing a novel learning approach based on DaRRM that can efficiently optimize the best noise distribution to achieve maximal utility. To the best of our knowledge, such optimization is the first of its kind and is the following:

$$\min_{\gamma \in [0,1]^{K+1}} \mathbb{E}_{p_1, p_2, \ldots, p_K \sim \mathcal{T}}[\mathcal{E}(\text{DaRRM}_\gamma)] \tag{3}$$

$$\text{s.t.} \max_{\{(p_i, p'_i) \in \mathcal{F}\}_{i=1}^K} f(p_1, \ldots, p_K, p'_1, \ldots, p'_K; \gamma) \leq e^{m\epsilon} - 1 + 2\delta$$

$$\gamma(l) = \gamma(K - l), \forall l \in \{0, 1, \ldots, K\}$$

where $f$ is the privacy cost objective defined in Lemma 3.3. $\mathcal{F}$ is the feasible region where $(p_i, p'_i)$ lies. Since $\gamma$ is symmetric around $\frac{K}{2}$, we only need to optimize $\frac{K+1}{2}$ variables. When we have no prior knowledge about $p_1, \ldots, p_K$, $\mathcal{T}$ is set to be the uniform distribution [2].

**Optimizing Over All Algorithms.** We want to stress that while it is in general hard to optimize for *all* algorithms, since we show in Lemma 3.2 DaRRM that captures *all reasonable* algorithms computing a private majority, we are indeed optimizing over *all* algorithms for maximal utility. Perhaps surprisingly, it turns out that optimizing for $\gamma^*$ is a Linear Programming (LP) problem!

**Linear Optimization Objective.**   Indeed, after expanding the objective by the utility definition (see Definition 2.1), optimizing the above objective is essentially same as optimizing the following objective linear in $\gamma$ (see a full derivation in Appendix E.1): $\min_{\gamma \in [0,1]^{K+1}} -\frac{1}{2} \sum_{l=\frac{K+1}{2}}^{K} \mathbb{E}_{p_1, p_2, \ldots, p_K \sim \mathcal{T}}[(\alpha_l - \alpha_{K-l})]\gamma(l)$.

Although taking the expectation over $p_1, \ldots, p_K$ involves integrating over $K$ variables and this can be computationally expensive, we discuss how to formulate a computationally efficient approximation of the objective in Appendix E.2 and demonstrate the effectiveness of optimization over $\gamma$ with the practical version of the objective in the experiments. Note that the objective only minimizes the utility and hence approximating the objective does not affect the privacy guarantee.

**Reducing Infinitely Many Constraints to A Polynomial Set.** From Eq. 37 the constraint of the optimization problem is linear in $\gamma$. Though it appears we need to solve for infinitely many constraints, we show that through a structural understanding of DaRRM, we can surprisingly reduce the number of privacy constraints from infinitely many to an exponential set, and further to a polynomial set. First, we observe the privacy cost objective $f$ is linear in each independent pair of $(p_i, p'_i)$, hence finding the worst case probability $(p_i^*, p_i'^*) = \arg\max_{(p_i, p'_i)} f$ that causes the maximum privacy loss is a linear programming (LP) problem. Furthermore, since $p_i$ and $p'_i$ are the probability of outputting 1 from the $i$-th $(\epsilon, \Delta)$-differentially private mechanism $M_i$, by definition, they are close and lie in a feasible region $\mathcal{F}$, which we show has 8 corners when $\delta > 0$ (and only 4 corners when $\delta = 0$). This implies $(p_i^*, p_i'^*)$ only happens at one of the corners of $\mathcal{F}$, and hence the number of constraints reduces to $K^8$ (and $K^4$ when $\delta = 0$). Second, observe that $\alpha_l$ in the privacy cost objective $f$ is the

---

[2]Note one also has the flexibility of incorporating prior knowledge about the mechanisms by choosing some prior distribution $\mathcal{T}$ to further improve the utility, if one has any.

pmf of a Poison Binomial (PB) distribution at $l \in \{0, \ldots, K\}$. Notice that PB is invariant under the permutation of its parameters, i.e. $\text{PB}(p_1, \ldots, p_K)$ has the same distribution as $\text{PB}(\pi(p_1, \ldots, p_K))$, under some permutation $\pi$. Based on this observation, we show the number of constraints can be further reduced to $O(K^7)$ (and this is $O(K^3)$ when $\delta = 0$). We formalize the two-step reduction of the number of privacy constraints in Lemma 5.1 (see a full proof in Appendix E.3) as follows [3].

**Lemma 5.1.** *Consider using DaRRM to solve Problem 1.1. Given an arbitrary $\gamma$, let the global worst case probabilities be $(p_1^*, \ldots, p_K^*, p_1'^*, \ldots, p_K'^*) = \arg\max_{\{(p_i, p_i')\}_{i=1}^K} f(p_1, \ldots, p_K, p_1', \ldots, p_K'; \gamma)$, where $f$ is the privacy cost objective defined in Lemma 3.3. Each pair $(p_i^*, p_i'^*)$ satisfies $(p_i^*, p_i'^*) \in \{(0,0), (1,1), (0, \Delta), (\Delta, 0), (1 - \Delta, 1), (1, 1 - \Delta), (\frac{e^\epsilon + \Delta}{e^\epsilon + 1}, \frac{1 - \Delta}{e^\epsilon + 1}), (\frac{1 - \Delta}{e^\epsilon + 1}, \frac{e^\epsilon + \Delta}{e^\epsilon + 1})\}$, $\forall i \in [K]$. Furthermore, there exists a set $\mathcal{P}$ of size $O(K^7)$ such that $(p_1^*, \ldots, p_K^*, p_1'^*, \ldots, p_K'^*) = \arg\max_{\{(p_i, p_i')\}_{i=1}^K \in \mathcal{P}} f(p_1, \ldots, p_K, p_1', \ldots, p_K'; \gamma)$ if $\delta > 0$ and a set $\mathcal{P}$ of size $O(K^3)$ if $\delta = 0$.*

## 6 EXPERIMENTS

We empirically solve[4] the above optimization problem (Eq. 3) using the `Gurobi`[5] solver and first present the shape of the optimized $\gamma$ function and its utility in Section 6.1. Then, we demonstrate the compelling effectiveness of DaRRM with an optimized $\gamma$ function, which we call optimized DaRRM$_\gamma$, in aggregating private mechanisms through two applications: private distributed Sign-SGD in Section 6.2 and private semi-supervised knowledge transfer in Section 6.3.

### 6.1 OPTIMAL $\gamma$ IN SIMULATIONS

We compare the shape and $\mathcal{E}(\text{DaRRM}_\gamma)$ of an optimized $\gamma$, $\gamma_{subsampling}$ (see Lemma 3.1) and the constant data-independent $\gamma$ in randomized response (see Lemma C.1) with $K = 11, \epsilon = 0.1, \Delta = 10^{-5}$ and $m \in \{1, 3, 5, 7\}$. Note when $K, m$ are small as in this case, simple composition indeed indicates a smaller privacy loss than advanced composition. Hence, the subsampling baseline we compare here subsamples $m$ votes from the private mechanisms. Since the failure probability of privacy composition composes linearly, to ensure a fair comparison against the subsampling baseline, we set $\delta = m \cdot \Delta$. That is, for all baselines and our optimized DaRRM$_\gamma$, the output is required to be $(m\epsilon, \delta)$-differentially private. We plot each $\gamma$ functions over the support $\{0, 1, \ldots, K\}$ and the corresponding error of each algorithm in Figure 2.

In summary, the optimized $\gamma$ has a larger magnitude over the support than the baselines. This implies the optimized $\gamma$ has lower error, which is verified on the right set of plots. More results on comparing the optimized DaRRM$_\gamma$ against the subsampling baseline by advanced composition and comparison under pure differential privacy settings (i.e. $\Delta = \delta = 0$) for large $K$ can be found in Appendix F.1.

### 6.2 APPLICATION 1: PRIVATE DISTRIBUTED SIGN-SGD

**Distributed Sign-SGD.** We now demonstrate the performance of our optimized DaRRM$_\gamma$ in aggregating private sign gradients with distributed Sign-SGD, a Byzantine resilient optimization algorithm. Consider a central server and $K$ distributed clients. At each communication round, each client sends the sign of each coordinate of its gradient to the server, and the server wants to aggregate the clients' sign gradients by taking a majority vote of the signs for each coordinate. The server then uses the aggregated sign gradient to update its model and sends the new model back to the clients.

We consider an "honest-but-curious" server which wants to infer clients' information through their sign gradients. And so the clients add noises to each coordinate of their gradients before sending them to the server. We adopt $\beta$ Stochastic Sign-SGD Xiang & Su (2023a;b), a variant of distributed Sign-SGD that ensures coordinate-wise $\epsilon$-differential privacy of the client's gradients. For completeness, we include the algorithm and its privacy guarantees in Appendix F.2.

---

[3]**Practical Limitation.** Although the number of constraints is polynomial in $K$ and optimizing $\gamma$ in DaRRM is an LP, $O(K^7)$ can still make the number of constraints intractably large when $K$ is large. In practice, we observe with the `Gurobi` optimizer, one can optimize $\gamma$ for $K \leq 41$ when $\delta > 0$. But when $\delta = 0$, since the number of privacy constraints is $O(K^3)$, one can optimize for $K$ over 100.

[4]All code for the experiments can be found at `https://anonymous.4open.science/r/optimized_private_majority-E469`

[5]`https://www.gurobi.com/`

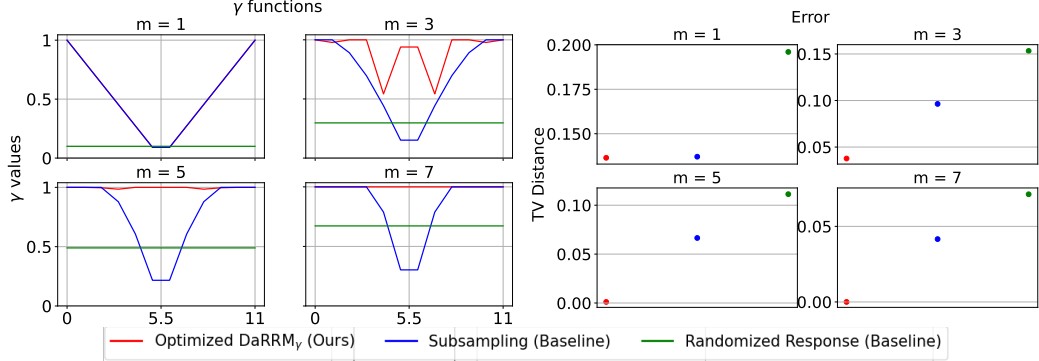

Figure 2: Plots of $\gamma$ functions corresponding to optimized, subsampling, the data-independent $\gamma$ in randomized response and the error in TV distance of the majority ensembling output of DaRRM with different $\gamma$ functions, when $K = 11, m \in \{1, 3, 5, 7\}, \Delta = 1e - 5$ and $\delta = m\Delta$.

We further consider a user who wants to obtain a trained model at the server and we want to ensure privacy of the final model w.r.t. all clients' training data. This requires the server to provide privacy guarantees of the aggregated sign gradient vector at each communication round, which by simple composition, is $K\epsilon$ per coordinate. To limit the per-coordinate privacy loss, we set a per-round privacy allowance of $m$ such that each coordinate of the aggregated signed gradient vector using majority voting is $m\epsilon$-differentially private at the server.

**Baselines.** "Laplacian" Papernot et al. (2017): the most naive baseline to aggregate $\epsilon$-differentially private clients is to add Laplacian noise to the counts of $\pm 1$ and pick the sign with the noisy maximum count. However, this method does not consider the gradient vectors being pri-

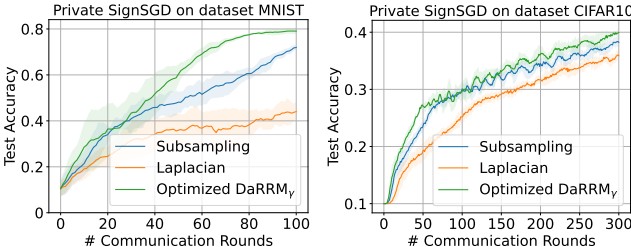

Figure 3: Test accuracy of the aggregated model by private clients trained using $\beta$ Sotchastic SignSGD on MNIST and CIFAR10. Each solid line represents the mean test accuracy across five random runs, and the shaded region represents 1 std. There are $K = 11$ clients, whose sign gradients are $\epsilon = 0.1$-differentially private coordinate-wise at each communication round. We aggregate client sign gradients using different private majority ensembling methods, ensuring that each coordinate of the aggregated sign gradient is $0.3$-differentially private. With the same privacy loss, it is clear that the proposed optimized DaRRM$_\gamma$ achieves the fastest convergence rate and the highest test accuracy within 100 and 300 communication rounds on the two datasets, respectively, compared to the other two baselines.

vate themselves and hence adds redundant noises, as we show in the results. An improved baseline is "Subsampling", which subsamples and aggregates $m$ client' gradient vectors.

**Experiment Setup and Results.** We collaboratively train a model with $K = 11$ clients on the MNIST and CIFAR10 datasets. The training set for each dataset is divided equally among the clients and each client has 5454 local training samples. Each dataset has 10000 test samples. At each communication round, each client sends a sign gradient vector, with each coordinate being $\epsilon = 0.1$-differentially private, to the server and we set the privacy allowance $m = 3$. We perform private distributed Sign-SGD for a total of 100 and 300 communication rounds on MNIST and CIFAR10, respectively[6], and report the test accuracy per communication round. We report the mean test accuracy across five random runs and one std. in Figure 3.

---

[6]Note the total privacy loss of the final model obtained from the server after $T$ communication rounds can be empirically computed using a tight moment accountant method for private majority ensembling from Papernot et al. (2017). The overall privacy loss of the model does scale with the number of parameters $d$. And to further reduce this privacy cost, Jin et al. (2021); Xiang & Su (2023b) proposes to sparsify the gradient vector so that only a subset of the coordinates is sent from each client to the server at each communication round. Note our private aggregation framework DaRRM can also be combined with any sparsification techinque.

### 6.3 APPLICATION 2: PRIVATE SEMI-SUPERVISED KNOWLEDGE TRANSFER

**Semi-supervised Knowledge Transfer.** We apply our DaRRM framework to aggregate labels from private teachers in the application of semi-supervised knowledge transfer. We follow a similar setup as in PATE Papernot et al. (2017; 2018), where $K$ teachers are trained on different sensitive datasets and one uses majority voting to aggregate the teachers' votes at inference time. Each time a label is queried from the teachers, one suffers certain privacy loss. To limit the total privacy loss over all queries, a student model is trained on a public dataset without labels. The student model queries labels of a small portion of the training samples from the teachers and is then trained using semi-supervised learning algorithms on both labeled and unlabeled samples. Different from PATE which aggregates non-private teachers while ensuring the aggregated vote from the teachers is private, we aggregate $(\epsilon, \Delta)$-differentially private teachers and ensure the aggregated vote to be $(m\epsilon, \delta)$-differentially private for $m < K$. The motivation is that the aggregator is not guaranteed to be trustworthy in practice, and so each teacher privatizes their votes before informing the aggregator. This semi-supervised knowledge transfer setting is illustrated in Figure 11 in Appendix F.3, where we highlight the difference between our setting and PATE's setting.

**Baselines.** "GNMax" Papernot et al. (2018), which adds Gaussian noise with parameter $\sigma$ [7] to the number of votes from each class, and report the vote with the noisy maximum count. Again, this approach does not consider each teacher being private and leads to adding redundant noise. "Subsampling", which outputs the majority vote from $m$ subsampled votes. The overall privacy loss of the aggregated vote is reasoned by simple composition. Note in our setting with a small $m$, simple composition indeed indicates less privacy loss than advanced composition.

**Experiment Setup and Results.** We use the `MNIST` dataset from two randomly chosen classes: class 5 and 8, resulting in a total of 11272 training samples and 1866 testing samples. We train $K = 11$ private teachers, each for 10 epochs, using DP-SGD Abadi et al. (2016), with zero-mean Guassian noises having std. $\sigma_{SGD} = 10$ and the gradient norm clipping threshold $C = 1$; this leads to each teacher being $(\epsilon, \Delta) = (0.2825, 10^{-8})$-differentially private after training. The privacy allowance is $m = 3$ so that each aggregated vote from the teachers is $(m\epsilon, m\Delta)$-differentially private.

After training the teachers, we treat the test dataset as the public dataset for training a student model. We query the teachers for $Q$ randomly chosen samples from the test dataset. Papernot et al. (2018) empirically shows querying $Q = 1\%N$, where $N$ is the size of the public dataset, suffices to train a student model with a good performance. Therefore, we pick $Q = \{20, 50, 100\}$. We repeat the selection of $Q$ samples 10 times and report the mean test accuracy with one std. in parentheses in Table 1. Note that the $Q$ queried samples serve as the labeled samples in training the student model. The higher the accuracy of the queried labels, the better the performance of the student model[8].

| # Queries / Majority Ensembling Algorithm | GNMax | Optimized DaRRM$_\gamma$ (Ours) | Subsampling |
|---|---|---|---|
| $Q = 20$ | 0.5950 (0.11) | **0.9150 (0.04)** | 0.8550 (0.07) |
| $Q = 50$ | 0.5480 (0.05) | **0.9120 (0.02)** | 0.8560 (0.06) |
| $Q = 100$ | 0.5970 (0.03) | **0.9130 (0.02)** | 0.8520 (0.04) |

Table 1: Accuracy of $Q$ query samples randomly chosen from the public test datast. The output label of each query sample is aggregated using different private majority ensembling algorithms. With the same output privacy guarantee, our optimized DaRRM$_\gamma$ achieves the highest accuracy on the queried samples. A higher accuracy of the queried samples indicates a higher performance of the student model trained with labels from those queried samples.

## 7 CONCLUSION

In computing a private majority from $K$ private mechanisms, we propose the DaRRM framework with a customizable $\gamma$ function that is provably general. We show a privacy amplification by a factor of 2 in the i.i.d. mechanisms and a pure differential privacy setting. For the general setting, we propose an efficient optimization algorithm that maximizes utility while ensuring privacy guarantees. We hope that this work inspires more research on the intersection of privacy frameworks and optimization.

---

[7]We discuss how to choose this $\sigma$ parameter to ensure the aggregated teachers' vote being $(\epsilon, \delta)$-differentially private and other details about GNMax in Appendix F.3.

[8]We skip the actual training of the student model with semi-supervised learning algorithms here.

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

CONTENTS

## A  RELATED WORK

**Private Composition.** In the blackbox composition setting, one can do no better than the $O(K\epsilon)$ privacy analysis for pure differential privacy Dwork et al. (2014). For approximate differential privacy, previous work has found optimal constants for advanced composition by reducing to the binary case of hypothesis testing with randomized response; and optimal tradeoffs between $\epsilon, \delta$ for black box composition are given in Kairouz et al. (2015), where there could be a modest improvement $20\%$.

Thus, for specific applications, some work has turned to white-box composition analysis for improved utility analysis. Abadi et al. (2016) applied a technique called moment accountant for private SGD to reduce the $\log(1/\delta)$ dependence in the $\epsilon$ term and linear dependence on $k$ in the $\delta$ term. For general private stochastic convex optimization, one can avoid the linear dependence on $k$ in $\epsilon$ by using iterative application of contractive maps Feldman et al. (2018). For the specific case of model ensembles, Papernot et al. (2018) uses student model learning to privately aggregate a ensemble of teacher models trained on disjoint datasets and shows a data-dependent privacy bound that vanishes as the probability of disagreement goes to $0$. Their method provides no utility analysis but they empirically observed less privacy loss when there is greater ensemble agreement.

When $g$ is the maximization function, some previous work shows that an approximately maximum value can be outputted with high probability while incurring $O(\epsilon)$ privacy loss, independently of $K$. They proposed a random stopping mechanism for $m = 1$ that draws samples uniformly at random from $M_i(\mathcal{D})$ at each iteration. In any given iteration, the sampling halts with probability $\gamma$ and the final output is computed based on the samples collected until that time. This leads to a final privacy cost of only $3\epsilon$ for the maximization function $g$, which can be improved to $2\epsilon$ (Papernot & Steinke,

2022). In addition to the aforementioned works, composing top-k and exponential mechanisms also enjoy slightly improved composition analysis via a bounded-range analysis Durfee & Rogers (2019); Dong et al. (2020).

**Bypassing the Global Sensitivity.** To ensure differential privacy, it is usually assumed the query function $g$ has bounded global sensitivity — that is, the output of $g$ does not change much on *any* adjacent input datasets differing in one entry. The noise added to the output is then proportional to the global sensitivity of $g$. If the sensitivity is large, the output utility will thus be terrible due to a large amount of noises added. However, the worst case global sensitivity can be rare in practice, and this observation has inspired a line of works on designing private algorithms with data-dependent sensitivity bound to reduce the amount of noises added.

Instead of using the maximum global sensitivity of $g$ on any dataset, the classical Propose-Test-Release framework of Dwork Dwork & Lei (2009) uses a local sensitivity value for robust queries that is tested privately and if the sensitivity value is too large, the mechanism is halted before the query release. The halting mechanism incurs some failure probability but deals with the worst-case sensitivity situations, while allowing for lower noise injection in most average-case cases.

One popular way to estimate average-case sensitivity is to use the Subsample-and-Aggregate framework by introducing the notion of *perturbation stability*, also known as *local sensitivity* of a function $g$ on a dataset $\mathcal{D}$ Thakurta & Smith (2013); Dwork et al. (2014), which represents the minimum number of entries in $\mathcal{D}$ needs to be changed to change $g(\mathcal{D})$. One related concept is *smooth sensitivity*, a measure of variability of $g$ in the neighborhood of each dataset instance. To apply the framework under *smooth sensitivity*, one needs to privately estimate a function's local sensitivity $L_s$ and adapt noise injection to be order of $O(\frac{L_s}{\epsilon})$, where $L_s$ can often be as small as $O(e^{-n})$, where $n = |\mathcal{D}|$, the total dataset size Nissim et al. (2007). Generally, the private computation of the smooth sensitivity of a blackbox function is nontrivial but is aided by the Subsample and Aggregate approach for certain functions.

These techniques hinge on the observation that a function with higher stability on $\mathcal{D}$ requires less noise to ensure worst case privacy. Such techniques are also applied to answer multiple online functions/queries in model-agnostic learning Bassily et al. (2018). However, we highlight two key differences in our setting with a weaker stability assumption. First, in order to estimate the *perturbation stability* of $g$ on $\mathcal{D}$, one needs to downsample or split $\mathcal{D}$ into multiple blocks Thakurta & Smith (2013); Dwork et al. (2014); Bassily et al. (2018), $\hat{\mathcal{D}}_1, \ldots, \hat{\mathcal{D}}_B$, and estimate the *perturbation stability* based on the mode of $g(\hat{\mathcal{D}}_1), \ldots, g(\hat{\mathcal{D}}_B)$. This essentially reduces the amount of change in the output of $g$ due to a single entry in $\mathcal{D}$, with high probability and replaces the hard-to-estimate *perturbation stability* of $g$ with an easy-to-compute *perturbation stability* of the mode. Such a notion of stability has also been successfully applied, along with the sparse vector technique, for model-agnostic private learning to handle exponentially number of queries to a model Bassily et al. (2018). Note that in these cases, since a private stochastic test is applied, one cannot achieve pure differential privacy Dwork et al. (2014). In practice, e.g. federated learning, however, one does not have direct access to $\mathcal{D}$, and thus it is impractical to draw samples from or to split $\mathcal{D}$. Second, to ensure good utility, one relies on a key assumption, i.e. the *subsampling stability* of $g$, which requires $g(\hat{\mathcal{D}}) = g(\mathcal{D})$ with high probability over the draw of subsamples $\hat{\mathcal{D}}$.

**Learning the Optimal Noise Distribution.** Most of these works only claim improved utility and there is no optimality guarantee. There have been limited works that attempt to derive or learn the best noise distribution. For deep neural networks inference, Mireshghallah et al. (2020) attempts to learn the best noise distribution to maximizing utility subject to an entropy Lagragian, but no formal privacy guarantees were derived. For queries with bounded sensitivity, Geng & Viswanath (2015) demonstrate that the optimal noise distribution is in fact a staircase distribution that approaches the Laplacian distribution as $\epsilon \to 0$.

## B PRELIMINARIES: DIFFERENTIAL PRIVACY

**Definition B.1** (Differential Privacy (DP) Dwork et al. (2014))**.** *A randomized mechanism* $\mathcal{M}$ : $\mathcal{D} \to \mathcal{R}$ *with a domain* $\mathcal{D}$ *and range* $\mathcal{R}$ *satisfies* $(\epsilon, \delta)$-*differential privacy for* $\epsilon, \delta \geq 0$ *if for any two* **adjacent datasets** $\mathcal{D}, \mathcal{D}'$ *and for any subset of outputs* $S \subseteq \mathcal{R}$ *it holds that* $\Pr[\mathcal{M}(\mathcal{D}) \in S] \leq$

$e^\epsilon \Pr[\mathcal{M}(\mathcal{D}') \in S] + \delta$. $\delta = 0$ *is often called pure differential privacy; while* $\delta > 0$ *is often called approximate differential privacy.*

**Theorem B.2** (Simple Composition Dwork et al. (2014)). *Let* $\mathcal{M}_1 : \mathcal{D} \to \mathcal{R}_1$ *be an* $\epsilon_1$-*differentially privacy mechanism and* $\mathcal{M}_2 : \mathcal{D} \to \mathcal{R}_2$ *be an* $\epsilon_2$-*differentially privacy mechanism, then their combination* $\mathcal{M}_{1,2}(x) = (\mathcal{M}_1(x), \mathcal{M}_2(x))$ *is* $(\epsilon_1 + \epsilon_2)$-*differentially private.*

**Theorem B.3** (Advanced Composition Dwork et al. (2014)). *For all* $\epsilon, \delta, \delta' \geq 0$, *the class of* $(\epsilon, \delta)$-*differentially private mechanisms satisfies* $(\epsilon', k\delta + \delta')$-*differential privacy under* $k$-*fold adaptive composition for*

$$\epsilon' = \sqrt{2k\ln(1/\delta')}\epsilon + k\epsilon(e^\epsilon - 1) \tag{4}$$

## C  DETAILS OF SECTION 3: PRIVATE MAJORITY ALGORITHMS

### C.1  RANDOMIZED RESPONSE WITH CONSTANT $\gamma$

Recall the classical Randomized Response (RR) algorithm that provides a binary output algorithm with differential privacy guarantee proceeds as follows: With probability $p_\gamma$, one returns the true output of the algorithm; otherwise, one returns a random answer. In this section, we show the magnitude of the constant probability $p_\gamma$ in RR to use RR to solve Problem 1.1 and to ensure RR is $(m\epsilon, \delta)$-differentially private. We can view the $p_\gamma$ probability in RR as a constant $\gamma : \{0, 1, \ldots, K\} \to [0, 1]$ function such that $\gamma(l) = p_\gamma, \forall l \in [K]$.

**Lemma C.1** (Randomized Response (constant) $\gamma$). *Consider Problem 1.1 with privacy allowance* $m > 0$ *and failure probability* $\delta \geq 0$. *Let* $p_\gamma$ *be the probability of outputting the true majority based on* $K$ *samples in Randomized Response (RR). Let the majority of* $K$ $(\epsilon, \Delta)$ *mechanisms be* $(\tau\epsilon, \lambda)$-*differentially private, reasoned by simple composition or advanced composition for some* $0 < \tau \leq K, 0 \leq \lambda < 1$. *If one sets*

$$p_\gamma = \frac{e^{m\epsilon} - 1 + 2\delta}{\frac{2(e^{\tau\epsilon} - e^{m\epsilon} + 2\lambda)}{e^{\tau\epsilon} + 1} + e^{m\epsilon} - 1} \tag{5}$$

*then RR is* $(m\epsilon, \delta)$-*differentially private.*

*Proof.* For convenience, let $x \in \{0, 1\}$ denote the output majority, and $q_x, q'_x$ denote the probability the aggregated majority from $K$ samples is $x$ on dataset adjacent $\mathcal{D}$ and $\mathcal{D}'$ respectively. Recall each mechanism we aggregate is $(\epsilon, \Delta)$-differentially private. The output of the aggregated majority from $K$ samples is $(\tau\epsilon, \lambda)$-differentially private, for some $\tau \leq K$. When $\Delta = 0$, $\tau = K$ and $\lambda = 0$ can be reasoned through simple composition. When $\Delta > 0$, $\tau \approx \sqrt{K}$ and $\lambda \approx K\Delta$ can be reasoned through advanced composition. And so simultaneously the following four constraints on $q_x, q'_x$ apply:

$$q_x \leq e^{\tau\epsilon}q'_x + \lambda, \quad \text{and} \quad 1 - q'_x \leq e^{\tau\epsilon}(1 - q_x) + \lambda \tag{6}$$

$$q'_x \leq e^{\tau\epsilon}q_x + \lambda, \quad \text{and} \quad 1 - q_x \leq e^{\tau\epsilon}(1 - q'_x) + \lambda \tag{7}$$

To ensure RR is $(m\epsilon, \delta)$-differentially private, one needs $\gamma$ such that for all possible $q_x, q'_x$,

$$\Pr[\text{RR}(\mathcal{D}) = x] \leq e^{m\epsilon} \Pr[\text{RR}(\mathcal{D}') = x] + \delta \tag{8}$$

$$\gamma \cdot q_x + \frac{1}{2}(1 - \gamma) \leq e^{m\epsilon}(\gamma \cdot q'_x + \frac{1}{2}(1 - \gamma)) + \delta \tag{9}$$

$$(q_x - e^{m\epsilon}q'_x + \frac{1}{2}e^{m\epsilon} - \frac{1}{2}) \cdot \gamma \leq \frac{1}{2}e^{m\epsilon} - \frac{1}{2} + \delta \tag{10}$$

To maximize the utility, one wants to maximize $\gamma$ while conforming to the above privacy constraints. Hence, we solve solve the following Linear Programming (LP) problem:

Objective: $$\max_{q_x, q'_x} f(q_x, q'_x) = q_x - e^{m\epsilon}q'_x + \frac{1}{2}e^{m\epsilon} - \frac{1}{2} \tag{11}$$

Subject to: $$0 \leq q_x \leq 1, 0 \leq q'_x \leq 1 \tag{12}$$

$$q_x \leq e^{\tau\epsilon}q'_x + \lambda, 1 - q'_x \leq e^{\tau\epsilon}(1 - q_x) + \lambda \tag{13}$$

$$q'_x \leq e^{\tau \epsilon} q_x + \lambda, 1 - q_x \leq e^{\tau \epsilon}(1 - q'_x) + \lambda \tag{14}$$

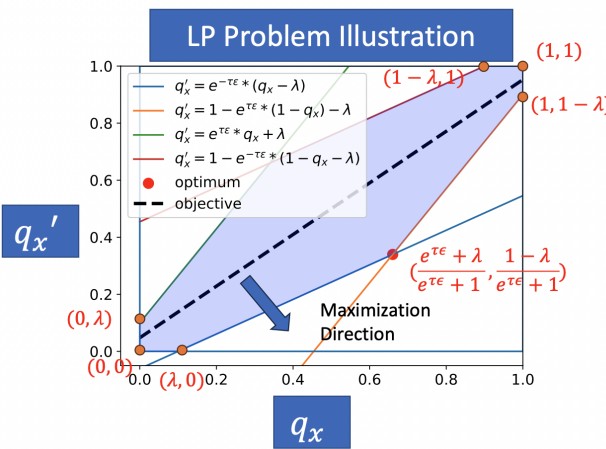

Figure 4: A visualization of the LP problem.

The optimum of any LP problem is at the corners of the feasible region. Here, the feasible region $\mathcal{F}$ is shown in Figure 4. This means $(q_x^*, q_x'^*) = \arg\max_{q_x, q_x'} f(q_x, q_x') \in \{(0,0), (1,1), (0,\lambda), (\lambda, 0), (1-\lambda, 1), (1, 1-\lambda), (\frac{1-\lambda}{e^{\tau\epsilon}+1}, \frac{e^{\tau\epsilon}+\lambda}{e^{\tau\epsilon}+1}), (\frac{e^{\tau\epsilon}+\lambda}{e^{\tau\epsilon}+1}, \frac{1-\lambda}{e^{\tau\epsilon}+1})\}$. The optimum of the above LP problem is at

$$q_x^* = \frac{e^{\tau\epsilon} + \lambda}{e^{\tau\epsilon} + 1}, \quad q_x'^* = \frac{1 - \lambda}{e^{\tau\epsilon} + 1} \tag{15}$$

We need to set $\gamma$ according to the following upper bound to ensure privacy while maximizing $\gamma$ to maximize utility,

$$\gamma \cdot \max_{q_x, q_x'} f(q_x, q_x') \leq \frac{1}{2}(e^{m\epsilon} - 1) + \delta \tag{16}$$

Hence, we want $\gamma$ that

$$\gamma \cdot \left( \frac{e^{\tau\epsilon} + \lambda}{e^{\tau\epsilon} + 1} - e^{m\epsilon} \frac{1 - \lambda}{e^{\tau\epsilon} + 1} + \frac{1}{2}e^{m\epsilon} - \frac{1}{2} \right) = \frac{1}{2}(e^{m\epsilon} - 1) + \delta \tag{17}$$

$$\gamma \cdot \left( \frac{e^{\tau\epsilon} - e^{m\epsilon} + 2\lambda}{e^{\tau\epsilon} + 1} + \frac{1}{2}(e^{m\epsilon} - 1) \right) = \frac{1}{2}(e^{m\epsilon} - 1) + \delta \tag{18}$$

$$\gamma = \frac{e^{m\epsilon} - 1 + 2\delta}{\frac{2(e^{\tau\epsilon} - e^{m\epsilon} + 2\lambda)}{e^{\tau\epsilon} + 1} + e^{m\epsilon} - 1} \tag{19}$$

For small $m, \epsilon, K$, using the approximation $e^y \approx 1 + y$,

$$\gamma \approx \frac{m\epsilon + 2\delta}{\frac{2(\tau\epsilon - m\epsilon + 2\lambda)}{\tau\epsilon + 2} + m\epsilon} = \frac{m + 2\delta/\epsilon}{\frac{2(\tau - m + 2\lambda/\epsilon)}{\tau\epsilon + 2} + m} \tag{20}$$

$\square$

## C.2   PROOF OF LEMMA 3.1: THE SUBSAMPLING $\gamma$ FUNCTION

**Lemma 3.1.** *Consider Problem 1.1, and the sum of observed outcomes of the mechanisms, $l = \sum_{i=1}^{K} S_i \in \{0, 1, \ldots, K\}$. For $m \in \mathbb{Z}^+$, $m \leq K$, if one sets a success probability $\gamma_{Subsampling}$,*

*dependent on the value of $l$, by*

$$\gamma_{Subsampling}(l) = \begin{cases} \gamma_{Subsampling}(K-l) = 1 - 2\sum_{j=\frac{m+1}{2}}^{m} \frac{\binom{l}{j}\binom{K-l}{m-j}}{\binom{K}{m}} & \text{for odd } m \\ \gamma_{Subsampling}(K-l) = 1 - 2\sum_{j=\frac{m}{2}+1}^{m} \frac{\binom{l}{j}\binom{K-l}{m-j}}{\binom{K}{m}} - \frac{\binom{l}{\frac{m}{2}}\binom{K-l}{\frac{m}{2}}}{\binom{K}{m}} & \text{for even } m \end{cases}$$

(21)

*then outputting the majority of $m$ out of $K$ subsampled mechanisms without replacement and DaRRM$_{\gamma_{Subsampling}}$ have the same output distribution.*

*Proof.* Let algorithm SS denote outputting the majority based on $m$ out of $K$ subsampled mechanisms without replacement. Note the output of SS is the same as drawing one sample per mechanism $\mathcal{S} = \{S_i\}_{i=1}^{K}$, where $S_i \sim M_i(\mathcal{D})$, subsample $m$ of the observed samples without replacement and outputs the majority based on the $m$ subsamples. Let $\mathcal{L} = \sum_{i=1}^{K} S_i$ be the sum of observed outcomes from $K$ mechanisms, and conditioned on $\mathcal{L}$, notice the output follows a hypergeometric distribution. Hence, the output probability of SS can be written as

$$\Pr[\text{SS}(\mathcal{D}) = 1] = \sum_{l=0}^{K} \Pr[\text{SS}(\mathcal{D}) = 1 \mid \mathcal{L} = l] \cdot \Pr[\mathcal{L} = l] \tag{22}$$

$$= \sum_{l=0}^{K} \Pr[\sum_{j=1}^{m} S_j \geq \frac{m}{2} \mid \mathcal{L} = l] \cdot \Pr[\mathcal{L} = l] \tag{23}$$

$$= \begin{cases} \sum_{l=0}^{K} (\sum_{j=\frac{m+1}{2}}^{m} \frac{\binom{l}{j}\binom{K-l}{m-j}}{\binom{K}{m}}) \cdot \Pr[\mathcal{L} = l] & \text{if } m \text{ is odd} \\ \sum_{l=0}^{K} (\sum_{j=\frac{m}{2}+1}^{m} \frac{\binom{l}{j}\binom{K-l}{m-j}}{\binom{K}{m}} + \frac{1}{2}\frac{\binom{l}{\frac{m}{2}}\binom{K-l}{\frac{m}{2}}}{\binom{K}{m}}) \cdot \Pr[\mathcal{L} = l] & \text{if } m \text{ is even} \end{cases} \tag{24}$$

Recall $\gamma : \{0, 1, \dots, K\} \to [0, 1]$ is the noise function in DaRRM. The output probability of DaRRM$_\gamma$ is:

$$\Pr[\text{DaRRM}_\gamma(\mathcal{D}) = 1] = \sum_{l=0}^{K} \Pr[\text{DaRRM}_\gamma(\mathcal{D}) = 1 \mid \mathcal{L} = l] \cdot \Pr[\mathcal{L} = l] \tag{25}$$

$$= \sum_{l=0}^{K} (\gamma(l) \cdot \mathbb{I}\{l \geq \frac{K+1}{2}\} + \frac{1}{2}(1 - \gamma(l))) \cdot \Pr[\mathcal{L} = l] \tag{26}$$

To let $\Pr[\text{DaRRM}_\gamma(\mathcal{D}) = 1] = \Pr[\text{SS}(\mathcal{D}) = 1]$, if $m$ is odd, for $l \leq \frac{K-1}{2}$,

$$\frac{1}{2}(1 - \gamma(l)) = \sum_{j=\frac{m+1}{2}}^{m} \frac{\binom{l}{j}\binom{K-l}{m-j}}{\binom{K}{m}} \quad \Rightarrow \gamma(l) = 1 - 2\sum_{j=\frac{m+1}{2}}^{m} \frac{\binom{l}{j}\binom{K-l}{m-j}}{\binom{K}{m}} \tag{27}$$

and for $l \geq \frac{K+1}{2}$,

$$\frac{1}{2} + \frac{1}{2}\gamma(l) = \sum_{j=\frac{m+1}{2}}^{m} \frac{\binom{l}{j}\binom{K-l}{m-j}}{\binom{K}{m}} \quad \Rightarrow \gamma(l) = 2\sum_{j=\frac{m+1}{2}}^{m} \frac{\binom{l}{j}\binom{K-l}{m-j}}{\binom{K}{m}} - 1 \tag{28}$$

Similarly, if $m$ is even, for $l \leq \frac{K-1}{2}$,

$$\frac{1}{2}(1 - \gamma(l)) = \sum_{j=\frac{m}{2}+1}^{m} \frac{\binom{l}{j}\binom{K-l}{m-j}}{\binom{K}{m}} + \frac{1}{2}\frac{\binom{l}{\frac{m}{2}}\binom{K-l}{\frac{m}{2}}}{\binom{K}{m}} \quad \Rightarrow \gamma(l) = 1 - 2\sum_{j=\frac{m}{2}+1}^{m} \frac{\binom{l}{j}\binom{K-l}{m-j}}{\binom{K}{m}} - \frac{\binom{l}{\frac{m}{2}}\binom{K-l}{\frac{m}{2}}}{\binom{K}{m}}$$

(29)

and for $l \geq \frac{K+1}{2}$,

$$\frac{1}{2} + \frac{1}{2}\gamma(l) = \sum_{j=\frac{m}{2}+1}^{m} \frac{\binom{l}{j}\binom{K-l}{m-j}}{\binom{K}{m}} + \frac{1}{2}\frac{\binom{l}{\frac{m}{2}}\binom{K-l}{\frac{m}{2}}}{\binom{K}{m}} \quad \Rightarrow \gamma(l) = 2\sum_{j=\frac{m}{2}+1}^{m} \frac{\binom{l}{j}\binom{K-l}{m-j}}{\binom{K}{m}} + \frac{\binom{l}{\frac{m}{2}}\binom{K-l}{\frac{m}{2}}}{\binom{K}{m}} - 1$$

(30)

Note that this $\gamma(l)$ is symmetric around $\frac{K}{2}$, since for $l \leq \frac{K-1}{2}$ (and so $K - l \geq \frac{K+1}{2}$), if $m$ is odd,

$$\gamma(K-l) = 2\sum_{j=\frac{m+1}{2}}^{m} \frac{\binom{K-l}{j}\binom{l}{m-j}}{\binom{K}{m}} - 1 = 2\Big(1 - \sum_{j=1}^{\frac{m-1}{2}} \frac{\binom{K-l}{j}\binom{l}{m-j}}{\binom{K}{m}}\Big) - 1 \tag{31}$$

$$= 1 - 2\sum_{j=1}^{\frac{m-1}{2}} \frac{\binom{K-l}{j}\binom{l}{m-j}}{\binom{K}{m}} = 1 - 2\sum_{j=\frac{m+1}{2}}^{m} \frac{\binom{l}{j}\binom{K-l}{m-j}}{\binom{K}{m}} \tag{32}$$

$$= \gamma(l) \tag{33}$$

Similarly, if $m$ is even,

$$\gamma(K-l) = 2\sum_{j=\frac{m}{2}+1}^{m} \frac{\binom{K-l}{j}\binom{l}{m-j}}{\binom{K}{m}} + \frac{\binom{l}{\frac{m}{2}}\binom{K-l}{\frac{m}{2}}}{\binom{K}{m}} - 1 = 2\Big(1 - \sum_{j=1}^{\frac{m}{2}-1} \frac{\binom{K-l}{j}\binom{l}{m-j}}{\binom{K}{m}} - \frac{1}{2}\frac{\binom{l}{\frac{m}{2}}\binom{K-l}{\frac{m}{2}}}{\binom{K}{m}}\Big) - 1 \tag{34}$$

$$= 1 - 2\sum_{j=1}^{\frac{m}{2}-1} \frac{\binom{K-l}{j}\binom{l}{m-j}}{\binom{K}{m}} - \frac{\binom{l}{\frac{m}{2}}\binom{K-l}{\frac{m}{2}}}{\binom{K}{m}} = 1 - 2\sum_{j=\frac{m}{2}+1}^{m} \frac{\binom{l}{j}\binom{K-l}{m-j}}{\binom{K}{m}} - \frac{\binom{l}{\frac{m}{2}}\binom{K-l}{\frac{m}{2}}}{\binom{K}{m}} \tag{35}$$

$$= \gamma(l) \tag{36}$$

Therefore, setting $\gamma$ as in Eq. 27 if $m$ is odd, and as in Eq. 29 if $m$ is even makes DaRRM$_\gamma$ have the same output distribution as SS. We hence call this $\gamma$ function $\gamma_{Subsampling}$. $\qquad\square$

## C.3 Proof of Lemma 3.2: Generality of DaRRM

**Lemma 3.2** (Generality of DaRRM). *Let $\mathcal{A}$ be any randomized algorithm to compute the majority function $g$ on $\mathcal{S}$ such that for all $\mathcal{S}$, $\Pr[\mathcal{A}(\mathcal{S}) = g(\mathcal{S})] \geq 1/2$ (i.e. $\mathcal{A}$ is at least as good as a random guess). Then, there exists a a general $\gamma \colon \{0,1\}^{K+1} \to [0,1]$ such that if one sets $p_\gamma$ by $\gamma(\mathcal{S})$ in DaRRM, the output distribution of DaRRM$_\gamma$ is the same as the output distribution of $\mathcal{A}$.*

*Proof.* For some $\mathcal{D}$ and conditioned on $\mathcal{S}$, we see that by definition $\Pr[\text{DaRRM}_\gamma(\mathcal{S}) = g(\mathcal{S})] = \gamma(\mathcal{S}) + (1/2)(1 - \gamma(\mathcal{S}))$. We want to set $\gamma$ such that $\Pr[\text{DaRRM}_\gamma(\mathcal{S}) = g(\mathcal{S})] = \Pr[\mathcal{A}(\mathcal{S}) = g(\mathcal{S})]$. Therefore, we set $\gamma(\mathcal{S}) = 2\Pr[\mathcal{A}(\mathcal{S}) = g(\mathcal{S})] - 1$.

Lastly, we need to justify that $\gamma \in [0,1]$. Clearly, $\gamma(\mathcal{S}) \leq 2 - 1 \leq 1$ since $\Pr[\mathcal{A}(\mathcal{S}) = g(\mathcal{S})] \leq 1$. Note that the non-negativity follows from assumption. $\qquad\square$

## C.4 Proof of Lemma 3.3: $\gamma$ Privacy Condition

**Lemma 3.3** ($\gamma$ privacy condition and privacy cost objective). *Consider using DaRRM to solve Problem 1.1. Let $\alpha_l = \Pr[\mathcal{L}(\mathcal{D}) = l]$ and $\alpha_l' = \Pr[\mathcal{L}(\mathcal{D}') = l]$, for $l$ in support $\{0, \ldots, K\}$ and adjacent datasets $\mathcal{D}, \mathcal{D}'$. For $\gamma \colon \{0, 1, \ldots, K\} \to [0,1]$ such that $\gamma(l) = \gamma(K-l), \forall l$, DaRRM$_\gamma$ is $(m\epsilon, \delta)$-differentially private if and only if for all $\alpha_l, \alpha_l'$,*

$$f(p_1, \ldots, p_K, p_1', \ldots, p_K'; \gamma) := \sum_{l=0}^{\frac{K-1}{2}} (e^{m\epsilon}\alpha_l' - \alpha_l) \cdot \gamma(l) + \sum_{l=\frac{K+1}{2}}^{K} (\alpha_l - e^{m\epsilon}\alpha_l') \cdot \gamma(l) \leq e^{m\epsilon} - 1 + 2\delta \tag{37}$$

*We call $f$ the privacy cost objective.*

*Proof.* By the definition of differential privacy,

DaRRM$_\gamma$ is $(m\epsilon, \delta)$-differentially private

$$\iff \Pr[\text{DaRRM}_\gamma(\mathcal{D}) = 1] \leq e^{m\epsilon} \Pr[\text{DaRRM}_\gamma(\mathcal{D}') = 1] + \delta, \tag{38}$$

$$\text{and } \Pr[\text{DaRRM}_\gamma(\mathcal{D}) = 0] \leq e^{m\epsilon} \Pr[\text{DaRRM}_\gamma(\mathcal{D}') = 0] + \delta, \quad \forall \text{ adjacent } \mathcal{D}, \mathcal{D}' \tag{39}$$

Consider random variables $\mathcal{L}(\mathcal{D}) = \sum_{i=1}^{K} S(\mathcal{D})$ and $\mathcal{L}(\mathcal{D}') = \sum_{i=1}^{K} S(\mathcal{D}')$, based on which one sets $\gamma$. Note, when the output is 1,

$$\Pr[\text{DaRRM}_\gamma(\mathcal{D}) = 1] \leq e^{m\epsilon} \Pr[\text{DaRRM}_\gamma(\mathcal{D}') = 1] + \delta \tag{40}$$

$$\iff \sum_{l=0}^{K} \Pr[\text{DaRRM}_\gamma(\mathcal{D}) = 1 \mid \mathcal{L}(\mathcal{D}) = 1] \cdot \Pr[\mathcal{L}(\mathcal{D}) = l] \tag{41}$$

$$\leq e^{m\epsilon} \Big( \sum_{l=0}^{K} \Pr[\text{DaRRM}_\gamma(\mathcal{D}') = 1 \mid \mathcal{L}(\mathcal{D}') = 1] \cdot \Pr[\mathcal{L}(\mathcal{D}') = l] \Big) + \delta$$

$$\iff \sum_{l=0}^{K} \Big( \gamma(l) \cdot \mathbb{I}\{l \geq \frac{K}{2}\} + \frac{1}{2}(1 - \gamma(l)) \Big) \cdot \Pr[\mathcal{L}(\mathcal{D}) = l] \tag{42}$$

$$\leq e^{m\epsilon} \Big( \sum_{l=0}^{K} \Big( \gamma(l) \cdot \mathbb{I}\{l \geq \frac{K}{2}\} + \frac{1}{2}(1 - \gamma(l))\} \Big) \cdot \Pr[\mathcal{L}(\mathcal{D}') = l] \Big) + \delta$$

$$\iff \sum_{l=\frac{K+1}{2}}^{K} \Big( \gamma(l) + \frac{1}{2}(1 - \gamma(l)) \Big) \cdot \Pr[\mathcal{L}(\mathcal{D}) = l] + \sum_{l=0}^{\frac{K-1}{2}} \frac{1}{2}(1 - \gamma(l)) \cdot \Pr[\mathcal{L}(\mathcal{D}) = l] \tag{43}$$

$$\leq e^{m\epsilon} \Big( \sum_{l=\frac{K+1}{2}}^{K} \Big( \gamma(l) + \frac{1}{2}(1 - \gamma(l)) \Big) \cdot \Pr[\mathcal{L}(\mathcal{D}) = l] \Big) + e^{m\epsilon} \Big( \sum_{l=0}^{\frac{K-1}{2}} \frac{1}{2}(1 - \gamma(l)) \cdot \Pr[\mathcal{L}(\mathcal{D}') = l] \Big) + \delta$$

$$\iff \sum_{l=\frac{K+1}{2}}^{K} \frac{1}{2}\gamma(l)\alpha_l - \sum_{l=0}^{\frac{K-1}{2}} \frac{1}{2}\gamma(l)\alpha_l + \frac{1}{2} \tag{44}$$

$$\leq e^{m\epsilon} \sum_{l=\frac{K+1}{2}}^{K} \frac{1}{2}\gamma(l)\alpha_l' - e^{m\epsilon} \sum_{l=0}^{\frac{K-1}{2}} \frac{1}{2}\gamma(l)\alpha_l' + \frac{1}{2}e^{m\epsilon} + \delta$$

$$\iff \sum_{l=\frac{K+1}{2}}^{K} (\alpha_l - e^{m\epsilon}\alpha_l')\gamma(l) - \sum_{l=0}^{\frac{K-1}{2}} (\alpha_l - e^{m\epsilon}\alpha_l')\gamma(l) \leq e^{m\epsilon} - 1 + 2\delta \tag{45}$$

where $\alpha_l = \Pr[\mathcal{L}(\mathcal{D}) = l]$ and $\alpha_l' = \Pr[\mathcal{L}(\mathcal{D}') = l]$, $\forall l \in \{0, 1, \ldots, K\}$.

Similarly, when the output is 0,

$$\Pr[\text{DaRRM}_\gamma(\mathcal{D}) = 0] \leq e^{m\epsilon} \Pr[\text{DaRRM}_\gamma(\mathcal{D}') = 0] + \delta \tag{46}$$

$$\iff \sum_{l=0}^{K} \Pr[\text{DaRRM}_\gamma(\mathcal{D}) = 0 \mid \mathcal{L}(\mathcal{D}) = 0] \cdot \Pr[\mathcal{L}(\mathcal{D}) = 0] \tag{47}$$

$$\leq e^{m\epsilon} \Big( \sum_{l=0}^{K} \Pr[\text{DaRRM}_\gamma(\mathcal{D}') = 0 \mid \mathcal{L}(\mathcal{D}') = 0] \cdot \Pr[\mathcal{L}(\mathcal{D}') = 0] \Big) + \delta$$

$$\iff \sum_{l=0}^{K} \Big( \gamma(l) \cdot \mathbb{I}\{l < \frac{K}{2}\} + \frac{1}{2}(1 - \gamma(l)) \Big) \cdot \Pr[\mathcal{L}(\mathcal{D}) = l] \tag{48}$$

$$\leq e^{m\epsilon} \Big( \sum_{l=0}^{K} \gamma(l) \cdot \mathbb{I}\{l < \frac{K}{2}\} + \frac{1}{2}(1 - \gamma(l)) \Big) \cdot \Pr[\mathcal{L}(\mathcal{D}') = l] + \delta$$

$$\iff \sum_{l=0}^{\frac{K-1}{2}} \left(\gamma(l) + \frac{1}{2}(1 - \gamma(l))\right) \cdot \Pr[\mathcal{L}(\mathcal{D}) = l] + \sum_{l=\frac{K+1}{2}}^{K} \frac{1}{2}(1 - \gamma(l)) \cdot \Pr[\mathcal{L}(\mathcal{D}) = l] \tag{49}$$

$$\leq e^{m\epsilon} \Big( \sum_{l=0}^{\frac{K-1}{2}} \left(\gamma(l) + \frac{1}{2}(1 - \gamma(l))\right) \cdot \Pr[\mathcal{L}(\mathcal{D}') = l] + \sum_{l=\frac{K+1}{2}}^{K} \frac{1}{2}(1 - \gamma(l)) \cdot \Pr[\mathcal{L}(\mathcal{D}') = l] \Big) + \delta$$

$$\iff \sum_{l=0}^{\frac{K-1}{2}} \frac{1}{2}\gamma(l)\alpha_l - \sum_{l=\frac{K+1}{2}}^{K} \frac{1}{2}\gamma(l)\alpha_l + \frac{1}{2} \tag{50}$$

$$\leq e^{m\epsilon} \sum_{l=0}^{\frac{K-1}{2}} \frac{1}{2}\gamma(l)\alpha_l' - e^{m\epsilon} \sum_{l=\frac{K+1}{2}}^{K} \frac{1}{2}\gamma(l)\alpha_l' + \frac{1}{2}e^{m\epsilon} + \delta$$

$$\iff \sum_{l=0}^{\frac{K-1}{2}} (\alpha_l - e^{m\epsilon}\alpha_l')\gamma(l) - \sum_{l=\frac{K+1}{2}}^{K} (\alpha_l - e^{m\epsilon}\alpha_l')\gamma(l) \leq e^{m\epsilon} - 1 + 2\delta \tag{51}$$

Therefore,

$$\text{DaRRM}_\gamma \text{ is } (m\epsilon, \delta)\text{-differentially private}$$

$$\iff \sum_{l=\frac{K+1}{2}}^{K} (\alpha_l - e^{m\epsilon}\alpha_l')\gamma(l) - \sum_{l=0}^{\frac{K-1}{2}} (\alpha_l - e^{m\epsilon}\alpha_l')\gamma(l) \leq e^{m\epsilon} - 1 + 2\delta \tag{52}$$

$$\text{and } \sum_{l=0}^{\frac{K-1}{2}} (\alpha_l - e^{m\epsilon}\alpha_l')\gamma(l) - \sum_{l=\frac{K+1}{2}}^{K} (\alpha_l - e^{m\epsilon}\alpha_l')\gamma(l) \leq e^{m\epsilon} - 1 + 2\delta \tag{53}$$

where $\alpha_l = \Pr[\mathcal{L}(\mathcal{D}) = l]$ and $\alpha_l' = \Pr[\mathcal{L}(\mathcal{D}') = l]$, $\forall l \in \{0, 1, \ldots, K\}$ and $\mathcal{D}, \mathcal{D}'$ are any adjacent datasets.

If $\gamma$ is symmetric around $\frac{K}{2}$, i.e. $\gamma(l) = \gamma(K - l)$, we show as follows satisfying either one of Eq. 52 or Eq. 53 implies satisfying the other one. The intuition is that there is nothing special about outputting 0 or 1.

$$\sum_{l=\frac{K+1}{2}}^{K} (\alpha_l - e^{m\epsilon}\alpha_l')\gamma(l) - \sum_{l=0}^{\frac{K-1}{2}} (\alpha_l - e^{m\epsilon}\alpha_l')\gamma(l) \leq e^{m\epsilon} - 1 + 2\delta \tag{54}$$

$$\iff \sum_{l=0}^{\frac{K-1}{2}} (\alpha_{K-l} - e^{m\epsilon}\alpha_{K-l}') \cdot \gamma(K - l) - \sum_{l=\frac{K-1}{2}}^{K} (\alpha_{K-l} - e^{m\epsilon}\alpha_{K-l}') \cdot \gamma(K - l) \leq e^{m\epsilon} - 1 + 2\delta$$
$$\tag{55}$$

$$\iff \sum_{l=0}^{\frac{K-1}{2}} (\alpha_{K-l} - e^{m\epsilon}\alpha_{K-l}') \cdot \gamma(l) - \sum_{l=\frac{K-1}{2}}^{K} (\alpha_{K-l} - e^{m\epsilon}\alpha_{K-l}') \cdot \gamma(l) \leq e^{m\epsilon} - 1 + 2\delta \tag{56}$$

Since $\gamma(l) = \gamma(K - l)$

$$\iff \sum_{l=0}^{\frac{K-1}{2}} (\alpha_l - e^{m\epsilon}\alpha_l') \cdot \gamma(l) - \sum_{l=\frac{K-1}{2}}^{K} (\alpha_l - e^{m\epsilon}\alpha_l') \cdot \gamma(l) \leq e^{m\epsilon} - 1 + 2\delta \tag{57}$$

Since the above holds for all possible $\alpha_l, \alpha_l'$, one can consider another distribution such that the pmf of $\mathcal{L}(\mathcal{D})$ is $\beta_l = \alpha_{K-l}$ and the pmf of $\mathcal{L}(\mathcal{D}')$ is $\beta_l' = \alpha_{K-l}'$. Then, we rename $\beta_l$ as $\alpha_l$ and $\beta_l'$ as $\alpha_l'$.

The above implies Eq. 52 and Eq. 53 are equivalent. Therefore,

$$\text{DaRRM}_\gamma \text{ is } (m\epsilon, \delta)\text{-differentially private}$$

$$\iff \sum_{l=\frac{K+1}{2}}^{K} (\alpha_l - e^{m\epsilon}\alpha_l')\gamma(l) - \sum_{l=0}^{\frac{K-1}{2}} (\alpha_l - e^{m\epsilon}\alpha_l')\gamma(l) \leq e^{m\epsilon} - 1 + 2\delta \tag{58}$$

$\square$

# D    PROVABLE PRIVACY AMPLIFICATION IN I.I.D. SETTING UNDER PURE DP

Recall in the i.i.d. mechanisms setting, $\Pr[M_i(\mathcal{D}) = 1] = p$ and $\Pr[M_i(\mathcal{D}') = 1] = p'$, for all mechanisms $M_i$. Under pure differential privacy setting, each mechanisms $M_i$ is $\epsilon$-differentially private, and we want the aggregated majority voting output by DaRRM$_\gamma$ with a certain choice of $\gamma$ to be $m\epsilon$-differentially private for $m < K$.

For analysis, we restrict our search for a $\gamma$ function with good utility to the class with a mild monotonicity assumption: $\gamma(l) \geq \gamma(l+1), \forall l \leq \frac{K-1}{2}$ and $\gamma(l) \leq \gamma(l+1), \forall l \geq \frac{K+1}{2}$. This matches our intuition that as $\mathcal{L}$, i.e., the number of mechanisms outputting 1, approaches 0 or $K$, there is a clearer majority and so not much noise is needed to ensure privacy, which implies a larger value of $\gamma$.

**Worst case probabilities.** We call $(p^*, p'^*) = \arg\max_{p,p'} f(p^*, p'^*; \gamma)$ the worst case probabilities since they incur the largest privacy loss, where $f$ is the simplified privacy cost objective defined in Eq. 37 with $\delta = \Delta = 0$. If we can show $f(p^*, p'^*; \gamma) \leq e^\epsilon - 1$ for some $\gamma$, then DaRRM$_\gamma$ is is $m\epsilon$-differentially private by Lemma 3.3. To find the worst case probabilities, first note $(p^*, p'^*)$ are close to each other and lie in a feasible region $\mathcal{F}$, due to each mechanism being $\epsilon$-differentially private in our setting. The feasible region is illustrated in Figure 5, and the four boundaries of which, i.e. (a) $p' \leq e^\epsilon p$ (b) $p \leq e^\epsilon p'$ (c) $1 - p' \leq e^\epsilon(1 - p)$, and (d) $1 - p \leq e^\epsilon(1 - p')$, are derived from the definition of differential privacy.

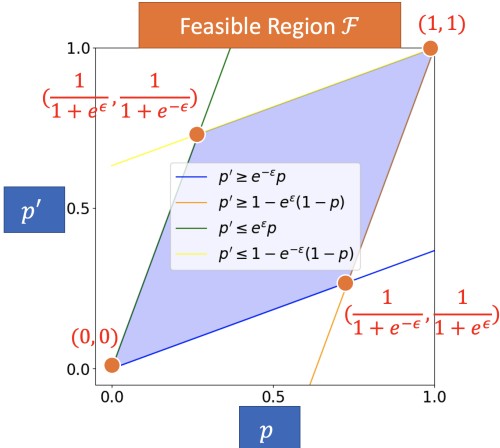

## D.1    CHARACTERIZING WORST CASE PROBABILITIES

Figure 5: The feasible region $\mathcal{F}$ is plotted as the blue area. The four boundaries are implied by $p, p'$ satisfying $\epsilon$-differential privacy.

We first show a key lemma, later used in the proof of our main privacy amplification result, that allows us to further refine the search region for $(p^*, p'^*)$ under $\gamma: \{0, 1, \ldots, K\} \to [0, 1]$ functions that are symmetric around $\frac{K}{2}$ and that satisfy the above mild monotonicity assumption. We call such $\gamma$ functions *well-behaved*.

**Lemma D.1** (Characteristics of worst case probabilities). *Consider well-behaved $\gamma$ functions such that $\gamma(\frac{K-1}{2}) > 0$ and $\gamma(\frac{K+1}{2}) > 0$, the worst case probabilities $(p^*, p'^*) = \arg\max_{p,p'} f(p, p'; \gamma)$ must satisfy exactly one of the following:*

$$p^* = e^\epsilon p'^*, \qquad \forall p \in [0, \frac{1}{e^{-\epsilon} + 1}], p' \in [0, \frac{1}{1 + e^\epsilon}] \qquad (59)$$

$$1 - p'^* = e^\epsilon(1 - p^*), \qquad \forall p \in [\frac{1}{1 + e^{-\epsilon}}, 1], p' \in [\frac{1}{1 + e^\epsilon}, 1] \qquad (60)$$

*See the blue line and the orange line in Figure 5, respectively.*

To show the above Lemma D.1, we show Lemma D.2 and Lemma D.3 as follows, each of which gives partial characteristics of the worst case probabilities. Lemma D.1 directly follows by combining the two lemmas.

**Lemma D.2.** *Consider a $\gamma: \{0, 1, \ldots, K\} \to [0, 1]$ function that is symmetric around $\frac{K}{2}$. If $\gamma$ further satisfies: 1) $\gamma(l+1) \leq \gamma(l), \forall l \leq \frac{K}{2}$, 2) $\gamma(l+1) \geq \gamma(l), \forall l \geq \frac{K}{2}$, and 3) $\gamma(\frac{K-1}{2}) > 0, \gamma(\frac{K+1}{2}) > 0$, then the worst case probabilities $(p^*, p'^*) = \arg\max_{p,p'} f(p, p'; \gamma)$ must satisfy one of the following four equalities:*

$$p'^* = e^\epsilon p^*, \qquad \forall p \in [0, \frac{1}{1 + e^\epsilon}], p' \in [0, \frac{1}{1 + e^{-\epsilon}}] \qquad (61)$$

$$p^* = e^\epsilon p'^*, \qquad \forall p \in [0, \frac{1}{e^{-\epsilon} + 1}], p' \in [0, \frac{1}{1 + e^\epsilon}] \qquad (62)$$

$$1 - p^* = e^\epsilon(1 - p'^*), \qquad \forall p \in [\frac{1}{1 + e^\epsilon}, 1], p' \in [\frac{1}{1 + e^{-\epsilon}}, 1] \qquad (63)$$

$$1 - p'^* = e^\epsilon(1 - p^*), \qquad \forall p \in [\frac{1}{1+e^{-\epsilon}}, 1], p' \in [\frac{1}{1+e^\epsilon}, 1] \tag{64}$$

*Proof of Lemma D.2.* Consider the privacy cost objective $f(p, p'; \gamma)$ as in Lemma 3.3, when the mechanisms are i.i.d. The gradients w.r.t. $p$ and $p'$ are

$$\nabla_p f(p, p'; \gamma) = \sum_{l=0}^{\frac{K-1}{2}} -\binom{K}{l}\gamma(l) \cdot (lp^{l-1}(1-p)^{K-l} - p^l(K-l)(1-p)^{K-l-1}) \tag{65}$$

$$+ \sum_{l=\frac{K+1}{2}}^{K} \binom{K}{l}\gamma(l) \cdot (lp^{l-1}(1-p)^{K-l} - p^l(K-l)(1-p)^{K-l-1})$$

and

$$\nabla_{p'} f(p, p'; \gamma) = \sum_{l=0}^{\frac{K-1}{2}} e^{m\epsilon} \binom{K}{l}\gamma(l) \cdot (lp'^{l-1}(1-p')^{K-l} - p'^l(K-l)(1-p')^{K-l-1}) \tag{66}$$

$$+ \sum_{l=\frac{K+1}{2}}^{K} -e^{m\epsilon} \binom{K}{l}\gamma(l) \cdot (lp'^{l-1}(1-p')^{K-l} - p'^l(K-l)(1-p')^{K-l-1})$$

We show $\forall p \in (0, 1)$, $\nabla_p f(p, p'; \gamma) > 0$ and $\nabla_{p'} f(p, p'; \gamma) < 0$. This implies there is no local maximum inside $\mathcal{F}$, and so $p^*, p'^* = \arg\max_{p,p'} f(p, p'; \gamma)$ must be on one of the four boundaries of $\mathcal{F}$.

To show $\nabla_p f(p, p'; \gamma) > 0$ for $p \in (0, 1)$, we first show for $p \in (0, 1)$

$$\sum_{l=0}^{\frac{K-1}{2}} \gamma(l)\binom{K}{l} \cdot (p^l(K-l)(1-p)^{K-l-1} - lp^{l-1}(1-p)^{K-l}) > 0 \tag{67}$$

$$\Leftrightarrow \sum_{l=0}^{\frac{K-1}{2}} \gamma(l)\binom{K}{l} \cdot p^l(K-l)(1-p)^{K-l-1} > \sum_{l=0}^{\frac{K-1}{2}} \gamma(l)\binom{K}{l} \cdot lp^{l-1}(1-p)^{K-l} \tag{68}$$

$$\Leftrightarrow \sum_{l=0}^{\frac{K-1}{2}} \gamma(l)\binom{K-1}{l}\frac{K}{K-l} \cdot p^l(K-l)(1-p)^{K-l-1} \tag{69}$$

$$> \sum_{l=1}^{\frac{K-1}{2}} \gamma(l)\binom{K-1}{l-1}\frac{K}{l} \cdot lp^{l-1}(1-p)^{K-l}$$

$$\Leftrightarrow K\sum_{l=0}^{\frac{K-1}{2}} \gamma(l)\binom{K-1}{l}p^l(1-p)^{K-l-1} > K\sum_{l=1}^{\frac{K-1}{2}} \gamma(l)\binom{K-1}{l-1}p^{l-1}(1-p)^{K-l} \tag{70}$$

$$\Leftrightarrow \sum_{l=0}^{\frac{K-1}{2}} \gamma(l)\binom{K-1}{l}p^l(1-p)^{K-l-1} > \sum_{l=0}^{\frac{K-1}{2}-1} \gamma(l+1)\binom{K-1}{l}p^l(1-p)^{K-l-1} \tag{71}$$

Note that for $l \leq \frac{K-1}{2}$, $\gamma(l) \geq \gamma(l+1)$. Since $p \in (0, 1)$, this implies for $l \in \{0, \dots, \frac{K-1}{2} - 1\}$,

$$\gamma(l)\binom{K-1}{l}p^l(1-p)^{K-l-1} \geq \gamma(l+1)\binom{K-1}{l}p^l(1-p)^{K-l-1} \tag{72}$$

Furthermore, note the L.H.S. of Eq. 71 has one additional term $\gamma(\frac{K-1}{2})\binom{K-1}{\frac{K-1}{2}}p^{\frac{K-1}{2}}(1-p)^{\frac{K-1}{2}}$. Since $\gamma(\frac{K-1}{2}) > 0$ and $p \in (0, 1)$,

$$\gamma(\frac{K-1}{2})\binom{K-1}{\frac{K-1}{2}}p^{\frac{K-1}{2}}(1-p)^{\frac{K-1}{2}} > 0 \tag{73}$$

Therefore, combining Eq. 72 and Eq. 73, we conclude Eq. 71 holds.

Next, we show for $p \in (0, 1)$,

$$\sum_{l=\frac{K+1}{2}}^{K} \binom{K}{l} \gamma(l) \cdot (lp^{l-1}(1-p)^{K-l} - p^l(K-l)(1-p)^{K-l-1}) > 0 \tag{74}$$

$$\Leftrightarrow \sum_{l=\frac{K+1}{2}}^{K} \binom{K}{l} \gamma(l) \cdot lp^{l-1}(1-p)^{K-l} > \sum_{l=\frac{K+1}{2}}^{K} \binom{K}{l} p^l(K-l)(1-p)^{K-l-1} \tag{75}$$

$$\Leftrightarrow \sum_{l=\frac{K+1}{2}}^{K} \gamma(l)\binom{K-1}{l-1}\frac{K}{l} \cdot lp^{l-1}(1-p)^{K-l} \tag{76}$$

$$> \sum_{l=\frac{K+1}{2}}^{K-1} \gamma(l)\binom{K-1}{l}\frac{K}{K-l} \cdot p^l(K-l)(1-p)^{K-l-1}$$

$$\Leftrightarrow K \sum_{l=\frac{K+1}{2}}^{K} \gamma(l)\binom{K-1}{l-1} \cdot p^{l-1}(1-p)^{K-l} \tag{77}$$

$$> K \sum_{l=\frac{K+1}{2}}^{K-1} \gamma(l)\binom{K-1}{l} \cdot p^l(1-p)^{K-l-1}$$

$$\Leftrightarrow \sum_{l=\frac{K+1}{2}}^{K} \gamma(l)\binom{K-1}{l-1} \cdot p^{l-1}(1-p)^{K-l} > \sum_{l=\frac{K+1}{2}+1}^{K} \gamma(l-1)\binom{K-1}{l-1} \cdot p^{l-1}(1-p)^{K-l} \tag{78}$$

Note that for $l \geq \frac{K+1}{2}+1$, $\gamma(l) \geq \gamma(l-1)$. Since $p \in (0, 1)$, this implies for $l \in \{\frac{K+1}{2}+1, \dots, K\}$,

$$\gamma(l)\binom{K-1}{l-1}p^{l-1}(1-p)^{K-l} \geq \gamma(l-1)\binom{K-1}{l-1}p^{l-1}(1-p)^{K-l} \tag{79}$$

Furthermore, note the L.H.S. of Eq. 78 has one additional term $\gamma(\frac{K+1}{2})\binom{K-1}{\frac{K-1}{2}}p^{\frac{K-1}{2}}(1-p)^{\frac{K-1}{2}}$. Since $\gamma(\frac{K+1}{2}) > 0$ and $p \in (0, 1)$,

$$\gamma(\frac{K+1}{2})\binom{K-1}{\frac{K-1}{2}}p^{\frac{K-1}{2}}(1-p)^{\frac{K-1}{2}} > 0 \tag{80}$$

Therefore, combining Eq. 79 and Eq. 80, we conclude Eq. 78 holds. Hence, combining Eq. 67 and Eq. 74, we have for $p \in (0, 1)$, if $\gamma$ satisfies the three conditions,

$$\nabla_p f(p, p'; \gamma) > 0 \tag{81}$$

Similarly, one can show for $p \in (0, 1)$, if $\gamma$ satisfies the three conditions,

$$\nabla_{p'} f(p, p'; \gamma) < 0 \tag{82}$$

This implies there is no local minima or local maxima inside the feasible region $\mathcal{F}$. Hence, the worst case probability $(p^*, p'^*) = \arg\max_{p,p'} f(p, p'; \gamma)$ is on one of the four boundaries of $\mathcal{F}$, that is, $(p^*, p'^*)$ satisfy exactly one of the following:

$$p'^* = e^\epsilon p^*, \qquad \forall p \in [0, \frac{1}{1+e^\epsilon}], p' \in [0, \frac{1}{1+e^{-\epsilon}}]$$

$$p^* = e^\epsilon p'^*, \qquad \forall p \in [0, \frac{1}{e^{-\epsilon}+1}], p' \in [0, \frac{1}{1+e^\epsilon}]$$

$$1 - p^* = e^\epsilon(1 - p'^*), \qquad \forall p \in [\frac{1}{1+e^\epsilon}, 1], p' \in [\frac{1}{1+e^{-\epsilon}}, 1]$$

$$1 - p'^* = e^\epsilon(1 - p^*), \qquad \forall p \in [\frac{1}{1+e^{-\epsilon}}, 1], p' \in [\frac{1}{1+e^\epsilon}, 1]$$

$\square$

**Lemma D.3.** *Consider a* $\gamma: \{0, 1, \ldots, K\} \to [0, 1]$ *function that is symmetric around* $\frac{K}{2}$. *If* $\gamma$ *satisfies:* $\gamma(l) \geq \gamma(l+1), \forall l \leq \frac{K}{2}$ *and* $\gamma(l+1) \geq \gamma(l), \forall l \geq \frac{K}{2}$, *then the privacy cost objective* $f(p, p'; \gamma)$ *is maximized when* $p \geq p'$.

*Proof of Lemma D.3.* WLOG, consider the output of DaRRM to be 1. By Eq. 41, the privacy cost objective $f(p, p'; \gamma)$, defined in Lemma 3.3 when $\delta = 0$, when the mechanisms are i.i.d., is equivalent to

$$f(p, p'; \gamma) = \frac{\Pr[\text{DaRRM}_\gamma(\mathcal{D}) = 1]}{\Pr[\text{DaRRM}_\gamma(\mathcal{D}') = 1]} - 1 \tag{83}$$

Hence, $f(p, p'; \gamma)$ is maximized when $\Pr[\mathcal{A}(\mathcal{D}) = 1] \geq \Pr[\mathcal{A}(\mathcal{D}') = 1]$. Note that

$$\Pr[\text{DaRRM}_\gamma(\mathcal{D}) = 1] = \frac{1}{2} \sum_{l=\frac{K+1}{2}}^{K} \gamma(l) \binom{K}{l} p^l (1-p)^{K-l} - \frac{1}{2} \sum_{l=0}^{\frac{K-1}{2}} \gamma(l) \binom{K}{l} p^l (1-p)^{K-l-1} + \frac{1}{2} \tag{84}$$

Define $g(l) = \begin{cases} -\frac{1}{2}\gamma(l) & \forall l \leq \frac{K}{2} \\ \frac{1}{2}\gamma(l) & \forall l \geq \frac{K}{2} \end{cases}$. Since $\gamma(l) \geq \gamma(l+1), \forall l \leq \frac{K}{2}$ and $\gamma(l+1) \geq \gamma(l), \forall l \geq \frac{K}{2}$, there is $g(l+1) \geq g(l), \forall l \in \{0, \ldots, K\}$. And replacing $\gamma(l)$ with $g(l)$,

$$\Pr[\text{DaRRM}_\gamma(\mathcal{D}) = 1] = \sum_{l=0}^{K} g(l) \binom{K}{l} p^l (1-p)^{K-l} \tag{85}$$

The gradient of the above probability w.r.t. $p$ is

$$\nabla_p \Pr[\text{DaRRM}_\gamma(\mathcal{D}) = 1] \tag{86}$$

$$= \sum_{l=0}^{K} g(l) \binom{K}{l} \left( l p^{l-1} (1-p)^{K-l} - (K-l) p^l (1-p)^{K-l-1} \right) \tag{87}$$

$$= \sum_{l=1}^{K} g(l) \binom{K-1}{l-1} \frac{K}{l} l p^{l-1} (1-p)^{K-l} - \sum_{l=0}^{K-1} \binom{K-1}{l} \frac{K}{K-l} (K-l) p^l (1-p)^{K-l-1} \tag{88}$$

$$= K \sum_{l=1}^{K} \binom{K-1}{l-1} p^{l-1} (1-p)^{K-l} - K \sum_{l=0}^{K-1} \binom{K-1}{l} p^l (1-p)^{K-l-1} \tag{89}$$

$$= K \sum_{l=0}^{K-1} g(l+1) \binom{K-1}{l} p^l (1-p)^{K-l-1} - K \sum_{l=0}^{K-1} g(l) \binom{K-1}{l} p^l (1-p)^{K-l-1} \tag{90}$$

$$= K \sum_{l=0}^{K-1} \left( g(l+1) - g(l) \right) \binom{K-1}{l} p^l (1-p)^{K-l-1} \tag{91}$$

Since $g(l+1) \geq g(l)$ and the binomial probability is always $\geq 0$, $\nabla_p \Pr[\text{DaRRM}_\gamma(\mathcal{D}) = 1] \geq 0$. This implies whenever $p \geq p'$, $\Pr[\text{DaRRM}_\gamma(\mathcal{D}) = 1] \geq \Pr[\text{DaRRM}_\gamma(\mathcal{D}') = 1]$. Hence, the privacy cost objective $f(p, p')$ is maximized when $p \geq p'$. $\square$

### D.2 Proof of Main Results on Privacy Amplification (Theorem 4.1)

**Roadmap.** To show Theorem 4.1, we show two parts separately: in section D.2.1, we show if the privacy allowance $m \geq \frac{K+1}{2}$, one can set $\gamma = 1$ (see Lemma D.4), and in section D.2.2 we show if $m \leq \frac{K-1}{2}$, one can set $\gamma$ to be the one such that DaRRM$_\gamma$ has the same output distribution as outputting the majority based on $2m - 1$ subsampled mechanisms and DaRRM$_\gamma$ still satisfies the privacy guarantee. We call this $\gamma$ function $\gamma_{\text{Double Subsampling}}$ (see Lemma D.8).

Showing Lemma D.4 is relatively straightforward (see Section D.2.1). To show Lemma D.8, we first introduce a class of *well-behaved* $\gamma$ functions called the *"symmetric-form"* family, and derive two clean sufficient conditions for $\gamma$ functions from the *"symmetric-form"* family such that DaRRM$_\gamma$

is $m\epsilon$-differentially private. After that, we show setting $\gamma_{\text{Double Subsampling}}$ as in Lemma D.8 satisfies the two conditions, and hence, DaRRM$_{\gamma_{\text{Double Subsampling}}}$ is $m\epsilon$-differentially private. Details are in Section D.2.2.

Finally, Theorem 4.1 follows directly from combining Lemma D.4 and Lemma D.8.

### D.2.1 Privacy Amplification under Large Privacy Allowance

We show the following lemma by showing that if one sets $\gamma(l) = 1, \forall l \in \{0, 1, \ldots, K\}$, then $m \geq \frac{K+1}{2}$ suffices to ensure the worst case probabilities $(p_i^*, p_i) = \arg\max_{p_i, p_i'} f(p, p'; \gamma)$ satisfy Eq. 37 in Lemma 3.3, and hence if $m \geq \frac{K+1}{2}$, DaRRM$_{\gamma=1}$ is $m\epsilon$-differentially private.

**Lemma D.4** (Privacy amplification under large privacy allowance $m \geq \frac{K+1}{2}$). *Consider using DaRRM to solve Problem 1.1 with $p_i = p$, $p_i' = p'$, $\forall i \in [K]$ and $\delta = \Delta = 0$. If the privacy allowance is $m \geq \frac{K+1}{2}$, one can set $\gamma(l) = 1, \forall l \in \{0, \ldots, K\}$ in DaRRM$_\gamma$ and DaRRM$_\gamma$ is $m\epsilon$ differentially private.*

*Proof of Lemma D.4.* Consider $\gamma(l) = 1, \forall l \in \{0, 1, \ldots, K\}$. Since $\gamma(l) \geq \gamma(l+1), \forall l \leq \frac{K-1}{2}$, $\gamma(l+1) \geq \gamma(l), \forall l \geq \frac{K+1}{2}$ and $\gamma(\frac{K-1}{2}) = \gamma(\frac{K+1}{2}) = 1 > 0$, by Lemma D.1, the worst case probabilities $(p^*, p'^*) = \arg\max_{p,p'} f(p, p')$ are on one of the two boundaries of $\mathcal{F}$, that is, they satisfy either $p = e^\epsilon p'$, $\forall p \in [0, \frac{1}{1+e^\epsilon}], p' \in [0, \frac{1}{1+e^\epsilon}]$ or $1 - p' = e^\epsilon(1-p)$, $\forall p \in [\frac{1}{1+e^{-\epsilon}}, 1], p' \in [\frac{1}{1+e^\epsilon}, 1]$. We now find the local maximums on the boundary $p = e^\epsilon p'$ and $1 - p' = e^\epsilon(1-p)$ separately and then find the global maximum $(p^*, p'^*) = \arg\max_{p,p'} f(p, p'; \gamma)$.

**Part I: Finding the local worse case probabilities on the boundary $p = e^\epsilon p'$.**

The privacy cost objective $f(p, p'; \gamma)$ on the boundary $p = e^\epsilon p'$, $\forall p \in [0, \frac{1}{e^{-\epsilon}+1}], p' \in [0, \frac{1}{1+e^\epsilon}]$, can be written as the following by substituting $p$ with $p'$ (and omitting $\gamma$ for convenience):

$$f(p') = \sum_{l=0}^{\frac{K-1}{2}} (e^{m\epsilon} \binom{K}{l} p'^l (1-p')^{K-l} - \binom{K}{l} (e^\epsilon p')^l (1 - e^\epsilon p')^{K-l}) \cdot \gamma(l) \tag{92}$$

$$+ \sum_{l=\frac{K+1}{2}}^{K} (\binom{K}{l} (e^\epsilon p')^l (1 - e^\epsilon p')^{K-l} - e^{m\epsilon} \binom{K}{l} p'^l (1-p')^{K-l}) \cdot \gamma(l)$$

And the gradient w.r.t. $p'$ is

$$\nabla_{p'} f(p') = \sum_{l=0}^{\frac{K-1}{2}} \left( e^{m\epsilon} \binom{K}{l} (l p'^{l-1} (1-p')^{K-l} - p'^l (K-l)(1-p')^{K-l-1}) \right. \tag{93}$$

$$\left. - e^\epsilon \binom{K}{l} (l(e^\epsilon p')^{l-1}(1 - e^\epsilon p')^{K-l} - e^{\epsilon l} p'^l (K-l)(1 - e^\epsilon p')^{K-l-1}) \right) \cdot \gamma(l)$$

$$+ \sum_{l=\frac{K+1}{2}}^{K} \left( e^\epsilon \binom{K}{l} (l(e^\epsilon p')^{l-1}(1 - e^\epsilon p')^{K-l} - e^{\epsilon l} p'^l (K-l)(1 - e^\epsilon p')^{K-l-1}) \right.$$

$$\left. - e^{m\epsilon} \binom{K}{l} (l p'^{l-1} (1-p')^{K-l} - p'^l (K-l)(1-p')^{K-l-1}) \right) \cdot \gamma(l)$$

$$\nabla_{p'} f(p') \tag{94}$$

$$= -K \sum_{l=0}^{\frac{K-1}{2}} e^{m\epsilon} \binom{K-1}{l} p'^l (1-p')^{K-l-1} \gamma(l) + K \sum_{l=\frac{K+1}{2}}^{K-1} e^{m\epsilon} \binom{K-1}{l} p'^l (1-p')^{K-l-1} \gamma(l)$$

$$+ K \sum_{l=0}^{\frac{K-1}{2}} e^\epsilon \binom{K-1}{l} (\epsilon p')^\epsilon (1 - e^\epsilon p')^{K-l-1} \gamma(l) - K \sum_{l=\frac{K+1}{2}}^{K-1} e^\epsilon \binom{K-1}{l} (e^\epsilon p')^l (1 - e^\epsilon p')^{K-l-1} \gamma(l)$$

$$+ K \sum_{l=0}^{\frac{K-1}{2}-1} e^{m\epsilon} \binom{K-1}{l} p'^l (1-p')^{K-l-1} \gamma(l+1) - K \sum_{l=\frac{K-1}{2}}^{K-1} e^{m\epsilon} \binom{K-1}{l} p'^l (1-p')^{K-l-1} \gamma(l+1)$$

$$- K \sum_{l=0}^{\frac{K-1}{2}-1} e^{\epsilon} \binom{K-1}{l} (e^\epsilon p')^l (1-e^\epsilon p')^{K-l-1} \gamma(l+1) + K \sum_{l=\frac{K-1}{2}}^{K-1} e^{\epsilon} \binom{K-1}{l} (e^\epsilon p')^l (1-e^\epsilon p')^{K-l-1} \gamma(l+1)$$

That is,

$$\frac{\nabla_{p'} f(p')}{K} \tag{95}$$

$$= e^{m\epsilon} \sum_{l=0}^{\frac{K-1}{2}-1} \binom{K-1}{l} p'^l (1-p')^{K-l-1} \Big( \gamma(l+1) - \gamma(l) \Big) - 2e^{m\epsilon} \binom{K-1}{\frac{K-1}{2}} p'^{\frac{K-1}{2}} (1-p')^{\frac{K-1}{2}} \gamma(\frac{K-1}{2})$$

$$+ e^{m\epsilon} \sum_{l=\frac{K+1}{2}}^{K-1} \binom{K-1}{l} p'^l (1-p')^{K-l-1} \Big( \gamma(l) - \gamma(l+1) \Big)$$

$$+ e^{\epsilon} \sum_{l=0}^{\frac{K-1}{2}-1} \binom{K-1}{l} (e^\epsilon p')^l (1-e^\epsilon p')^{K-l-1} \Big( \gamma(l) - \gamma(l+1) \Big) + 2e^{\epsilon} \binom{K-1}{\frac{K-1}{2}} (e^\epsilon p')^{\frac{K-1}{2}} (1-e^\epsilon p')^{\frac{K-1}{2}} \gamma(\frac{K-1}{2})$$

$$+ e^{\epsilon} \sum_{l=\frac{K+1}{2}}^{K-1} \binom{K-1}{l} (e^\epsilon p')^l (1-e^\epsilon p')^{K-l-1} \Big( \gamma(l+1) - \gamma(l) \Big)$$

When $\gamma(l) = 1$, the above gradient is then

$$\frac{\nabla_{p'} f(p')}{K} = -2e^{m\epsilon} \binom{K-1}{\frac{K-1}{2}} p'^{\frac{K-1}{2}} (1-p')^{\frac{K-1}{2}} + 2e^{\epsilon} \binom{K-1}{\frac{K-1}{2}} (e^\epsilon p')^{\frac{K-1}{2}} (1-e^\epsilon p')^{\frac{K-1}{2}} \tag{96}$$

If $p' = 0$, then $p = 0$, and the original privacy cost objective is $f(0, 0'; \gamma = 1) = e^{m\epsilon} - 1$, which satisfies Eq. 37 in Lemma 3.3 (i.e. DaRRM$_{\gamma=1}$ is $m\epsilon$-differentially private at $(p, p') = (0, 0)$).

For $p' > 0$, if $\nabla_{p'} f(p') \le 0$, then we know $f(p') \le f(0)$, i.e. the worst case probabilities on the boundary $p = e^\epsilon p'$ is $(p, p') = (0, 0)$. To ensure $\nabla_{p'} f(p') \le 0$,

$$\nabla_{p'} f(p') \le 0 \tag{97}$$

$$\Leftrightarrow -2e^{m\epsilon} \binom{K-1}{\frac{K-1}{2}} p'^{\frac{K-1}{2}} (1-p')^{\frac{K-1}{2}} \le -2e^{\epsilon} \binom{K-1}{\frac{K-1}{2}} (e^\epsilon p')^{\frac{K-1}{2}} (1-e^\epsilon p')^{\frac{K-1}{2}} \tag{98}$$

$$\Leftrightarrow e^{m\epsilon} \binom{K-1}{\frac{K-1}{2}} p'^{\frac{K-1}{2}} (1-p')^{\frac{K-1}{2}} \ge e^{\epsilon} \binom{K-1}{\frac{K-1}{2}} (e^\epsilon p')^{\frac{K-1}{2}} (1-e^\epsilon p')^{\frac{K-1}{2}} \tag{99}$$

Let

$$\mathcal{R} := \frac{\text{L.H.S.}}{\text{R.H.S.}} = \frac{e^{m\epsilon} \binom{K-1}{\frac{K-1}{2}} p'^{\frac{K-1}{2}} (1-p')^{\frac{K-1}{2}}}{e^{\epsilon} \binom{K-1}{\frac{K-1}{2}} (e^\epsilon p')^{\frac{K-1}{2}} (1-e^\epsilon p')^{\frac{K-1}{2}}} = \frac{e^{m\epsilon}}{e^{\frac{K+1}{2}\epsilon}} \cdot \left( \frac{1-p'}{1-e^\epsilon p'} \right)^{\frac{K-1}{2}} \tag{100}$$

and $\mathcal{R} \ge 1 \Leftrightarrow \nabla_{p'} f(p') \le 0$. Since $\frac{1-p'}{1-e^\epsilon p'} \ge 1$, $\mathcal{R} \ge e^{(m-\frac{K+1}{2}) \cdot \epsilon}$.

Hence, to make sure $\mathcal{R} \ge 1$ (and so $\nabla_{p'} f(p') \le 0$), $m \ge \frac{K+1}{2}$ suffices.

**Part II: Finding the local worst case probabilities on the boundary** $1 - p' = e^\epsilon (1 - p)$.

Now consider the maximum point on the other boundary $1 - p' = e^\epsilon (1 - p)$ for $p' \in [\frac{1}{1+e^\epsilon}, 1]$ and $p \in [\frac{1}{1+e^{-\epsilon}}, 1]$. Following the privacy cost objective $f(p, p'; \gamma)$ and let $q = 1 - p$ and $p' = 1 - q'$, the objective is

$$f(q, q'; \gamma) = \sum_{l=0}^{\frac{K-1}{2}} \left( e^{m\epsilon} \binom{K}{l} (1-q')^l q'^{K-l} - \binom{K}{l} (1-q)^l q^{K-l} \right) \cdot \gamma(l) \tag{101}$$

$$+ \sum_{l=\frac{K+1}{2}}^{K} \left( \binom{K}{l}(1-q)^l q^{K-l} - e^{m\epsilon}\binom{K}{l}(1-q')^l {q'}^{K-l} \right) \cdot \gamma(l)$$

Substituting $1 - p' = e^\epsilon(1-p) \Leftrightarrow q' = e^\epsilon q$ (and omitting $\gamma$ for convenience), the privacy cost objective can be written as

$$f(q) = \sum_{l=0}^{\frac{K-1}{2}} \left( e^{m\epsilon}\binom{K}{l}(1-e^\epsilon q)^l(e^\epsilon q)^{K-l} - \binom{K}{l}(1-q)^l q^{K-l} \right) \cdot \gamma(l) \tag{102}$$

$$+ \sum_{l=\frac{K+1}{2}}^{K} \left( \binom{K}{l}(1-q)^l q^{K-l} - e^{m\epsilon}\binom{K}{l}(1-e^\epsilon q)^l(e^\epsilon q)^{K-l} \right) \cdot \gamma(l)$$

And the gradient w.r.t. $q$ is

$$\nabla_q f(q) = \sum_{l=0}^{\frac{K-1}{2}} \left( e^{m\epsilon}\binom{K}{l}\left((-e^\epsilon)l(1-e^\epsilon q)^{l-1}(e^\epsilon q)^{K-l} + e^\epsilon(K-l)(1-e^\epsilon q)^l(e^\epsilon q)^{K-l-1}\right) \right.$$

$$\tag{103}$$

$$\left. - \binom{K}{l}\left(-l(1-q)^{l-1}q^{K-l} + (K-l)(1-q)^l q^{K-l-1}\right) \right) \cdot \gamma(l)$$

$$+ \sum_{l=\frac{K+1}{2}}^{K} \left( \binom{K}{l}\left(-l(1-q)^{l-1}q^{K-l} + (K-l)(1-q)^l q^{K-l-1}\right) \right.$$

$$\left. - e^{m\epsilon}\binom{K}{l}\left((-e^\epsilon)l(1-e^\epsilon q)^{l-1}(e^\epsilon q)^{K-l} + e^\epsilon(K-l)(1-e^\epsilon q)^l(e^\epsilon q)^{K-l-1}\right) \right) \cdot \gamma(l)$$

$$\nabla_q f(q) = - \sum_{l=1}^{\frac{K-1}{2}} e^{(m+1)\epsilon}\binom{K-1}{l-1}\frac{K}{l}l(1-e^\epsilon q)^{l-1}(e^\epsilon q)^{K-l} \cdot \gamma(l) \tag{104}$$

$$+ \sum_{l=0}^{\frac{K-1}{2}} e^{(m+1)\epsilon}\binom{K-1}{l}\frac{K}{K-l}(K-l)(1-e^\epsilon q)^l(e^\epsilon q)^{K-l-1} \cdot \gamma(l)$$

$$+ \sum_{l=1}^{\frac{K-1}{2}} \binom{K-1}{l-1}\frac{K}{l}l(1-q)^{l-1}q^{K-l} \cdot \gamma(l) - \sum_{l=0}^{\frac{K-1}{2}} \binom{K-1}{l}\frac{K}{K-l}(K-l)(1-q)^l q^{K-l-1} \cdot \gamma(l)$$

$$- \sum_{l=\frac{K+1}{2}}^{K} \binom{K-1}{l-1}\frac{K}{l}l(1-q)^{l-1}q^{K-l} \cdot \gamma(l) + \sum_{l=\frac{K+1}{2}}^{K-1} \binom{K-1}{l}\frac{K}{K-l}(K-l)(1-q)^l q^{K-l-1} \cdot \gamma(l)$$

$$+ \sum_{l=\frac{K+1}{2}}^{K} e^{(m+1)\epsilon}\binom{K-1}{l-1}\frac{K}{l}l(1-e^\epsilon q)^{l-1}(e^\epsilon q)^{K-l} \cdot \gamma(l)$$

$$- \sum_{l=\frac{K+1}{2}}^{K-1} e^{(m+1)\epsilon}\binom{K-1}{l}\frac{K}{K-l}(K-l)(1-e^\epsilon q)^l(e^\epsilon q)^{K-l-1} \cdot \gamma(l)$$

$$\nabla_q f(q) = -K \sum_{l=1}^{\frac{K-1}{2}} e^{(m+1)\epsilon}\binom{K-1}{l-1}(1-e^\epsilon q)^{l-1}(e^\epsilon q)^{K-l} \cdot \gamma(l) \tag{105}$$

$$+ K \sum_{l=0}^{\frac{K-1}{2}} e^{(m+1)\epsilon}\binom{K-1}{l}(1-e^\epsilon q)^l(e^\epsilon q)^{K-l-1} \cdot \gamma(l)$$

$$+ K \sum_{l=1}^{\frac{K-1}{2}} \binom{K-1}{l-1}(1-q)^{l-1}q^{K-l} \cdot \gamma(l) - K \sum_{l=0}^{\frac{K-1}{2}} \binom{K-1}{l}(1-q)^l q^{K-l-1} \cdot \gamma(l)$$

$$- K \sum_{l=\frac{K+1}{2}}^{K} \binom{K-1}{l-1}(1-q)^{l-1}q^{K-l} \cdot \gamma(l) + K \sum_{l=\frac{K+1}{2}}^{K-1} \binom{K-1}{l}(1-q)^l q^{K-l-1} \cdot \gamma(l)$$

$$+ K \sum_{l=\frac{K+1}{2}}^{K} e^{(m+1)\epsilon} \binom{K-1}{l-1}(1-e^\epsilon q)^{l-1}(e^\epsilon q)^{K-l} \cdot \gamma(l)$$

$$- K \sum_{l=\frac{K+1}{2}}^{K-1} e^{(m+1)\epsilon} \binom{K-1}{l}(1-e^\epsilon q)^l(e^\epsilon q)^{K-l-1} \cdot \gamma(l)$$

The above is

$$\frac{\nabla_q f(q)}{K}$$

$$= - \sum_{l=0}^{\frac{K-1}{2}-1} e^{(m+1)\epsilon} \binom{K-1}{l}(1-e^\epsilon q)^l(e^\epsilon q)^{K-l-1} \cdot \gamma(l+1) \tag{106}$$

$$+ \sum_{l=0}^{\frac{K-1}{2}} e^{(m+1)\epsilon} \binom{K-1}{l}(1-e^\epsilon q)^l(e^\epsilon q)^{K-l-1} \cdot \gamma(l)$$

$$+ \sum_{l=0}^{\frac{K-1}{2}-1} \binom{K-1}{l}(1-q)^l q^{K-l-1} \cdot \gamma(l+1) - \sum_{l=0}^{\frac{K-1}{2}} \binom{K-1}{l}(1-q)^l q^{K-l-1} \cdot \gamma(l)$$

$$- \sum_{l=\frac{K-1}{2}}^{K-1} \binom{K-1}{l}(1-q)^l q^{K-l-1} \cdot \gamma(l+1) + \sum_{l=\frac{K+1}{2}}^{K-1} \binom{K-1}{l}(1-q)^l q^{K-l-1} \cdot \gamma(l)$$

$$+ \sum_{l=\frac{K-1}{2}}^{K-1} e^{(m+1)\epsilon} \binom{K-1}{l}(1-e^\epsilon q)^l(e^\epsilon q)^{K-l-1} \cdot \gamma(l+1)$$

$$- \sum_{l=\frac{K+1}{2}}^{K-1} e^{(m+1)\epsilon} \binom{K-1}{l}(1-e^\epsilon q)^l(e^\epsilon q)^{K-l-1} \cdot \gamma(l)$$

$$= \sum_{l=0}^{\frac{K-1}{2}-1} e^{(m+1)\epsilon} \binom{K-1}{l}(1-e^\epsilon q)^l(e^\epsilon q)^{K-l-1} \cdot \Big(\gamma(l) - \gamma(l+1)\Big) \tag{107}$$

$$+ \sum_{l=\frac{K+1}{2}}^{K-1} \binom{K-1}{l}(1-e^\epsilon q)^l(e^\epsilon q)^{K-l-1} \cdot \Big(\gamma(l+1) - \gamma(l)\Big)$$

$$+ 2e^{(m+1)\epsilon} \binom{K-1}{\frac{K-1}{2}}(1-e^\epsilon q)^{\frac{K-1}{2}}(e^\epsilon q)^{\frac{K-1}{2}} \cdot \gamma(\frac{K-1}{2}) \qquad \text{Since } \gamma(\frac{K+1}{2}) = \gamma(\frac{K-1}{2})$$

$$+ \sum_{l=0}^{\frac{K-1}{2}-1} \binom{K-1}{l}(1-q)^l q^{K-l-1} \cdot \Big(\gamma(l+1) - \gamma(l)\Big)$$

$$+ \sum_{l=\frac{K+1}{2}}^{K-1} (1-q)^l q^{K-l-1} \cdot \Big(\gamma(l) - \gamma(l+1)\Big)$$

$$- 2\binom{K-1}{\frac{K-1}{2}}(1-q)^{\frac{K-1}{2}}q^{\frac{K-1}{2}} \cdot \gamma(\frac{K-1}{2}) \qquad \text{Since } \gamma(\frac{K-1}{2}) = \gamma(\frac{K+1}{2})$$

When $\gamma(l) = 1$, then the above gradient is then

$$\frac{\nabla_q f(q)}{K} = 2e^{(m+1)\epsilon} \binom{K-1}{\frac{K-1}{2}}(1 - e^\epsilon q)^{\frac{K-1}{2}}(e^\epsilon q)^{\frac{K-1}{2}} - 2\binom{K-1}{\frac{K-1}{2}}(1-q)^{\frac{K-1}{2}} q^{\frac{K-1}{2}} \quad (108)$$

Since $p \in [\frac{1}{1+e^{-\epsilon}}, 1]$, and $q = 1 - p \in [0, \frac{1}{1+e^\epsilon}]$, $(1 - e^\epsilon q)(e^\epsilon q) \geq (1 - q)q$. Furthermore, since $e^{(m+1)\epsilon} \geq 1$, $\nabla_q f(q) \geq 0$. Hence, $f(q)$ achieves the maximum at $q = \frac{1}{1+e^\epsilon}$ — that is, at $p = 1 - \frac{1}{1+e^\epsilon} = \frac{1}{1+e^{-\epsilon}}$. Since $1 - p' = e^\epsilon(1-p)$, $f(q)$ achieves the maximum at $p' = 1 - e^\epsilon(1-p) = 1 - e^\epsilon(1 - \frac{1}{1+e^{-\epsilon}}) = \frac{1}{1+e^\epsilon}$. Notice that $p, p' = (\frac{1}{1+e^{-\epsilon}}, \frac{1}{1+e^\epsilon}) = \arg\min_{p,p'} f(p, p'; \gamma)$ on the other boundary $p = e^\epsilon p'$, $\forall p \in [0, \frac{1}{e^{-\epsilon}+1}]$. Hence, the global worst case probabilities $(p^*, p'^*)$ are not on the boundary $1 - p' = e^\epsilon(1 - p)$.

**Part III: Global maximum point** $(p, p')$

The above implies the global maximum points (aka. global worst case probabilities) $(p^*, p'^*) = \arg\max_{p,p'} f(p, p') = (0, 0)$ if $m \geq \frac{K+1}{2}$, and $f(0, 0) = e^{m\epsilon} - 1$.

Therefore, by Lemma 3.3, if $m \geq \frac{K+1}{2}$, setting $\gamma(l) = 1, \forall l \in \{0, \ldots, K\}$ ensures DaRRM$_\gamma$ is $m\epsilon$-differentially private. $\qquad\square$

### D.2.2 PRIVACY AMPLIFICATION UNDER SMALL PRIVACY ALLOWANCE

**Roadmap.** To show the privacy amplification under a small privacy allowance $m \leq \frac{K-1}{2}$ in Lemma D.8, we first observe that the $\gamma$ function corresponding to natural subsampling as shown in Lemma 3.1 falls into a special family of $\gamma$ functions, which we call the "symmetric form family", that are a combination of two functions of a specific form on support $\{0, \ldots, \frac{K}{2}\}$ and $\{\frac{K}{2}, \ldots, K\}$ and are symmetric around $\frac{K}{2}$ — that is, $\gamma(l) = \begin{cases} 1 - 2h(l) & l \leq \frac{K}{2} \\ 2h(l) - 1 & l \geq \frac{K}{2} \end{cases}$ and $h(l) + h(K-1) = 1$, where $h(l)$ is monotonically increasing on the support. It is not hard to see these functions are *well-behaved*, and so we can apply Lemma D.1 in such cases to limit the region to search for the worst case probabilities. For a $\gamma$ function that falls under this "symmetric form family", we show two clean sufficient conditions for DaRRM$_\gamma$ to be $m\epsilon$-differentially private in terms of the expectation of the $\gamma$ function applied to some Binomial random variables, as in Lemma D.5.

To show the privacy amplification results under a small privacy allowance $m$, we further need two building blocks on recurrence relationships in expectation of Binomial random variables and Hypergeometric random variables in Lemma D.6 and Lemma D.7.

Finally, based on Lemma D.6 and Lemma D.7, we show in Lemma D.8 that $\gamma_{\text{Double Subsampling}}$, i.e., the $\gamma$ function that enables DaRRM to have the same distribution as outputting the majority of $2m - 1$ subsampled mechanisms, belongs to the "symmetric form family", and satisfies the sufficient conditions as stated in Lemma D.5. Hence DaRRM$_{\gamma_{\text{Double Subsampling}}}$ is $m\epsilon$-differentially private.

**Lemma D.5** (Privacy conditions of "symmetric form family"). *Consider a* $\gamma : \{0, 1, \ldots, K\} \to [0, 1]$ *function that is of the form*

$$\gamma(l) = \begin{cases} 1 - 2h(l) & l \in \{0, 1, \ldots, \frac{K-1}{2}\} \\ 2h(l) - 1 & l \in \{\frac{K+1}{2}, \ldots, K\} \end{cases} \quad (109)$$

*where $h(l)$ is a monotonically increasing function on $l \in \{0, \ldots, K\}$ and $h(l) + h(K - l) = 1$. Let random variables $X \sim Binom(K - 1, p)$ and $Y \sim Binom(K - 1, e^\epsilon p)$. Let random variables $\hat{X} \sim Binom(K - 1, 1 - e^\epsilon(1 - p))$ and $\hat{Y} \sim Binom(K - 1, p)$. If this $\gamma$ function further satisfies the following two conditions:*

$$e^{m\epsilon}\mathbb{E}_X[h(X + 1) - h(X)] \geq e^\epsilon \mathbb{E}_Y[h(Y + 1) - h(Y)], \quad \forall p \in [0, \frac{1}{1+e^\epsilon}] \quad (110)$$

$$e^{(m+1)\epsilon}\mathbb{E}_{\hat{X}}[h(\hat{X} + 1) - h(\hat{X})] \geq \mathbb{E}_{\hat{Y}}[h(\hat{Y} + 1) - h(\hat{Y})], \quad \forall p \in [\frac{1}{1+e^{-\epsilon}}, 1] \quad (111)$$

*then Algorithm DaRRM$_\gamma$ is $m\epsilon$-differentially private.*

*Proof of Lemma D.5.* Since $h(l+1) \geq h(l)$ on $l \in \{0, \ldots, K\}$, there is $\gamma(l) \geq \gamma(l+1), \forall l \leq \frac{K}{2}$ and $\gamma(l+1) \geq \gamma(l), \forall l \geq \frac{K}{2}$. Furthermore, since $h(l) + h(K-l) = 1, \gamma(\frac{K-1}{2}) = 1 - 2h(\frac{K-1}{2}) = 1 - 2(1 - h(\frac{K+1}{2})) = 2h(\frac{K+1}{2}) - 1$. And so by Lemma D.1, the worst case probabilities $(p^*, p'^*) = \arg\max_{p,p'} f(p, p'; \gamma)$ satisfy one of the two following: $p = e^\epsilon p'$, $\forall p \in [0, \frac{1}{1+e^{-\epsilon}}], p' \in [0, \frac{1}{1+e^\epsilon}]$, or $1 - p' = e^\epsilon(1 - p), \forall p \in [\frac{1}{1+e^{-\epsilon}}, 1], p' \in [\frac{1}{1+e^\epsilon}, 1]$.

On the boundary $p = e^\epsilon p'$, where $p' \in [0, \frac{1}{1+e^\epsilon}]$, the privacy cost objective can be re-written as

$$f(p, p') = f(p') = \sum_{l=0}^{\frac{K-1}{2}} (e^{m\epsilon}\binom{K}{l}p'^l(1-p')^{K-l} - \binom{K}{l}(e^\epsilon p')^l(1 - e^\epsilon p')^{K-l}) \cdot \gamma(l) \quad (112)$$

$$+ \sum_{l=\frac{K+1}{2}}^{K} (\binom{K}{l}(e^\epsilon p')^l(1 - e^\epsilon p')^{K-l} - e^{m\epsilon}\binom{K}{l}p'^l(1-p')^{K-l}) \cdot \gamma(l)$$

as in Eq. 92, and as in Eq. 95, the gradient w.r.t. $p'$ is

$$\frac{\nabla_{p'} f(p')}{K} = e^{m\epsilon}\sum_{l=0}^{\frac{K-1}{2}-1}\binom{K-1}{l}p'^l(1-p')^{K-l-1}\Big(\gamma(l+1) - \gamma(l)\Big) - 2e^{m\epsilon}\binom{K-1}{\frac{K-1}{2}}p'^{\frac{K-1}{2}}(1-p')^{\frac{K-1}{2}}\gamma(\frac{K-1}{2})$$

$$(113)$$

$$+ e^{m\epsilon}\sum_{l=\frac{K+1}{2}}^{K-1}\binom{K-1}{l}p'^l(1-p')^{K-l-1}\Big(\gamma(l) - \gamma(l+1)\Big)$$

$$+ e^\epsilon\sum_{l=0}^{\frac{K-1}{2}-1}\binom{K-1}{l}(e^\epsilon p')^l(1 - e^\epsilon p')^{K-l-1}\Big(\gamma(l) - \gamma(l+1)\Big) + 2e^\epsilon\binom{K-1}{\frac{K-1}{2}}(e^\epsilon p')^{\frac{K-1}{2}}(1 - e^\epsilon p')^{\frac{K-1}{2}}\gamma(\frac{K-1}{2})$$

$$+ e^\epsilon\sum_{l=\frac{K+1}{2}}^{K-1}\binom{K-1}{l}(e^\epsilon p')^l(1 - e^\epsilon p')^{K-l-1}\Big(\gamma(l+1) - \gamma(l)\Big)$$

With this family of $\gamma$ function,

1. When $l \leq \frac{K}{2}, \gamma(l) - \gamma(l+1) = (1 - 2h(l)) - (1 - 2h(l+1)) = 2h(l+1) - 2h(l)$

2. When $l \geq \frac{K}{2}, \gamma(l+1) - \gamma(l) = (2h(l+1) - 1) - (2h(l) - 1) = 2h(l+1) - 2h(l)$

3. Since $\gamma(\frac{K-1}{2}) = \gamma(\frac{K+1}{2})$,

$$2\gamma(\frac{K-1}{2}) = \Big(\gamma(\frac{K-1}{2}) + \gamma(\frac{K+1}{2})\Big) \quad (114)$$

$$= \Big(1 - 2h(\frac{K-1}{2}) + 2h(\frac{K+1}{2}) - 1\Big) \quad (115)$$

$$= 2h(\frac{K+1}{2}) - 2h(\frac{K-1}{2}) \quad (116)$$

and so the gradient is equivalent to

$$\frac{\nabla_{p'} f(p')}{K} = -e^{m\epsilon}\sum_{l=0}^{K-1}\binom{K-1}{l}p'^l(1-p')^{K-l}\Big(2h(l+1) - 2h(l)\Big) \quad (117)$$

$$+ e^\epsilon\sum_{l=0}^{K-1}\binom{K-1}{l}(e^\epsilon p')^l(1 - e^\epsilon p')^{K-l-1}\Big(2h(l+1) - 2h(l)\Big)$$

$$= -2e^{m\epsilon}\mathbb{E}_X[h(X+1) - h(X)] + 2e^\epsilon\mathbb{E}_Y[h(Y+1) - h(Y)] \quad (118)$$

where $X \sim \text{Binom}(K-1, p')$ and $Y \sim \text{Binom}(K-1, e^\epsilon p')$. Hence,

$$\nabla_{p'} f(p') \leq 0 \Leftrightarrow e^\epsilon\mathbb{E}_Y[h(Y+1) - h(Y)] \leq e^{m\epsilon}\mathbb{E}_X[h(X+1) - h(X)] \quad (119)$$

On the boundary $1 - p' = e^\epsilon(1 - p)$, where $p \in [\frac{1}{1+e^{-\epsilon}}, 1]$. Let $q = 1 - p$ and $q' = 1 - p'$ for $q \in [0, \frac{1}{1+e^\epsilon}]$, the privacy cost objective can be re-written as

$$f(q) = \sum_{l=0}^{\frac{K-1}{2}} \left( e^{m\epsilon} \binom{K}{l}(1 - e^\epsilon q)^l (e^\epsilon q)^{K-l} - \binom{K}{l}(1-q)^l q^{K-l} \right) \cdot \gamma(l) \tag{120}$$

$$+ \sum_{l=\frac{K+1}{2}}^{K} \left( \binom{K}{l}(1-q)^l q^{K-l} - e^{m\epsilon}\binom{K}{l}(1 - e^\epsilon q)^l (e^\epsilon q)^{K-l} \right) \cdot \gamma(l)$$

as in Eq. 102, and as in Eq. 103, the gradient w.r.t. $q$ is

$$\frac{\nabla_q f(q)}{K} = \sum_{l=0}^{\frac{K-1}{2}-1} e^{(m+1)\epsilon}\binom{K-1}{l}(1 - e^\epsilon q)^l (e^\epsilon q)^{K-l-1} \cdot \Big(\gamma(l) - \gamma(l+1)\Big) \tag{121}$$

$$+ \sum_{l=\frac{K+1}{2}}^{K-1} \binom{K-1}{l}(1 - e^\epsilon q)^l (e^\epsilon q)^{K-l-1} \cdot \Big(\gamma(l+1) - \gamma(l)\Big)$$

$$+ 2e^{(m+1)\epsilon}\binom{K-1}{\frac{K-1}{2}}(1 - e^\epsilon q)^{\frac{K-1}{2}}(e^\epsilon q)^{\frac{K-1}{2}} \cdot \gamma(\frac{K-1}{2})$$

$$+ \sum_{l=0}^{\frac{K-1}{2}-1} \binom{K-1}{l}(1-q)^l q^{K-l-1} \cdot \Big(\gamma(l+1) - \gamma(l)\Big)$$

$$+ \sum_{l=\frac{K+1}{2}}^{K-1} (1-q)^l q^{K-l-1} \cdot \Big(\gamma(l) - \gamma(l+1)\Big)$$

$$- 2\binom{K-1}{\frac{K-1}{2}}(1-q)^{\frac{K-1}{2}} q^{\frac{K-1}{2}} \cdot \gamma(\frac{K-1}{2})$$

With this family of $\gamma$ function, the gradient above is equivalent to

$$\frac{\nabla_q f(q)}{K} = e^{(m+1)\epsilon}\sum_{l=0}^{K-1}\binom{K-1}{l}(1 - e^\epsilon q)^l (e^\epsilon q)^{K-l-1} \cdot \Big(2h(l+1) - 2h(l)\Big) \tag{122}$$

$$- \sum_{l=0}^{K}\binom{K-1}{l}(1-q)^l q^{K-l-1} \cdot \Big(2h(l+1) - 2h(l)\Big)$$

$$= 2e^{(m+1)\epsilon}\mathbb{E}_{\hat{X}}[h(\hat{X}+1) - h(\hat{X})] - 2\mathbb{E}_{\hat{Y}}[h(\hat{Y}+1) - h(\hat{Y})] \tag{123}$$

where $\hat{X} \sim \text{Binom}(K-1, 1 - e^\epsilon(1-p))$ and $\hat{Y} \sim \text{Binom}(K-1, p)$.

$$\nabla_q f(q) \geq 0 \Leftrightarrow e^{(m+1)\epsilon}\mathbb{E}_{\hat{X}}[h(\hat{X}+1) - h(\hat{X})] \geq \mathbb{E}_{\hat{Y}}[h(\hat{Y}+1) - h(\hat{Y})] \tag{124}$$

Recall $q \in [0, \frac{1}{1+e^\epsilon}]$. The above implies the maximum on this boundary is at point $q = \frac{1}{1+e^\epsilon}$ — that is, at point $(p, p') = (\frac{1}{1+e^{-\epsilon}}, \frac{1}{1+e^\epsilon})$. Notice this is the minimum on the first boundary $p = e^\epsilon p'$. Hence, the global maximum of the cost objective is at $(p, p') = (0, 0)$, and since the maximum $f(0,0) = e^{m\epsilon} - 1 \leq e^{m\epsilon} - 1$, this further implies the algorithm is $m\epsilon$ differentially private. $\square$

**Lemma D.6** (Binomial Expectation Recurrence Relationship (Theorem 2.1 of Zhang et al. (2019))). *Let $X_{(K-1)} \sim Binom(K-1, p)$ and $X_{(K)} \sim Binom(K, p)$. Let $g(x)$ be a function with $-\infty < \mathbb{E}[g(X_{(K-1)})] < \infty$ and $-\infty < g(-1) < \infty$, then*

$$Kp\mathbb{E}_{X_{(K-1)}}[g(X_{(K-1)})] = \mathbb{E}_{X_{(K)}}[X_{(K)}g(X_{(K)}-1)] \tag{125}$$

**Lemma D.7.** *Given $i, m, K \in \mathbb{Z}$, $K \geq 1$, $0 \leq i \leq m \leq K$, let $X_{(K)} \sim Binom(K, p)$ for some $p \in [0, 1]$, there is*

$$\frac{1}{\binom{K}{m}}\mathbb{E}_{X_{(K)}}\left[\binom{X}{i}\binom{K-X}{m-i}\right] = \binom{m}{i}p^i(1-p)^{m-i} \tag{126}$$

*Proof of Lemma D.7.* We show the above statement by induction on $K$ and $m$.

Base Case: $K = 1$.

1. If $m = 0$, then $i = 0$. $\frac{1}{\binom{1}{0}}\mathbb{E}_{X_{(1)}}[\binom{X}{0}\binom{1-X}{0}] = \mathbb{E}_{X_{(1)}}[1] = 1$, and $\binom{0}{0}p^0(1-p)^0 = 1$.

2. If $m = 1$,

   (a) $i = 0$, $\frac{1}{\binom{1}{1}}\mathbb{E}_{X_{(1)}}[\binom{X}{0}\binom{1-X}{1}] = \mathbb{E}_{X_{(1)}}[1 - X] = 1 - p$, and $\binom{1}{0}p^0(1-p)^1 = 1 - p$

   (b) $i = 1$, $\frac{1}{\binom{1}{1}}\mathbb{E}_{X_{(1)}}[\binom{X}{1}\binom{1-X}{0}] = \mathbb{E}_{X_{(1)}}[X] = p$, and $\binom{1}{1}p^1(1-p)^0 = p$.

Hence, the statement holds for the base case.

Induction Hypothesis: Suppose the statement holds for some $K \geq 1$ and $0 \leq i \leq m \leq K$. Consider $1 \leq i \leq m \leq K + 1$,

$$\frac{1}{\binom{K+1}{m}}\mathbb{E}_{X_{(K+1)}}\left[\binom{X}{i}\binom{K+1-X}{m-i}\right] \tag{127}$$

$$= \frac{1}{\binom{K+1}{m}}\mathbb{E}_{X_{(K+1)}}\left[\frac{X!}{i!(X-i)!}\frac{(K+1-X)!}{(m-i)!(K+1-X-(m-i))!}\right] \tag{128}$$

$$= \frac{1}{\binom{K+1}{m}i!(m-i)!}\mathbb{E}_{X_{(K+1)}}\left[X\frac{(X-1)!}{((X-1)-(i-1))!}\frac{(K-(X-1))!}{(K-(X-1)-((m-1)-(i-1)))!}\right] \tag{129}$$

$$= \frac{1}{\binom{K+1}{m}i!(m-i)!}\mathbb{E}_{X_{(K)}}\left[\frac{X!}{(X-(i-1))!}\frac{(K-X)!}{(K-X-((m-1)-(i-1)))!}\right] \tag{130}$$

(By Lemma D.6)

$$= \frac{(i-1)!(m-i)!}{\binom{K+1}{m}i!(m-i)!}\mathbb{E}_{X_{(K)}}\left[\binom{X}{i-1}\binom{K-X}{(m-1)-(i-1)}\right] \tag{131}$$

$$= \frac{(i-1)!}{\binom{K+1}{m}i!}(K+1)p\binom{K}{m-1}\binom{m-1}{i-1}p^{i-1}(1-p)^{m-i} \tag{132}$$

(By Induction Hypothesis)

$$= \frac{m!(K+1-m)!}{(K+1)!i}\frac{K!}{(m-1)!(K-m+1)!}\frac{(m-1)!}{(i-1)!(m-i)!}(K+1)p^i(1-p)^{m-i} \tag{133}$$

$$= \frac{m!}{i!(m-i)!}p^i(1-p)^{m-i} = \binom{m}{i}p^i(1-p)^{m-i} \tag{134}$$

Now we consider the edge cases when $0 = i \leq m$.

If $i = 0$ and $m = 0$,

$$\frac{1}{\binom{K+1}{0}}\mathbb{E}_{X_{(K+1)}}\left[\binom{X}{0}\binom{K+1-X}{0}\right] = 1 \cdot \mathbb{E}_{X_{(K+1)}}[1] = 1 = \binom{0}{0}p^0(1-p)^0 \tag{135}$$

If $i = 0$ and $m > 0$,

$$\frac{1}{\binom{K+1}{m}}\mathbb{E}_{X_{(K+1)}}\left[\binom{K+1-X}{m}\right] \tag{136}$$

$$= \frac{1}{\binom{K+1}{m}}\sum_{x=0}^{K+1}\binom{K+1-x}{m}\binom{K+1}{x}p^x(1-p)^{K+1-x} \tag{137}$$

$$= \frac{1}{\binom{K+1}{m}}\sum_{x=0}^{K+1}\binom{K+1-x}{m}\left(\binom{K}{x}+\binom{K}{x-1}\mathbb{I}\{x \geq 1\}\right)p^x(1-p)^{K+1-x} \tag{138}$$

$$= \frac{1}{\binom{K+1}{m}} \sum_{x=0}^{K} \binom{K+1-x}{m}\binom{K}{x}p^x(1-p)^{K+1-x} + \frac{1}{\binom{K+1}{m}} \sum_{x=1}^{K+1} \binom{K+1-x}{m}\binom{K}{x-1}p^x(1-p)^{K+1-x} \tag{139}$$

(Since when $x = K+1$ and $m > 0$, $\binom{K+1-x}{m} = 0$)

$$= \frac{1}{\binom{K+1}{m}}\left(\sum_{x=0}^{K}\binom{K-x}{m}\binom{K}{x}p^x(1-p)^{K+1-x} + \sum_{x=0}^{K}\binom{K-x}{m-1}\binom{K}{x}p^x(1-p)^{K+1-x}\right) \tag{140}$$

$$+ \frac{1}{\binom{K+1}{m}}\sum_{x=0}^{K}\binom{K-x}{m}\binom{K}{x}p^{x+1}(1-p)^{K-x}$$

(Since $\binom{K+1-x}{m} = \binom{K-x}{m} + \binom{K-x}{m-1}$)

$$= \frac{1}{\binom{K+1}{m}}\left((1-p)\mathbb{E}_{X_{(K)}}[\binom{K-X}{m}] + (1-p)\mathbb{E}_{X_{(k)}}[\binom{K-X}{m-1}]\right) + \frac{1}{\binom{K+1}{m}}p\mathbb{E}_{X_{(K)}}[\binom{K-X}{m}] \tag{141}$$

$$= \frac{1}{\binom{K+1}{m}}\left(\mathbb{E}_{X_{(K)}}[\binom{K-X}{m}] + (1-p)\mathbb{E}_{X_{(K)}}[\binom{K-X}{m-1}]\right) \tag{142}$$

$$= \frac{1}{\binom{K+1}{m}}\left(\binom{K}{m}(1-p)^m + (1-p)\binom{K}{m-1}(1-p)^{m-1}\right) \tag{143}$$

(By Induction Hypothesis) $\tag{144}$

$$= \frac{1}{\binom{K+1}{m}}\binom{K+1}{m}(1-p)^m \tag{145}$$

$$= (1-p)^m \tag{146}$$

$\square$

Based on Lemma D.6 and Lemma D.7 and using the sufficient conditions in Lemma D.5, we are now ready to present the privacy amplification results under a small privacy allowance $m$ as follows.

**Lemma D.8** (Privacy amplification under small privacy allowance $m \leq \frac{K-1}{2}$). *Consider using DaRRM to solve Problem 1.1 with $p_i = p$, $p'_i = p'$, $\forall i \in [K]$ and $\delta = \Delta = 0$. If the privacy allowance is $m \leq \frac{K-1}{2}$, one can set $\gamma(l) = \begin{cases} 1 - 2h(l) & \forall l \in \{0, 1, \dots, \frac{K-1}{2}\} \\ 2h(l) - 1 & \forall l \in \{\frac{K+1}{2}, \dots, K\} \end{cases}$, where $h : \{0, 1, \dots, K\} \to [0, 1]$ and $h(l) = \sum_{i=m}^{2m-1} \frac{\binom{l}{i}\binom{K-l}{2m-1-i}}{\binom{K}{2m-1}}$, and Algorithm DaRRM$_\gamma$ is $m\epsilon$-differentially private.*

*Proof of Lemma D.8.* Let $X_{(K-1)} \sim \text{Binom}(K-1, p)$ and $Y_{(K-1)} \sim \text{Binom}(K-1, e^\epsilon p)$.

$$\mathbb{E}_{X_{(K-1)}}[h(X+1)] = \frac{1}{\binom{K}{2m-1}}\sum_{i=m}^{2m-1}\mathbb{E}_{X_{(K-1)}}[\binom{X+1}{i}\binom{K-X-1}{2m-1-i}] \tag{147}$$

$$= \frac{1}{\binom{K}{2m-1}}\sum_{i=m}^{2m-1}\mathbb{E}_{X_{(K-1)}}[\binom{X}{i}\binom{K-X-1}{2m-1-i} + \binom{X}{i-1}\binom{K-X-1}{2m-1-i}] \tag{148}$$

(Since $\binom{X+1}{i} = \binom{X}{i} + \binom{X}{i-1}\mathbb{I}\{i \geq 1\}$)

$$= \frac{1}{\binom{K}{2m-1}} \sum_{i=m}^{2m-1} \left( \mathbb{E}_{X_{(K-1)}}\left[ \binom{X}{i} \binom{K-1-X}{2m-1-i} \right] + \mathbb{E}_{X_{(K-1)}}\left[ \binom{X}{i-1} \binom{K-1-X}{(2m-2)-(i-1)} \right] \right)$$
(149)

$$= \frac{1}{\binom{K}{2m-1}} \sum_{i=m}^{2m-1} \left( \binom{K-1}{2m-1} \binom{2m-1}{i} p^i (1-p)^{2m-1-i} \right.$$
(150)
$$\left. + \binom{K-1}{2m-2} \binom{2m-2}{i-1} p^{i-1} (1-p)^{2m-1-i} \right)$$
(By Lemma D.7)

$$\mathbb{E}_{X_{(K-1)}}[h(X)] = \frac{1}{\binom{K}{2m-1}} \sum_{i=m}^{2m-1} \mathbb{E}_{X_{(K-1)}}\left[ \binom{X}{i} \binom{K-X}{2m-1-i} \right]$$
(151)

$$\left( \text{Since } \binom{K-X}{2m-1-i} = \binom{K-1-X}{2m-1-i} + \binom{K-1-X}{2m-2-i} \right)$$

$$= \frac{1}{\binom{K}{2m-1}} \sum_{i=m}^{2m-1} \left( \mathbb{E}_{X_{(K-1)}}\left[ \binom{X}{i} \binom{K-1-X}{2m-1-i} \right] + \mathbb{E}_{X_{(K-1)}}\left[ \binom{X}{i} \binom{K-1-X}{2m-2-i} \right] \mathbb{I}\{i \le 2m-2\} \right)$$
(152)

$$= \frac{1}{\binom{K}{2m-1}} \sum_{i=m}^{2m-1} \left( \binom{K-1}{2m-1} \binom{2m-1}{i} p^i (1-p)^{2m-1-i} \right.$$
(153)
$$\left. + \binom{K-1}{2m-2} \binom{2m-2}{i} p^i (1-p)^{2m-2-i} \mathbb{I}\{i \le 2m-2\} \right)$$
(By Lemma D.7)

Hence,

$$\mathbb{E}_{X_{(K-1)}}[h(X+1) - h(X)]$$
(154)

$$= \frac{1}{\binom{K}{2m-1}} \left( \sum_{i=m}^{2m-1} \binom{K-1}{2m-2} \binom{2m-2}{i-1} p^{i-1} (1-p)^{2m-1-i} - \sum_{i=m}^{2m-2} \binom{K-1}{2m-2} \binom{2m-2}{i} p^i (1-p)^{2m-2-i} \right)$$
(155)

$$= \frac{1}{\binom{K}{2m-1}} \left( \sum_{i=m-1}^{2m-2} \binom{K-1}{2m-2} \binom{2m-2}{i} p^i (1-p)^{2m-2-i} - \sum_{i=m}^{2m-2} \binom{K-1}{2m-2} \binom{2m-2}{i} p^i (1-p)^{2m-2-i} \right)$$
(156)

$$= \frac{2m-1}{K} \binom{2m-2}{m-1} p^{m-1} (1-p)^{m-1}$$
(157)

Similarly,

$$\mathbb{E}_{Y_{(K-1)}}[h(Y+1) - h(Y)] = \frac{2m-1}{K} \binom{2m-2}{m-1} (e^\epsilon p)^{m-1} (1 - e^\epsilon p)^{m-1}$$
(158)

Since $p(1-p) \ge e^{-\epsilon} e^\epsilon p(1 - e^\epsilon p)$ for $p \in [0, \frac{1}{1+e^\epsilon}]$,

$$e^{(m-1)\epsilon} \mathbb{E}_{X_{(K-1)}}[h(X+1) - h(X)] = \frac{2m-1}{K} \binom{2m-2}{m-1} e^{(m-1)\epsilon} p^{m-1} (1-p)^{m-1}$$
(159)

$$\ge \frac{2m-1}{K} \binom{2m-2}{m-1} e^{(m-1)\epsilon} (e^{-\epsilon} e^\epsilon p(1 - e^\epsilon p))^{m-1}$$
(160)

$$= \frac{2m-1}{K} \binom{2m-2}{m-1} (e^\epsilon p)^{m-1} (1 - e^\epsilon p)^{m-1}$$
(161)

$$= \mathbb{E}_{Y_{(K-1)}}[h(Y+1) - h(Y)] \tag{162}$$

and so

$$e^{m\epsilon}\mathbb{E}_{X_{(K-1)}}[h(X+1) - h(X)] \geq e^{\epsilon}\mathbb{E}_{Y_{(K-1)}}[h(Y+1) - h(Y)] \tag{163}$$

The above shows $\gamma(l) = \begin{cases} 1 - 2h(l) & l \in \{0, 1, \dots, \frac{K-1}{2}\} \\ 2h(l) - 1 & l \in \{\frac{K+1}{2}, \dots, K\} \end{cases}$, where $h = \sum_{i=m}^{2m-1} \frac{\binom{l}{i}\binom{K-1}{2m-1-i}}{\binom{K}{2m-1}}$, satisfies the first condition in Eq. 110 of Lemma D.5. To ensure the Algorithm is $m\epsilon$ differentially private, we next show this $\gamma$ also satisfies the second condition in Eq. 111 of Lemma D.5.

Let $\hat{X}_{(K-1)} \sim \mathrm{Binom}(K-1, 1-e^{\epsilon}(1-p))$ and $\hat{Y}_{(K-1)} \sim \mathrm{Binom}(K-1, p)$. By Eq. 149, we know

$$\mathbb{E}_{\hat{X}_{(K-1)}}[h(X+1)] = \frac{1}{\binom{K}{2m-1}}\sum_{i=m}^{2m-1}\left(\mathbb{E}_{\hat{X}_{(K-1)}}[\binom{\hat{X}}{i}\binom{K-1-\hat{X}}{2m-1-i}] + \mathbb{E}_{\hat{X}_{(K-1)}}[\binom{\hat{X}}{i-1}\binom{K-1-\hat{X}}{(2m-2)-(i-1)}]\right) \tag{164}$$

$$= \frac{1}{\binom{K}{2m-1}}\sum_{i=m}^{2m-1}\left(\binom{K-1}{2m-1}\binom{2m-1}{i}(1-e^{\epsilon}(1-p))^i(e^{\epsilon}(1-p))^{2m-1-i}\right. \tag{165}$$

$$\left. + \binom{K-1}{2m-2}\binom{2m-2}{i-1}(1-e^{\epsilon}(1-p))^{i-1}(e^{\epsilon}(1-p))^{2m-1-i}\right)$$

By Lemma D.7

and by Eq. 152, we know

$$\mathbb{E}_{\hat{X}_{(K-1)}}[h(\hat{X})] = \frac{1}{\binom{K}{2m-1}}\sum_{i=m}^{2m-1}\left(\mathbb{E}_{\hat{X}_{(K-1)}}[\binom{\hat{X}}{i}\binom{K-1-\hat{X}}{2m-1-i}] + \mathbb{E}_{\hat{X}_{(K-1)}}[\binom{\hat{X}}{i}\binom{K-1-\hat{X}}{2m-2-i}]\mathbb{I}\{i \leq 2m-2\}\right) \tag{166}$$

$$= \frac{1}{\binom{K}{2m-1}}\sum_{i=m}^{2m-1}\left(\binom{K-1}{2m-1}\binom{2m-1}{i}(1-e^{\epsilon}(1-p))^i(e^{\epsilon}(1-p))^{2m-1-i}\right. \tag{167}$$

$$\left. + \binom{K-1}{2m-2}\binom{2m-2}{i}(1-e^{\epsilon}(1-p))^i(e^{\epsilon}(1-p))^{2m-2-i}\mathbb{I}\{i \leq 2m-2\}\right)$$

By Lemma D.7

Hence,

$$\mathbb{E}_{\hat{X}_{(K-1)}}[h(\hat{X}+1) - h(\hat{X})] \tag{168}$$

$$= \frac{1}{\binom{K}{2m-1}}\left(\sum_{i=m}^{2m-1}\binom{K-1}{2m-2}\binom{2m-2}{i-1}(1-e^{\epsilon}(1-p))^{i-1}(e^{\epsilon}(1-p))^{2m-1-i}\right. \tag{169}$$

$$\left. - \sum_{i=m}^{2m-2}\binom{K-1}{2m-2}\binom{2m-2}{i}(1-e^{\epsilon}(1-p))^i(e^{\epsilon}(1-p))^{2m-2-i}\right)$$

$$= \frac{1}{\binom{K}{2m-1}}\left(\sum_{i=m-1}^{2m-2}\binom{K-1}{2m-2}\binom{2m-2}{i}(1-e^{\epsilon}(1-p))^i(e^{\epsilon}(1-p))^{2m-2-i}\right. \tag{170}$$

$$\left. - \sum_{i=m}^{2m-2}\binom{K-1}{2m-2}\binom{2m-2}{i}(1-e^{\epsilon}(1-p))^i(e^{\epsilon}(1-p))^{2m-2-i}\right)$$

$$= \frac{2m-1}{K}\binom{2m-2}{m-1}(1-e^{\epsilon}(1-p))^{m-1}(e^{\epsilon}(1-p))^{m-1} \tag{171}$$

Similarly,

$$\mathbb{E}_{\hat{Y}_{(K-1)}}[h(\hat{Y}+1) - h(\hat{Y})] = \frac{2m-1}{K}\binom{2m-2}{m-1}p^{m-1}(1-p)^{m-1} \tag{172}$$

Hence,

$$e^{(m+1)\epsilon}\mathbb{E}_{\hat{X}_{(K-1)}}[h(\hat{X}+1)-h(\hat{X})] = e^{(m+1)\epsilon}\frac{2m-1}{K}\binom{2m-2}{m-1}(1-e^{\epsilon}(1-p))^{m-1}(e^{\epsilon}(1-p))^{m-1}$$
(173)

$$\geq \frac{2m-1}{K}\binom{2m-2}{m-1}(1-e^{\epsilon}(1-p))^{m-1}e^{(m-1)\epsilon}(1-p)^{m-1}$$
(174)

$$= \frac{2m-1}{K}\binom{2m-2}{m-1}(e^{\epsilon}-e^{2\epsilon}(1-p))^{m-1}(1-p)^{m-1}$$
(175)

Note that

$$e^{\epsilon}-e^{2\epsilon}(1-p) = e^{\epsilon}-e^{2\epsilon}+e^{2\epsilon}p \geq p$$
(176)

$$\Leftrightarrow (e^{\epsilon}+1)(e^{\epsilon}-1)p \geq e^{\epsilon}(e^{\epsilon}-1)$$
(177)

$$\Leftrightarrow p \geq \frac{e^{\epsilon}}{e^{\epsilon}+1} = \frac{1}{1+e^{-\epsilon}}$$
(178)

and the second condition in Eq. 111 of Lemma D.5 is on $p \in [\frac{1}{1+e^{-\epsilon}}, 1]$.

Therefore, following Eq. 175,

$$e^{(m+1)\epsilon}\mathbb{E}_{\hat{X}_{(K-1)}}[h(\hat{X}+1)-h(\hat{X})] \geq \frac{2m-1}{K}\binom{2m-2}{m-1}p^{m-1}(1-p)^{m-1}$$
(179)

$$= \mathbb{E}_{\hat{Y}_{(K-1)}}[h(\hat{Y}+1)-h(\hat{Y})]$$
(180)

which means the second condition in Eq. 111 of Lemma D.5 is also satisfied.

Therefore, by Lemma D.5, DaRRM$_\gamma$ with this specific choice of $\gamma$ is $m\epsilon$-differentially private. $\square$

Now, Theorem 4.1 follows from combining Lemma D.4 and Lemma D.8.

# E DETAILS OF OPTIMIZING $\gamma$ IN DaRRM

## E.1 DERIVING THE OPTIMIZATION OBJECTIVE

For $\gamma$ that is symmetric around $\frac{K}{2}$, we can write the objective as

$$\mathbb{E}_{p_1,p_2,\ldots,p_K \sim \mathcal{T}}[\mathcal{E}(\text{DaRRM}_\gamma)] \tag{181}$$

$$= \mathbb{E}_{p_1,p_2,\ldots,p_K \sim \mathcal{T}}[\mathcal{D}_{TV}(\text{DaRRM}_\gamma(\mathcal{S}) \parallel f(\mathcal{S}))] \tag{182}$$

$$= \mathbb{E}_{p_1,p_2,\ldots,p_K \sim \mathcal{T}}[|\Pr[\text{DaRRM}_\gamma(\mathcal{S}) = 1] - \Pr[f(\mathcal{S}) = 1]|] \tag{183}$$

$$= \mathbb{E}_{p_1,p_2,\ldots,p_K \sim \mathcal{T}}\left[\left| \sum_{l=\frac{K+1}{2}}^{K} \left(\alpha_l \cdot (\gamma(l) + \frac{1}{2}(1-\gamma(l))) - \alpha_l\right) + \sum_{l=0}^{\frac{K-1}{2}} \alpha_l \cdot \frac{1}{2}(1-\gamma(l))\right|\right] \tag{184}$$

$$= \mathbb{E}_{p_1,p_2,\ldots,p_K \sim \mathcal{T}}\left[\left| \sum_{l=0}^{\frac{K-1}{2}} \alpha_l(\frac{1}{2}\gamma(l) - \frac{1}{2}) + \sum_{l=\frac{K+1}{2}}^{K} \alpha_l(\frac{1}{2} - \frac{1}{2}\gamma(l))\right|\right] \tag{185}$$

The above follows by conditioning on $\mathcal{L} = \{0, 1, \ldots, K\}$, i.e. the sum of observed outcomes in $\mathcal{S}$

$$= \mathbb{E}_{p_1,p_2,\ldots,p_K \sim \mathcal{T}}\left[\left|\frac{1}{2} \sum_{l=\frac{K+1}{2}}^{K} (\alpha_l - \alpha_{K-l})(1-\gamma(l))\right|\right] \tag{186}$$

The above follows by symmetry of $\gamma$

Furthermore, notice the objective is symmetric around 0, and can be written as

$$\mathbb{E}_{p_1,p_2,\ldots,p_K \sim \mathcal{T}}\left[\frac{1}{2} \sum_{l=\frac{K+1}{2}}^{K} (\alpha_l - \alpha_{K-l})(1-\gamma(l))\right] \tag{187}$$

$$= \frac{1}{2}\mathbb{E}_{p_1,p_2,\ldots,p_K \sim \mathcal{T}}\left[\sum_{l=\frac{K+1}{2}}^{K} \left((\alpha_l - \alpha_{K-l}) - (\alpha_l - \alpha_{K-l})\gamma(l)\right)\right] \tag{188}$$

$$= \frac{1}{2}\mathbb{E}_{p_1,p_2,\ldots,p_K \sim \mathcal{T}}\left[\sum_{l=\frac{K+1}{2}}^{K} (\alpha_l - \alpha_{K-l})\right] - \frac{1}{2}\mathbb{E}_{p_1,p_2,\ldots,p_K \sim \mathcal{T}}\left[\sum_{l=\frac{K+1}{2}}^{K} (\alpha_l - \alpha_{K-l})\gamma(l)\right] \tag{189}$$

and this is the same as optimizing

$$-\frac{1}{2}\mathbb{E}_{p_1,p_2,\ldots,p_K \sim \mathcal{T}}\left[\sum_{l=\frac{K+1}{2}}^{K} (\alpha_l - \alpha_{K-l})\gamma(l)\right] = -\frac{1}{2} \sum_{l=\frac{K+1}{2}}^{K} \mathbb{E}_{p_1,p_2,\ldots,p_K \sim \mathcal{T}}[(\alpha_l - \alpha_{K-l})]\gamma(l) \tag{190}$$

which is linear in $\gamma$.

## E.2 PRACTICAL APPROXIMATION OF THE OBJECTIVE

Since the optimization objective in Eq. 190 requires taking an expectation over $p_1, \ldots, p_K$, and this invovles integrating over $K$ variables, which can be slow in practice, we propose the following approximation to efficiently compute the objective. We start with a simple idea to compute the objective, by sampling $p_i$'s from $[0, 1]$ and take an empirical average of the objective value over all subsampled sets of $p_1, \ldots, p_K$ as the approximation of the expectation in Section E.2.1. However, we found this approach is less numerically stable. We then propose the second approach to approximate the objective in Section E.2.2, which approximates the integration over $p_i$'s instead of directly approximating the objective value. We use the second approximation approach in our experiments and empirically demonstrates its effectiveness. Note approximation the optimization objective has no affect on the privacy guarantee.

### E.2.1 APPROXIMATION VIA DIRECT SAMPLING OF $p_i$'S

We start with a straightforward way of approximating the objective:

1. Step 1: Sample $p_1, p_2, \ldots, p_K \sim \mathcal{T}$
2. Step 2: Compute the sampled objective value $g = -\frac{1}{2}\sum_{l=\frac{K+1}{2}}^{K}(\alpha_l - \alpha_{K-l})\gamma(l))$ based on the sampled $p_i$'s.
3. Repeat Step 1 and Step 2 for $T = 10000$ times. Let $g_t$ denotes the objective value in $t$-th trial. Use $\frac{1}{T}\sum_{t=1}^{T} g_t$ as an unbiased estimation of the true objective.

However, we found this approximation is less numerically stable in the experiments and so we propose and adpot the second approach as follows.

### E.2.2 APPROXIMATING THE INTEGRATION OVER $p_i$'S

Consider the following surrogate objective:

$$-\frac{1}{2}\sum_{l=\frac{K+1}{2}}^{K}\int_{0.5}^{1}\int_{0.5}^{1}\cdots\int_{0.5}^{1}(\alpha_l - \alpha_{K-l})dp_1 dp_2 \ldots dp_K \cdot \gamma(l) \tag{191}$$

where we approximate the integration instead of directly approximating the objective value. The approximation of the integration is based on the rectangular rule and that the Poison Binomial (PB) distribution is invariant to the order of its probability parameters.

First, we discretize the integration over $p_i$'s: pick $\tau = 50$ points representing probabilities between $[0.5, 1)$ with equal distance $\theta$. Denote this set of points as $\mathcal{W}$. We pick only $\tau = 50$ samples to ensure the distance between each sample, i.e., $\theta$, is not too small; or this can cause numerical instability. For each $l \in \{\frac{K+1}{2}, \frac{K+1}{2}+1, \ldots, K\}$, we want to compute an approximated coefficient for $\gamma(l)$ as follows:

$$\int_{0.5}^{1}\int_{0.5}^{1}\cdots\int_{0.5}^{1}(\alpha_l - \alpha_{K-l})dp_1 dp_2 \ldots dp_K \approx \sum_{p_1 \in \mathcal{W}}\sum_{p_2 \in \mathcal{W}}\cdots\sum_{p_K \in \mathcal{W}}(\alpha_l - \alpha_{K-l}) \tag{192}$$

which approximates integration over a $K$-dimensional grid $\mathcal{W}^K$.

The idea is then to sample points from this $K$-dimensional grid $\mathcal{W}^K$ and compute an empirical mean of the integration based on the sample probabilities for $p_1, \ldots, p_K$ from $\mathcal{W}^K$ as the approximation of the integration in the objective.

Let $(s_1, s_2, \ldots, s_K)$ be randomly sampled probability values from $\mathcal{W}^K$ and we want to compute $(\alpha_l - \alpha_{K-l})$ for all $l$ based on $(p_1, \ldots, p_K) = (s_1, \ldots, s_K)$. To apply the rectangular rule, since the grid of probabilities is $K$-dimensional, the weight of $(\alpha_l - \alpha_{K-l})$ in the approximate integration is $\theta^K$. Furthermore, observe that $\alpha_l$ is the pmf at $l$ from a Poison Binomial (PB) distribution in our case, and $\text{PB}(p_1, \ldots, p_K) \sim \text{PB}(\pi(p_1, \ldots, p_K))$, where $\pi$ denotes a permutation of $p_1, \ldots, p_K$ and $\sim$ denotes "the same distribution". Hence, with a single probability sample $(s_1, \ldots, s_K)$, we can indeed compute $\alpha_l - \alpha_{K-l}$ for each $l$ at $K!$ points from the grid $\mathcal{W}^K$, since they all have the same value. Therefore, we should set the weight of $\alpha_l - \alpha_{K-l}$ in the approximate integration as $w = \theta^K \cdot K!$. Furthermore, since the order of $(p_1, \ldots, p_K)$ does not affect the objective value, there is a total of ($\tau$ choose $K$ with replacement) $= \binom{\tau+K-1}{K} := P$ different points in the grid $\mathcal{W}^K$.

In summary, our approximation of the integration proceeds as follows: let $w = \theta^K \cdot K!$ and $P = \binom{\tau+K-1}{K}$.

1. Step 1: Generate a set $\mathcal{W}$ with 50 values of equal distance between 0.5 and 1.
2. Step 2: Randomly sample $(s_1, s_2, \ldots, s_K) \sim \mathcal{W}^K$. Compute $w \cdot (\alpha_l - \alpha_{K-l})$ based on $(p_1, p_2, \ldots, p_K) = (s_1, s_2, \ldots, s_K)$.
3. Step 3: repeat Step 2 for $N = 10000$ times.
4. Step 4: Let $g_t = \sum_{l=\frac{K+1}{2}}^{K} w \cdot (\alpha_l - \alpha_{K-l})$ denotes the approximate integration value in $t$-th trial.
   Form an unbiased estimation of the integration as $\frac{P}{N}\sum_{t=1}^{N} g_t$.

### E.3 Reducing # Constraints From ∞ to A Polynomial Set

**Lemma 5.1.** *Consider using DaRRM to solve Problem 1.1. Given an arbitrary $\gamma$, let the global worst case probabilities be $(p_1^*, \ldots, p_K^*, p_1'^*, \ldots, p_K'^*) = \arg\max_{\{(p_i, p_i')\}_{i=1}^K} f(p_1, \ldots, p_K, p_1', \ldots, p_K'; \gamma)$, where $f$ is the privacy cost objective defined in Lemma 3.3. Each pair $(p_i^*, p_i'^*)$ satisfies $(p_i^*, p_i'^*) \in \{(0,0), (1,1), (0, \Delta), (\Delta, 0), (1 - \Delta, 1), (1, 1 - \Delta), (\frac{e^\epsilon + \Delta}{e^\epsilon + 1}, \frac{1 - \Delta}{e^\epsilon + 1}), (\frac{1 - \Delta}{e^\epsilon + 1}, \frac{e^\epsilon + \Delta}{e^\epsilon + 1})\}$, $\forall i \in [K]$. Furthermore, there exists a set $\mathcal{P}$ of size $O(K^7)$ such that $(p_1^*, \ldots, p_K^*, p_1'^*, \ldots, p_K'^*) = \arg\max_{\{(p_i, p_i')\}_{i=1}^K \in \mathcal{P}} f(p_1, \ldots, p_K, p_1', \ldots, p_K'; \gamma)$ if $\delta > 0$ and a set $\mathcal{P}$ of size $O(K^3)$ if $\delta = 0$.*

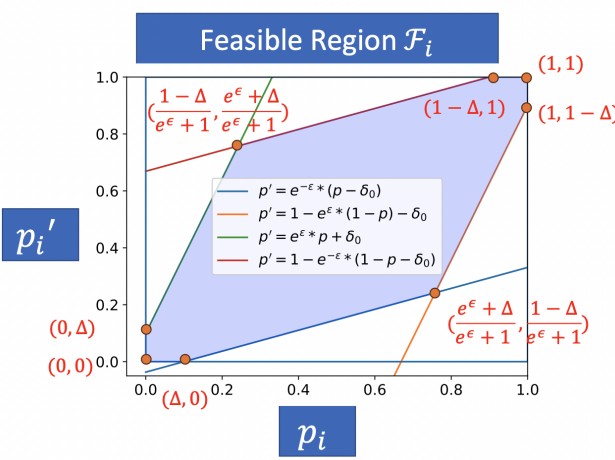

Figure 6: An illustration of the feasible region $\mathcal{F}_i$.

*Proof.* **Part I: Reducing # privacy constraints from ∞ to exponential.** Consider $(p_i, p_i')$ for an arbitrary $i \in [K]$ and fixing $(p_j, p_j'), \forall j \neq i$. The privacy cost objective $f(p_1, \ldots, p_K, p_1', \ldots, p_K'; \gamma)$, as defined in Lemma 3.3, is then linear in $(p_i, p_i')$. To ensure DaRRM$_\gamma$ is differentially private with a target privacy loss $m\epsilon$, we need to consider the worst case probabilities $(p_i^*, p_i'^*) = \arg\max_{(p_i, p_i')} f$, given $(p_j, p_j'), \forall j \neq i$. Since mechanism $M_i$ is $(\epsilon, \Delta)$-differentially private, by definition, the following constraints on $(p_i, p_i')$ apply simultaneously,

$$p_i \leq e^\epsilon p_i' + \Delta, \quad p_i' \leq e^\epsilon p + \Delta$$
$$1 - p_i \leq e^\epsilon (1 - p_i') + \Delta, \quad 1 - p_i' \leq e^\epsilon (1 - p_i) + \Delta$$

This implies $(p_i, p_i')$ lies in a feasible region $\mathcal{F}_i$ (see Figure 6). Notice the constraints on $(p_i, p_i')$, that is, the boundaries of $\mathcal{F}_i$, are linear in $p_i$ and $p_i'$, $\max_{(p_i, p_i')} f(p_1, \ldots, p_K, p_1', \ldots, p_K'; \gamma)$ is hence a Linear Programming (LP) problem in $(p_i, p_i')$ for $i \in [K]$. Hence, the $(p_i^*, p_i'^*)$ has to be on one of the eight corners of $\mathcal{F}_i$ — that is $(p_i^*, p_i'^*) \in \{(0,0), (1,1), (0, \Delta), (\Delta, 0), (1 - \Delta, 1), (1, 1 - \Delta), (\frac{e^\epsilon + \Delta}{e^\epsilon + 1}, \frac{1 - \Delta}{e^\epsilon + 1}), (\frac{1 - \Delta}{e^\epsilon + 1}, \frac{e^\epsilon + \Delta}{e^\epsilon + 1})\} := \mathcal{C}$. Therefore, the infinitely many privacy constraints are now reduced to only $8^K$ in optimizing for the best $\gamma$ function in DaRRM.

**Part II: Reducing # privacy constraints from exponential to polynomial.** To further reduce the number of privacy constraints in optimization, recall by Lemma 3.3 we need $\gamma$ such that

$$f(p_1, \ldots, p_K, p_1', \ldots, p_K'; \gamma) = \sum_{l=0}^{\frac{K-1}{2}} (e^{m\epsilon} \alpha_l' - \alpha_l) \cdot \gamma(l) + \sum_{l=\frac{K+1}{2}}^{K} (\alpha_l - e^{m\epsilon} \alpha_l') \cdot \gamma(l) \leq e^{m\epsilon} - 1 + 2\delta \tag{193}$$

where $\alpha_l = \Pr[\mathcal{L} = \sum_{i=1}^K M_i(\mathcal{D}) = l]$ and $\alpha_l' = \Pr[\mathcal{L}' = \sum_{i=1}^K M_i(\mathcal{D}') = l]$. Note $\mathcal{L}$ follows a Poisson Binomial (PB) distribution parameterized by $p_1, \ldots, p_K$, and $\mathcal{L}'$ follows a PB distribution

parameterized by $p'_1, \ldots, p'_K$. Observe that PB distribution[9] is invariant under the permutation of parameters. That is, $\text{PB}(p_1, \ldots, p_K)$ has the same distribution as $\text{PB}(\pi(p_1, \ldots, p_K))$, where $\pi$ denotes permutation; and similarly, $\text{PB}(p'_1, \ldots, p'_K)$ has the same distribution as $\text{PB}(\pi(p'_1, \ldots, p'_K))$.

Consider a set $\mathcal{P}$ of privacy constraints as Eq. 193, where each constraint in $\mathcal{P}$ is constructed by setting $(p_1, p'_1), (p_2, p'_2), \ldots, (p_K, p'_K) = (v_1, v_2, \ldots, v_K)$, where $v_i \in \mathcal{C}, \forall i \in [K]$, such that constraints constructed by $(p_1, p'_1), (p_2, p'_2), \ldots, (p_K, p'_K) = \pi(v_1, v_2, \ldots, v_K)$ is not in $\mathcal{P}$ — that is, $\mathcal{P}$ has size (8 chooses K with replacement) = $\binom{K+8-1}{K} = O(K^7)$. Then, the global worst case probabilities $(p^*_1, \ldots, p^*_K, p'^*_1, \ldots, p'^*_K)$ must satisfy one of the constraints in $\mathcal{P}$, i.e. $(p^*_1, \ldots, p^*_K, p'^*_1, \ldots, p'^*_K) = \max_{\{(p_i, p'_i)\}^K_{i=1} \in \mathcal{P}} f(p_1, \ldots, p_K, p'_1, \ldots, p'_K; \gamma)$. This implies we only need $O(K^7)$ privacy constraints in optimizing for the best noise function $\gamma$ in DaRRM.

Note when $\Delta = 0$, i.e., under pure differential privacy setting, the feasible region $\mathcal{F}_i$ has only 4 corners instead of 8, that is, $(p^*_i, p'^*_i) \in \mathcal{C} = \{(0,0), (1,1), (\frac{e^\epsilon}{e^\epsilon+1}, \frac{1}{e^\epsilon+1}), (\frac{1}{e^\epsilon+1}, \frac{e^\epsilon}{e^\epsilon+1})\}$. Hence, when $\Delta = 0$, $\mathcal{P}$ has size (4 choose $K$ with replacement) = $\binom{K+4-1}{K} = O(K^3)$, implying we only need $O(K^3)$ privacy constraints in optimizing for the best noise function $\gamma$. $\qquad\square$

# F   Full Experiment Results

## F.1   Optimal $\gamma$ in Simulations

### F.1.1   Comparison Against Advanced Composition

Advanced composition indiates less privacy loss than simple composition when the number of compositions, $m$, is large, or when the failure probability $\delta$ is large. To enable meaningful comparison against advanced composition, we consider a larger $K$ and a larger failure probability.

Consider $K = 35, \epsilon = 0.1, \Delta = 10^{-5}$. By advanced composition, if one outputs the majority of $M$ subsampled mechanisms for some $M < K$, the majority output is $(\sqrt{2M\log(1/\delta')}\epsilon + M\epsilon(e^\epsilon - 1), M\Delta + \delta')$-differentially private for some $\delta' > 0$. We set this as the privacy guarantee of all majority ensembling algorithms. That is, if we want the majority output to be $(m\epsilon, \delta)$-differentially private, we set $m = \sqrt{2M\log(1/\delta')} + M(e^\epsilon - 1)$ and $\delta = M\Delta + \delta'$ accordingly. The parameters $\tau$ and $\lambda$ for the constant $\gamma$ in randomized response (see Lemma C.1) are set to be $\tau = \sqrt{2K\log(1/\delta')} + K(e^\epsilon - 1)$ and $\lambda = K\Delta + \delta'$.

In the experiments, we consider $M = \{10, 13, 15, 20\}$ and $\delta' = 0.1$.

All values of the parameters of the private ensembling algorithms are computed and listed in the table:

| # Subsampled mechanisms | $M$ | 10 | 13 | 15 | 20 |
|---|---|---|---|---|---|
| Privacy allowance | $m$ | 7.8378 | 9.1046 | 9.8888 | 11.7005 |
| Parameter of constant $\gamma$ | $\tau$ | 16.3766 | 16.3767 | 16.3767 | 16.3767 |
| Parameter of constant $\gamma$ | $\lambda$ | 0.10035 | 0.10035 | 0.10035 | 0.10035 |
| Overall privacy loss | $m\epsilon$ | 0.7837 | 0.9104 | 0.9889 | 1.1700 |
| Overall failure probability | $\delta$ | 0.10010 | 0.10013 | 0.10015 | 0.1002 |

Table 2: Parameters of all algorithms. Note all the private ensembling algorithms for comparison in the experiment is required to be $(m\epsilon, \delta)$-differentially private. $K = 35, \epsilon = 0.1, \delta = 10^{-5}$ and $\delta' = 0.1$.

---

[9]See, e.g. `https://en.wikipedia.org/wiki/Poisson_binomial_distribution`, for the pmf of Poisson Binomial (PB) distribution.

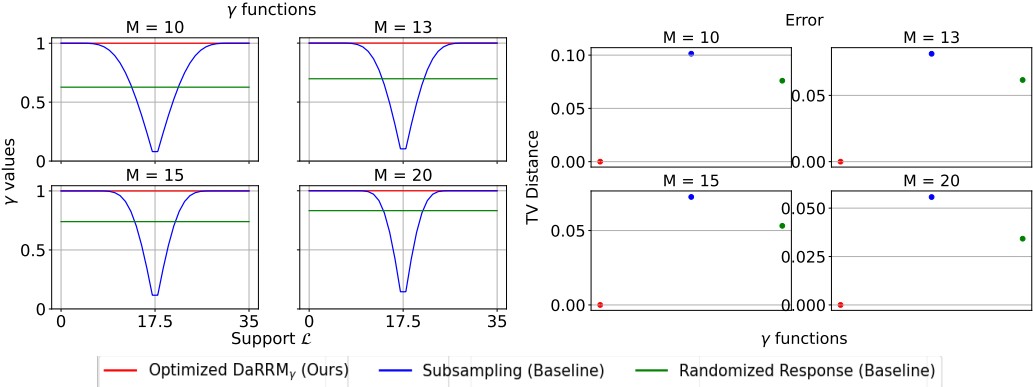

Figure 7: Plots of $\gamma$ functions corresponding to optimized, subsampling (whose privacy guarantee is reasoned through advanced composition), the data-independent $\gamma$ in randomized response and the error in TV distance of the majority ensembling output of DaRRM with different $\gamma$ functions, when $K = 35$, the number of subsamples $M \in \{10, 20, 30, 40\}$, $\Delta = 10^{-5}$, and $\delta' = 0.1$.

### F.1.2 COMPARISON UNDER PURE DP SETTINGS

Consider the pure differential privacy setting, where $\Delta = \delta = 0$. Note in such pure differential privacy case, simple composition is tight. The subsampling baseline here outputs the majority of $m$ out of $K$ subsampeld mechansims (without replacement). The majority output of different ensembling algorithms for comparison is required to be $m\epsilon$-differentially private. Furthermore, since the number of constraints in our optimization framework is $O(K^3)$ under pure differential privacy (see Lemma 5.1), we can optimize DaRRM$_\gamma$ for aggregating a larger number $K$ of mechanisms. In this section, we present the simulation results for $K \in \{11, 101\}$ and compare the utility of three majority ensembling algorithms: optimized DaRRM, subsampling and randomizied response, under the same privacy loss.

**Setting 1.** $K = 11, m \in \{1, 3, 5, 7\}$.

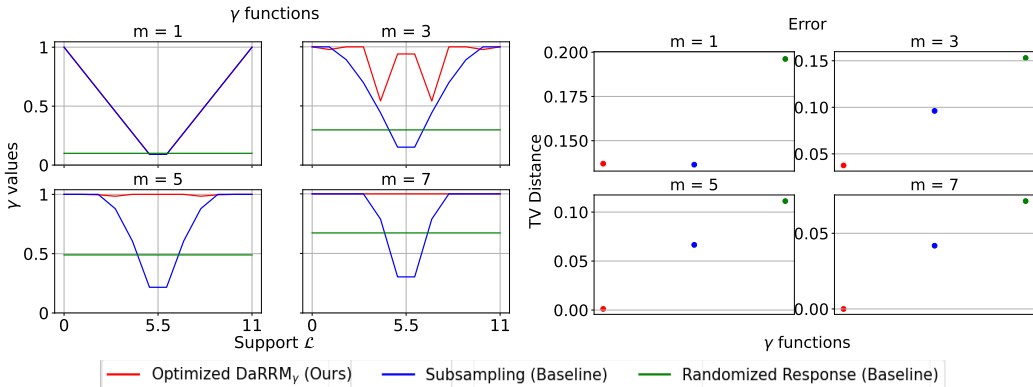

Figure 8: Plots of $\gamma$ functions corresponding to optimized, subsampling, the data-independent $\gamma$ in randomized response and the error in TV distance of the majority ensembling output of DaRRM with different $\gamma$ functions, when $K = 11, m \in \{1, 3, 5, 7\}$, $\Delta = \delta = 0$.

**Setting 2.** $K = 101, m \in \{10, 20, 30, 40\}$.

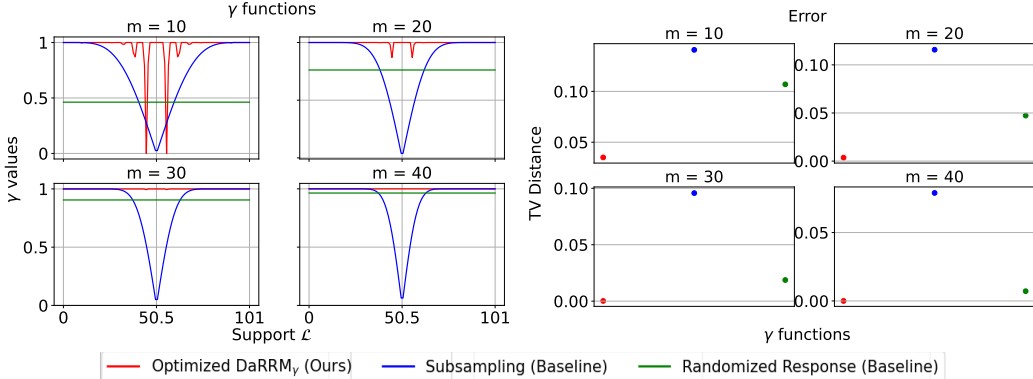

Figure 9: Plots of $\gamma$ functions corresponding to optimized, subsampling, the data-independent $\gamma$ in randomized response and the error in TV distance of the majority ensembling output of DaRRM with different $\gamma$ functions, when $K = 101, m \in \{10, 20, 30, 40\}, \Delta = \delta = 0$.

### F.1.3  COMPARISON UNDER DIFFERENT PRIOR DISTRIBUTIONS OF $p_i$'S

Recall $p_i = \Pr[M_i(\mathcal{D}) = 1], \forall i \in [K]$. We stress that our optimization procedure applies to any prior distribution of $p_i$'s. Let $\mathcal{U}$ denote the distribution Uniform$([0, 1])$. We present results when $p_i \sim \mathcal{U}, \forall i \in [K]$, in the previous sections to show the performance of optimized DaRRM$_\gamma$ in the most general case when we do not have any prior knowledge of the mechanisms $M_i$'s output, i.e., $p_i$. It is possible to consider a different prior distribution $\mathcal{T}$ of $p_i$'s. If the true distribution of $p_i$'s is closer to $\mathcal{T}$ than $\mathcal{U}$, than we get improved utility with the same privacy guarantee by optimizing $\gamma$ under $\mathcal{T}$ than under $\mathcal{U}$; otherwise, if the true distribution of $p_i$ is very different from $\mathcal{T}$, we suffer utility loss.

To illustrate this point, consider the following experiment setting. Suppose our prior belief is that each mechanism $M_i$ has a clear tendency towards voting 0 or 1, i.e., $p_i$ is far from 0.5. Let the new distribution $\mathcal{T}$ be Uniform$([0, 0.3] \cup [0.7, 1])$.

To optimize $\gamma$ under $\mathcal{T}$, we change the approximate objective in Eq. 191, which optimizes $\gamma$ assuming $p_i \sim \mathcal{U}$, to be the following, which optimizes $\gamma$ assuming $p_i \sim \mathcal{T}, \forall i \in [K]$,

$$-\frac{1}{2} \sum_{l=\frac{K+1}{2}}^{K} \int_{0.7}^{1} \int_{0.7}^{1} \cdots \int_{0.7}^{1} (\alpha_l - \alpha_{K-l}) dp_1 dp_2 \ldots dp_K \cdot \gamma(l) \tag{194}$$

**Setting.**  $K = 11, m \in \{3, 5\}, \delta = \Delta = 0$.

We compute the error of the optimized DaRRM$_\gamma$ in three different settings with three different actual $p_i$ distributions:

1. "Actual: Uniform$([0, 1])$", which means we take $p_i \sim \mathcal{U}, \forall i \in [K]$ when computing the error

2. "Actual: Uniform$([0, 0.1])$", which means we take $p_i \sim$ Uniform$([0, 0.1]), \forall i \in [K]$ when computing the error

   This setting implies the mechanisms have a clear majority (of 0)

3. "Actual: $p_i = 0.5$", which means we take $p_i = 0.5, \forall i \in [K]$ when computing the error

   This setting implies the mechanisms have no clear majority

Note since $\mathcal{T}$ is closer to the distribution in the second setting, we would expect DaRRM$_\gamma$ has a lower error when $\gamma$ is optimized under $\mathcal{T}$ than under $\mathcal{U}$ in this setting. Also, since $\mathcal{T}$ is very different from the distribution in the third setting, we would expect DaRRM$_\gamma$ has a lower error when $\gamma$ is optimized under $\mathcal{U}$ than under $\mathcal{T}$.

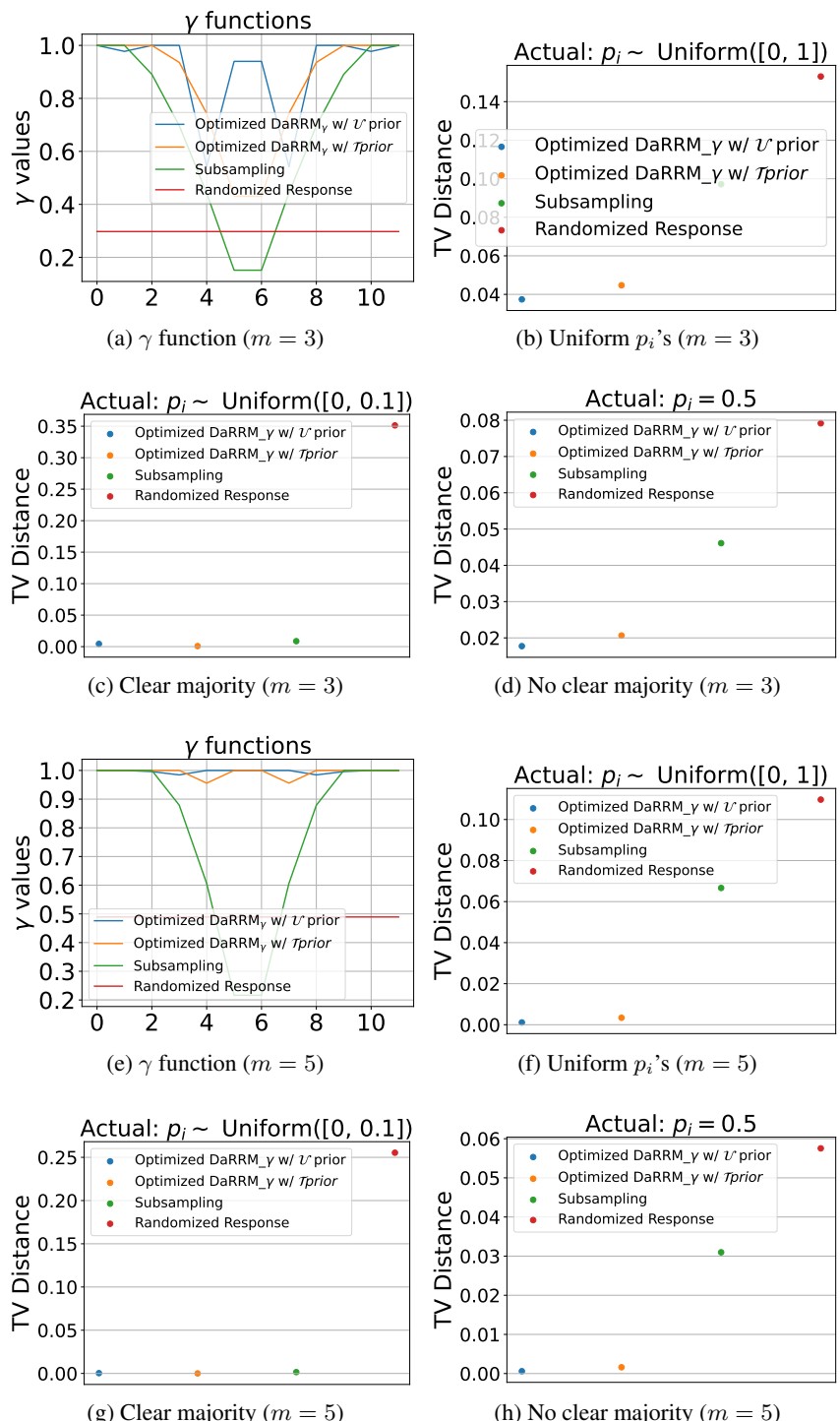

Figure 10: Comparison of the error of DaRRM$_\gamma$ with optimized $\gamma$ under two different prior distributions of $p_i$, i.e., $\mathcal{U}$ and $\mathcal{T}$, in the setting where $m \in \{3, 5\}, K = 11$. Observe that if the prior distribution of $p_i$ we use when optimizing $\gamma$ is closer to the actual distribution, we have additional utility gain (i.e., decreased error); otherwise, we suffer utility loss (i.e., increased error), compared to optimize $\gamma$ under the uniform distribution $\mathcal{U}$ of $p_i$ over $[0, 1]$. Furthermore, regardless of the choice of the prior distribution of $p_i$, optimized DaRRM$_\gamma$ achieves a lower error compared to the two baselines: Subsampling and Randomized Response.

## F.2 PRIVATE DISTRIBUED SIGN-SGD

**Notation.** $w(t)$ denotes the parameter of the model at the $t$-th communication round.

**Additional Experiment Details.** In our experiments, each client computes its gradient based on the entire local dataset at each communication around. Also for simplicity, all clients are participated in training at each round.

---

**Algorithm 2** $\beta$-Stochastic Sign SGD (Algorithm 2 of Xiang & Su (2023b)) without client subsampling

---

1: **Input:** $K$ clients, $T$ communication rounds, batch size $n$, hyperparameters $B, \beta$,
2: **Output:** $w(T)$
3: **Initialization:** $w(0) \leftarrow \mu$ for each $i \in [K]$
4: **for** communication round $t = 1, 2, \ldots, T$ **do**
5:     **Client:**
6:     **for** Client $i \in [K]$ **do**
7:         Each client $i \in [K]$ computes $n$ stochastic gradients $\mathbf{g}_i^1(t), \ldots, \mathbf{g}_i^{(n)}(t)$
8:         **for** coordinate $j = 1, 2, \ldots, d$ **do**
9:             $\hat{g}_{i,j}(t) \leftarrow 1$ with probability $\frac{B + \beta + \text{clip}\{\frac{1}{n}\sum_{i=1}^n \mathbf{g}_{m,j}^{(i)}(t)\}}{2B + 2\beta}$; $\hat{g}(t)_{i,j} \leftarrow -1$ otherwise
10:         **end for**
11:         Report $\hat{\mathbf{g}}_i(t)$ to the server.
12:     **end for**
13:     **Server:**
14:     On receiving one-bit encoded gradients $\hat{\mathbf{g}}_1(t), \ldots, \hat{\mathbf{g}}_K(t)$ from the clients, compute $\tilde{\mathbf{g}}(t) \leftarrow$ **Aggregate**$(\{\hat{\mathbf{g}}_i(t)\}_{i=1}^K)$
15:     Send $\tilde{\mathbf{g}}(t)$ to all clients
16:     Upon receiving $\tilde{\mathbf{g}}(t)$: $w(t+1) \leftarrow w(t) - \eta\tilde{\mathbf{g}}(t)$
17: **end for**

---

Coordinate wise pure DP guarantee:

**Theorem F.1** (Theorem 4 of Xiang & Su (2023b)). *0-StoSign is not differentially private. When $\beta > 0$, $\beta$-StoSign is coordinate-wise $\log(\frac{2B+\beta}{\beta})$-differentially private. That is, $\beta$-StoSign is $d \cdot \log(\frac{2B+\beta}{\beta})$-differentially private.*

## F.3 PRIVATE SEMI-SUPERVISED KNOLWEDGE TRANSFER

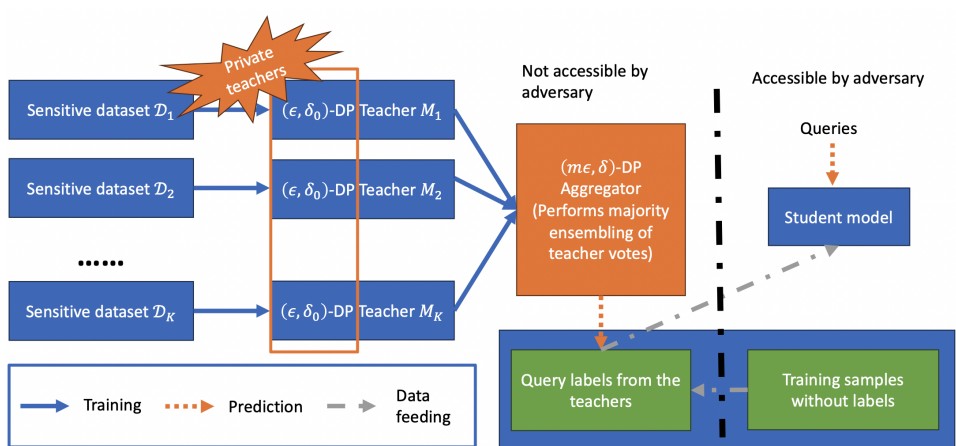

Figure 11: Semi supervised knowledge transfer setting. This figure is adapted from Figure 1 of PATE Papernot et al. (2017). Unlike PATE, we consider an untrustworthy aggregator and aggregate private teachers through private majority ensembling.

**More Details About the Baseline GNMax Papernot et al. (2018)**

The GNMax aggregation mechanism proceeds as follows (Section 4.1 of Papernot et al. (2018)): on input $x$,

$$M_\sigma(x) = \arg\max_i\{n_i(x) + \mathcal{N}(0, \sigma^2)\} \tag{195}$$

where $n_i(x)$ is # teachers who vote for class $i$.

Note GNMax works perfectly in aggregating non-private teachers, in our setting, it does not exploit the fact that each teacher is $(\epsilon, \Delta)$-differentially private. Hence, GNMax can add more noise than necessary to ensure the final aggregated output is $(m\epsilon, \delta)$-differentially private, especially when $\epsilon$ is small.

The privacy analysis Papernot et al. (2018) mainly focuses on computing the overall privacy loss of multiple private majority ensembling queries, while our analysis focuses on the privacy loss of a single-step aggregation from "prviate" teachers. Note the privacy composition analysis in Papernot et al. (2018) also applies to our setting to reason about the privacy loss through multiple queries.

**How to set $\sigma$ in GNMax?**

Section 4.1 of Papernot et al. (2018) states the GNMax mechanism is $(\lambda, \lambda/\sigma^2)$-Renyi differentially private (RDP), for all $\lambda \geq 1$.

Although there is a data-dependent bound for GNMax that is tighter than the above mentioned RDP bound in Section 4.1 and in Appendix A of Papernot et al. (2018), according to Corollary 11 of Papernot et al. (2018), this analysis applies to majority voting when the number of output classes is $\geq 3$, which does not directly apply to our binary-output case. Hence, we use the data-independent RDP bound for GNMax.

The following theorem shows the relationship between RDP and differential privacy (DP):

**Theorem F.2** (RDP to DP (Theorem 5 of Papernot et al. (2018))). *If a mechanism $M$ guarantees $(\lambda, \epsilon)$-RDP, then $M$ guarantees $(\epsilon + \frac{\log 1/\delta}{\lambda - 1}, \delta)$-differential privacy for $\delta \in (0, 1)$.*

Therefore, GNMax with parameter $\sigma^2$ guarantees $(\frac{\lambda}{\sigma^2} + \frac{\log 1/\delta}{\lambda - 1}, \delta)$-differential privacy, $\forall \lambda \geq 1$. Now, if we want the aggregated output to be $(m\epsilon, \delta)$-differentially private, the $\sigma^2$ in GNMax can be set as follows: 1) Since the above holds for all $\lambda \geq 1$, we first pick a proper $\lambda$ that does not cause numerical instability and that ensures $\sigma^2 > 0$ by setting $\lambda = \frac{\log 1/\delta}{\epsilon} + 5$. 2) Now set $\sigma^2 = \lambda/(\epsilon - \frac{\log 1/\delta}{\lambda - 1})$ by the above theorem.

