# OpenReview forum: "Optimized Tradeoffs for Private Majority Ensembling"
_ICLR.cc/2024/Conference — Submitted to ICLR 2024_

### Official Review · Reviewer_DeQA · 2023-10-21

**Soundness:** 3 good
**Presentation:** 3 good
**Contribution:** 3 good
**Rating:** 8
**Confidence:** 3

**Summary:**

Authors show that a private majority algorithm with maximal utility can be computed tractably under certain assumptions. They introduce a privacy framework characterized by a data-dependent noise function  called "Data-dependent Randomized Response Majority" (DaRRM) that allows for efficient utility optimization  subject to privacy constraints. Considerable theoretical results and some empirical evidence is presented.

**Strengths:**

The proposed framework called "Data-dependent Randomized Response Majority" (DaRRM) is interesting and innovative. There seems to be some significant breakthroughs arising from this framework. Designing the tuning parameter $\gamma$ with provable privacy amplification, and optimization for $\gamma$ are important developments as well.

Significant theoretical details have been established (although I dd not check the proof in details). The reported results from the experiments are plausible and seem reasonable.

**Weaknesses:**

The writing is dense in some parts. The technical assumptions and the mathematical details are not clearly stated in the main paper (although they can presumably be found in the supplementary materials), and hence the various theoretical results referring to Problem~1.1 lack adequate discussion and contextualization.

**Questions:**

While realizing that there is limited space, I would request authors to discuss the main technical assumptions they need for their theoretical results. This looks like a very solid work otherwise.

---

> ### Author Response · Authors · 2023-11-18
> **Response to reviewer comments**
>
> We would like to thank the reviewer for providing valuable and detailed feedback.
> The setting and the problem we consider are formally stated in Problem 1.1.
> The assumptions for theoretical results presented in section 4 are clearly stated at the beginning of this section, that is, we consider the setting when all mechanisms are i.i.d. under pure differential privacy guarantee.

---

> > ### Comment · Reviewer_DeQA · 2023-11-22
> > **Thank you**
> >
> > Thank you authors, for your response.

---

### Official Review · Reviewer_SRCJ · 2023-11-01

**Soundness:** 3 good
**Presentation:** 3 good
**Contribution:** 2 fair
**Rating:** 6
**Confidence:** 3

**Summary:**

This paper focuses on the problem of exploring the optimal utility of an (m\epsilon,\delta)-DP mechanism to compute the majority function under mild conditions. In this paper, the authors proposed the Data-dependent Randomized Response Majority (DaRRM) framework that approaches the problem of interest by improving the classical Randomized Response (RR) mechanism on the subsampling probability. The authors provides theoretical guarantees and compare the mechanism to the baselines empirically.

**Strengths:**

1. The paper is sound in theory and supported by empirical comparison with the state-of-art benchmarks.

2. The paper is well organized and written, and lays out its contributions clearly.

3. Empirical results in the paper showed that the proposed DaRRM framework outperforms the state-of-art benchmarks for different tasks.

**Weaknesses:**

Although the authors proposed the optimization procedure to tackle the problem of designing $\gamma$ in general DP setting, it is in general computationally intractable to optimize a set of $O(K^7)$ constraints in the $(\epsilon,\delta)$-DP setting.

**Questions:**

How will different priors of $p_i$ affect the result both theoretically and computationally? Have the authors try any experiments without assuming uniform distribution?

---

> ### Author Response · Authors · 2023-11-18
> **Response to reviewer comments**
>
> We would like to thank the reviewer for providing valuable and detailed feedback.
>
> Our results hold for any prior $p_i$. The key motivation in developing the $\text{DaRRM}$ framework is to automatically calibrate the noise added, i.e., the design of our $\gamma$ function, so that the utility can be maximized under the same privacy guarantee, without knowing the underlying distribution of $p_i$'s.
> We choose the uniform prior distribution of $p_i$ in our experiments to present results in the most general case when one does not have prior knowledge of $p_i$'s.
> If we have prior knowledge of the $p_i$'s and by incorporating the prior distribution of $p_i$ in the optimization framework, we can get even higher utility with the same privacy guarantee. Computationally, since we approximate the optimization objective (note this does not affect the privacy guarantee) by subsampling $p_i$'s from its distribution, using a different prior of $p_i$ does not slow down the run time.
>
> We added more experiment results in section F.1.3 to compare the error of $\text{DaRRM}_\gamma$ when $\gamma$ is optimized using the objective under
>
> $p_i \sim \text{Uniform}([0, 1])$
>
> as we did before and under another distribution $\mathcal{T} \sim \text{Uniform}([0, 0.3]\cup [0.7, 1])$, which represents one's prior belief that there is a clear majority among the mechanisms.
> We then compute $\text{DaRRM}_\gamma$'s error in three settings with three different actual $p_i$ distributions: 1. $p_i \sim \text{Uniform}([0, 1])$,$\forall i\in [K]$, 2. $p_i \sim \text{Uniform}([0, 0.1])$, $\forall i\in [K]$, implying a clear majority of 0 among the mechanisms, and 3. $p_i = 0.5, \forall i\in [K]$, implying there is no clear majority.
>
> As one can see from the results, if the prior distribution of $p_i$ is close to the actual distribution, optimizing $\gamma$ under this prior distribution leads to additional utility gain (i.e., less error); while if the prior distribution of $p_i$ is far from the actual distribution, optimizing $\gamma$ under the prior distribution is not as optimal but still better than naive methods. We find that using a uniform prior generally leads to good performance.
>
> Please see the updated draft for more results.

---

> > ### Comment · Reviewer_SRCJ · 2023-11-21
> >
> > Thanks for the response. After reading the reviews and responses, I decide to keep my current score.

---

### Official Review · Reviewer_pyVN · 2023-11-01

**Soundness:** 3 good
**Presentation:** 2 fair
**Contribution:** 3 good
**Rating:** 3
**Confidence:** 3

**Summary:**

The paper studies an intriguing question: If we have K $(\epsilon, \delta)$-DP mechanisms, can we release their majority vote with a privacy guarantee that's better than $(K\epsilon, K\delta)$-DP? This paper focuses on the binary voting scenario, and presents a data-dependent randomized response mechanism where the probability of releasing the true majority vote is based on the count of the actually majority vote. To maximize the utility, the paper identifies the worst-case distribution pair and reduces the problem into a constraint optimization problem which can be solved in acceptable runtime.

**Strengths:**

This work identifies a very interesting problem. Releasing the majority voting for an ensemble of DP mechanisms is a very good extension for the well-known private selection problem (release the index of the maximum among an ensemble of DP mechanisms.

The formulation of a semi-infinite program for maximizing the utility sounds a very interesting technique, and the author shows it's practical through Gurobi.

**Weaknesses:**

The writing of the paper can be improved. For example:
- in the last line of Problem 1.1, $S_i$ does not need $(D)$.
- line 6 of Algorithm 1 should be $S_i$ instead of $M_i$.
- Lemma 3.2: have you defined $f$?
- Lemma 3.3: there is another $f$.

I am quite worried about the experiment. For Figure 3, could you plot privacy-utility tradeoff instead? (i.e., change the x-axis from communication rounds to $\epsilon$). The same thing also applies to Table 1. For DP experiments, the comparison is only fair when the $\epsilon$ of different techniques are aligned to be the same.

What is the dimension of the gradient? And how is the privacy parameter for releasing a single dimension being composed? For sign-sgd experiment, the total number of composition is gradient dimension x number of rounds, which seems to be very very large and I am not sure what's the final privacy parameter.

Also, it seems the author assumes each client trains DP models, but for PATE each teacher does not need to be trained differentially private. Therefore, I am not sure whether the comparison in the experiment is fair given that each teacher is already DP but still adds the same noise as what is stated in PATE. Can the author provide a comparison with the original version of PATE for both experiments in Section 6.2 and 6.3?

**Questions:**

Is it possible to extend the proposed framework to a multi-class setting? (this does not affect the score given the technical contribution of the paper is rich enough, but I am curious about authors' opinion)

Could you provide some intuition for Theorem 4.1? Especially why when $m > K/2$ the privacy improvement seems to be automatically applied?

---

> ### Author Response · Authors · 2023-11-15
> **Response to reviewer comments**
>
> We would like to thank the reviewer for providing valuable and detailed feedback.
> We have uploaded an updated version the draft to reflect changes to improve the presentation. All modified places in the draft are colored in blue.
>
> $\textbf{Response to Weakness. }$
>
> $\textbf{Writing.}$
>
> - Point 1 \& 2 \& 3 See the updated draft.
>  - Point 4: The $f$ is defined in Lemma 3.3. To make it clearer, we changed $f(...)=$ to be $f(...) :=$.
>
> $\textbf{Experiments. }$
>
> In the experiments on private Sign SGD, we fix the total privacy loss of each communication around privacy loss to be $m\epsilon = 0.3$. The total privacy loss for each coordinate in the gradient vector is composed across $T$ iterations, and the resulting total privacy loss across all communication rounds, i.e., the overall $\epsilon$ parameter, is the same for each private majority ensembling technique we use in the experiments.
> We used two simple CNN models on the two datasets \texttt{MNIST} and \texttt{CIFAR10} with $28938$ and $9074218$ parameters
> Hence, the dimension of the gradient vector here is $d = 28938$ on \texttt{MNIST} and $d = 9074218$ on \texttt{CIFAR10}.
> Since the dimension $d$ for both datasets are different, we think making a plot with x-axis as the \# communication rounds is clearer.
>
> You are correct that the total number of composition is gradient dimension $d$ $\times$ \# rounds, which can be very large.
> This is indeed a drawback of private Sign SGD (which guarantees pure DP), compared to the more widely applied DP-SGD (which only comes with approximate DP guarantees), although people use private Sign SGD instead of DP-SGD to combat Byzantine adversaries.
> Designing Byzantine resilient and private optimization algorithm with good privacy-utility trade-offs is out of the scope of this work, but can be interesting to think about.
>
> We emphasize that we only use the base version of private Sign SGD as an application to demonstrate the usage of our theory and the $\text{DaRRM}$ framework; more advanced versions of Sign SGD with higher utility that exploit sparsity and round skipping will also benefit from our framework. For example, as we have mentioned in the footnote on page 8, to limit the privacy loss, one can apply sparsification techniques to reduce the number of gradient coordinates sent by the clients.
>
> As the reviewer mentioned, our setting and PATE's setting are different and are not directly comparable. PATE aggregates outputs from non-private teachers while we consider private teachers, while our $\text{DaRRM}$ framework does not apply when the teachers are non-private. In our setting, when the teachers are private, the ensembling guarantee of PATE still applies as it relies on the typical sensitivity analysis, yet gives much worse utility guarantee, since it cannot exploit the fact the teachers are private themselves. Therefore, we show that the naive PATE extension is suboptimal but there is no way to extend our algorithms into the original PATE setting, and comparing to ensembling non-private teachers in term of downstream accuracy is outside the scope of our setting. Note that the main goal is to show that the typical sensitivity-based noise injection, via "Laplacian" and "GNMax" which are two subroutines used in PATE, are suboptimal in terms of ensembling utility and can be used as naive baselines for majority ensembling in the experiments.

---

> > ### Author Response · Authors · 2023-11-15
> > **Response to reviewer comments (cont'd)**
> >
> > $\textbf{Response to Questions. }$
> >
> > $\textbf{Extension to multi-class output.}$
> >
> > Extension to multi-class output is feasible but there are a few challenges.
> >
> > Suppose we have $K$ mechanisms, each outputting a label in $T$ classes, i.e., $M_i: \mathcal{D} \rightarrow \{0,1,\dots,T - 1\}$.
> > In the binary output case, the support of $\gamma$ is simply $\{0,1,\dots, K\}$, which indicates the number mechanisms voting for class 1. However, if there are $T$ classes, the support of $\gamma$ needs to be $(v_1, v_2,\dots,v_{T-1}) \in \mathbb{N}^{T-1}$, which specifies the number of outputs in $T-1$ classes. That is, $\gamma: \mathbb{N}^{T-1} \rightarrow [0,1]$.
> >
> > The $\text{DaRRM}$ algorithm generally follows the one for binary output, except that after we obtain samples  $\mathcal{S} = \{S_1, \dots, S_K\}$ from the mechanisms, we compute a vector $\mathbf{v} = (v_1,\dots, v_{T-1})$, where $v_t = \sum_{i=1}^{K} \mathbb{I}$ {$S_i = t$}, $\forall t\in [T-1]$. The probability $p_\gamma$ is set by $\gamma(\mathbf{v})$, and with probability $p_\gamma$ we output the true majority; otherwise, we output a random class.
> >
> > The privacy objective $f$, which is later used as the privacy constraint in optimizing $\gamma$, is harder to compute when the output is multi-class.
> > Specifically, one needs to compute
> >
> > $\Pr[\text{DaRRM}_\gamma(\mathcal{D}) = t]$ for some class $t$ on dataset $\mathcal{D}$.
> >
> > This is now
> >
> > $\sum_{ \text{all possible } (v_1, v_2,\dots,v_{T-1})} \Pr[ \text{DaRRM}_\gamma(\mathcal{D}) = t \mid$
> >
> > $\mathbf{v} = (v_1, v_2, \dots, v_{T-1}) ] \cdot \Pr[\mathbf{v} = (v_1,v_2,\dots,v_{T-1})]$.
> >
> > Also, since the support of $\gamma$ is now $(T-1)$-dim, we will now need to optimize $(T-1)\cdot (K+1)$ variables, instead of $(K+1)$ variables as in the binary output case.
> > For simplicity, here we do not take into account the symmetry property of $\gamma$.
> >
> > Designing more efficient optimization algorithms to find the best $\gamma$ in the multi-class output case is an interesting future direction.
> >
> > $\textbf{Intuition of Theorem 4.1. }$
> >
> > We have replaced the proof sketch of Theorem 4.1 with more intuition (in blue). Please see the updated draft.

---

> ### Comment · Reviewer_pyVN · 2023-11-21
>
> Thanks for the response!
>
> In your response, you write "we fix the total privacy loss of each communication around privacy loss to be 0.3". However, in your Figure 3's caption, you write "ensuring that each coordinate of the aggregated sign gradient is 0.3-differentially private". So does it mean for each round, the privacy loss is 0.3*d-DP? Then even just 1 round will be more than 8000-DP? I would say such an epsilon is just meaningless. Or you mean each coordinate aggregation is 0.3/d-DP? Given the magnitude of d, it's almost 0-DP and it's hard to believe there will be any utility.
>
> Furthermore, in Appendix, I saw "each client computes its gradient based on the
> entire local dataset at each communication around", so is the privacy calculation correct in the paper? What does it even mean by "each ensemble itself is $\varepsilon$-DP" in this case? (if the dataset for training different ensemble is different, the naive privacy loss is no longer $m \varepsilon$-DP)
>
> Although I understand that the paper is considering the case that each teacher is already $(\varepsilon, \delta)$-DP, I think it's beneficial to add a comparison with original PATE. If the original PATE significantly outperforms aggregating private teachers, then what is the main motivation for this work? E.g., are there scenarios where we must have differentially private teachers and cannot have non-private teachers?

---

### Official Review · Reviewer_3EZ4 · 2023-11-03

**Soundness:** 2 fair
**Presentation:** 2 fair
**Contribution:** 3 good
**Rating:** 3
**Confidence:** 4

**Summary:**

This paper studies how differentially private is the majority vote of K (epsilon,delta)-DP classifiers.

**Strengths:**

The paper studies the interesting question of reducing the cost of majority voting in terms of privacy budget, in particular, a very naive approach would say that $K$ classifiers are computed, and hence we can say (by the post-processing property) the majority vote is $(K\epsilon,\delta)$-DP.  The paper studies the interesting question whether we can have a noisy majority vote which is $m\epsilon$-DP with $m<K$.

**Weaknesses:**

While the problem studied is interesting, the text is insufficiently rigorous.

The main idea of the text is mostly clear, but sometimes overloading of notations and other elements lead to confusion.  For example:
* In Algorithm 1 line 4, $\gamma$ is a real number
* In Eq (1), $\gamma$ is a function $\gamma:\mathbb{N}\to \{0,1\}$
* In Lemma 3.2, $\gamma$ is a function taking as input $S\in\{0,1\}^K$
* In Theorem 4.1, $\gamma$ is a function $\gamma:\mathbb{N}\to\{0,1\}$
* In Eq (3), $\gamma$ is a tuple of $(K+1)/2$ (not $K$) real numbers (not booleans), i.e., $\gamma\in[0,1]^{(K+1)/2}$

While the $\gamma$ confusion is not very harmful, some issues are more problematic, e.g., the formulation of Theorem 4.1:
* Theorem 4.1 says: "Consider Problem 1.1 when $p_i = p$, ..." but Problem 1.1 does not feature a variable $p_i$, $p_i^\prime$,
* Theorem 4.1 says: "Given a privacy budget $m \in [K]$", but usually $\epsilon$ is called a privacy budget and $\epsilon$ is rarely restricted to be an integer.  Is $m$ really to be interpreted as a privacy budget?  In fact, according to Algorithm 1, $m\epsilon$ (not just m) is the "target privacy cost".
* Theorem 4.1 says "Given a privacy budget $m$, if one sets $\gamma(l) = ...$ when $m\ge (K+1)/2$".  The sentence seems ill-structured, I suppose you mean "if" rather than "when", and it becomes easier to break the sentence down into smaller parts, i.e., "Let $m\in[K]$.  If $m\ge (K+1)/2" then set $\gamma(l) = ...$, else ....".
* Theorem 4.1 mentions $\gamma$, which does not occur in Problem 1.1, while Problem 1.1 mentions the majority function g not used in Theorem 4.1.

Other notation and related issues:
* Lemma 3.1 says $\gamma_{subsampling}$ depends on $\mathcal{L}$ and next defines $\gamma_{subsampling(l)$ rather than $\gamma_{subsampling}(\mathcal{L})$.
* Eq 1 defines both $\gamma$ and $\gamma_{subsampling}$.  What is the difference in meaning between the two notations?
* In Lemma 3.3, what are $\mathcal{D}$ and $\mathcal{D}'$ ?  Are these arbitrary datasets, or do you assume they are adjacent datasets?
* In Lemma 3.3, $\gamma$ is required to be a "symmetric function".  What does this mean exactly?  I guess that the meaning of "symmetric" depends on the signature of $\gamma$, for example, if $\gamma$ is a function of a single variable, we could call a function for which $\gamma(l)=\gamma(K-l)$ symmetric.  If $\gamma$ is a function of tuples $S$, then we could call $\gamma$ symmetric if it is invariant under permuting $S$.

There are a few minor language issues, e.g.,
* Title of Section 5: "Optimizing $\gamma$-function" -> "Optimizing the function $\gamma$"
* Section 5: "On the other hand, one can to optimize for such $\gamma^*$ but this involves solving “Semi-infinite Programming”, due to the infinitely many privacy constraints.":  Especially the first part of the sentence doesn't make sense grammatically.  Also, you probably mean 'solving a semi-infinite programming problem' rather than 'solving "semi-infinite programming"'.

Some claims are too optimistic.  For example, Lemma 3.3 says " ... is $(m\epsilon, \delta)$-differentially private if and only if ...".  There doesn't seem to be a proof (in particular, Appendix B is about preliminaries while appendix C is already about Section 4).  I doubt a proof can exist for this statement, since under the provided conditions DaRRM_\gamma could be $(m\epsilon,\delta$-DP even if Eq (2) is not satisfied, for example if the individual mechanisms $M_i$ are $\epsilon'$-DP for some $\epsilon' < \epsilon$, or if the mechanisms $M_i$ satisfy other favorable conditions.  So while I believe one can prove "if", I don't believe one can also prove "and only if".

Similar issues arise in the appendices, which provide a number of proofs but with insufficient rigor to easily determine what is the exact claim nor to easily verify them.

**Questions:**

--

**Details Of Ethics Concerns:**

--

---

> ### Author Response · Authors · 2023-11-15
> **Response to reviewer comments**
>
> We would like to thank the reviewer for providing valuable and detailed feedback.
> We have uploaded an updated version the draft to reflect changes to improve the presentation. All modified places in the draft are colored in blue.
>
> $\textbf{Overloading of $\gamma$ notations. }$
> - We updated Algorithm 1 by introducing a different notation, $p_\gamma$, as the coin-flipping probability.
> - In most places in the draft, $\gamma$ is the function $\gamma: \{0,1,\dots,K\} \rightarrow [0, 1]$, except in Lemma 3.2, where
>     we show the generality of $\text{DaRRM}$ holds with an even more general $\gamma: \{0, 1 \}^{K+1} \rightarrow [0, 1]$.
> - To make the $\gamma$ notation consistent, we rephrased the optimization problem in Eq.3 so that the variable is now $\gamma \in [0, 1]^{K+1}$, with an additional constraint saying $\gamma$ is symmetric around $\frac{K+1}{2}$.
>
> $\textbf{The formulation of Theorem 4.1.}$
> - Variables $p_i, p_i'$ are defined in "Notations", which presents all notations used throughout the paper as is stated in this section "throughout the paper, we use ...", in section 2.1 "preliminaries".
> - To avoid confusion, we now call $m$ the privacy allowance and have updated this in the draft, including the Appendix. $m\epsilon$ now refers to the usual notion of privacy budget. Just to be clear, $\epsilon$ is never an integer in the context. $m$ does not have to be an integer in our context either. However, the total privacy loss of a subsampling algorithm is an integer multiplier of $\epsilon$ by nature. For a privacy allowance $m \in (i, i+1)$, $i\in \mathbb{N}^{+}$, we can only subsample $i$ items to ensure the privacy loss is within the total budget. Hence, to simplify the presentation, we consider an integer $m$ in the context of the subsampling algorithm, as in Theorem 4.1.
> - We have slightly re-structured Theorem 4.1 to make it clearer. See the updated draft.
> - The goal is to design $\gamma$ in our proposed $\text{DaRRM}$ framework to solve Problem 1.1. We have updated the theorem statement to make it clearer.
>
> $\textbf{Other notation and related issues. }$
> - Point 1 \& 2, we updated the statement of Lemma 3.1, changing $\mathcal{L} \Rightarrow l$ and $\gamma \Rightarrow \gamma_{Subsampling}$ as suggested.
> - $\mathcal{D}$ and $\mathcal{D'}$ are defined as the adjacent datasets in the "notation" section in section 2.1 "preliminaries". We also re-stated that $\mathcal{D}$ and $\mathcal{D'}$ are adjacent datasets in the paragraph right before introducing Lemma 3.3, in the original draft.
> - In Lemma 3.3, it is stated "For $\gamma$ that is a symmetric function of the sample sum", which implies "symmetric" meaning $\gamma(l) = \gamma(K-l)$, where $l$ denotes the sample sum, i.e., $\sum_{i=1}^{K} S_i$. To make it clearer, we updated the draft by replacing this sentence with "For $\gamma: \{0,1,\dots,K\}\rightarrow [0, 1]$ such that $\gamma(l) = \gamma(K-l),\forall l,$"
>
> $\textbf{Minor language issues. }$
>
> We have corrected those issues in the updated draft.
>
> $\textbf{Response to Lemma 3.3 being too optimistic. }$
>
> Lemma 3.3 indeed indicates "if and only if". This is because the statement directly follows the definition of differential privacy and applies only algebraic manipulations. We note that for the "only if" direction, the condition that we assume is that DaRRM is private for any $\epsilon$-private mechanisms; therefore even if favorable conditions on the mechanisms occur, our privacy cost inequality will still hold. We slightly modified and simplified part of the proof of Lemma 3.3 to emphasize both "if" and "only if" directions. See the updated proof in section C.4. Note that the same lemma statement as in Lemma 3.1 followed by its proof was indeed in section C.4 in the Appendix of the original draft, as promised in the main paper. The confusing part might be that we manually numbered all lemmas in the Appendix and there was a mismatch in the lemma numbers (although the title of each section in the Appendix has the correct lemma number).
> We corrected all lemma numbers in the Appendix.

---

> > ### Comment · Reviewer_3EZ4 · 2023-11-15
> >
> > Thanks for your answer and the clarifications.
> > I see that when pointed to insufficiently rigorous parts of the text, the authors can improve the text significantly.
> > As said in my review, the lack of rigor is a general issue in the text.  I have provided some detailed comments on a few sampled sections, but similar issues also exist most other sections (including, as said, the proofs in appendix).  I hence suggest the authors take the time to go step by step over the complete text to make everything rigorous and clear.

---

> ### Author Response · Authors · 2023-11-18
> **Response to reviewer comments**
>
> We have made a pass through the entire draft, including the main paper as well as all theorems/lemmas/proofs in the Appendix. We tried our best to make all theorems/lemmas/proofs as clear and rigorous as possible. Please see the updated draft. Thanks.

---

### Meta-Review · Area_Chair_PAV5 · 2023-12-06

**Metareview:**

The paper provides a framework to privately compute the majority vote of multiple classifiers using a data-dependent noise function.

Pros: Reviewers generally find the problem interesting and were impressed by the nice theoretical guarantee.

Cons: The main criticism is the quality of presentation, as commented by three reviewers. This is specifically unsatisfactory for a paper in which theoretical results are main contributions. Additional, Reviewer pyVN also concerned the design of experiments.

The rebuttal went well---authors responded to reviewers questions in details and reviewers acknowledged authors' efforts. There have been some discussions among the reviewers on whether the significance of the results outweigh the presentational issues. Eventually everyone agreed with rejection so that the authors can improve the paper to get more attention once it is published.

**Justification For Why Not Higher Score:**

No reviewers seems to be able to confidently verify the correctness of the results. So, quality of presentation is indeed a valid reason to reject the paper.

**Justification For Why Not Lower Score:**

N/A

---

### Decision · Program_Chairs · 2024-01-16

Reject